Resource

# Cooperative genetic networks drive embryonic stem cell transition from naïve to formative pluripotency

Andreas Lackner[1,†,§] iD, Robert Sehlke[2,†], Marius Garmhausen[2,†] iD, Giuliano Giuseppe Stirparo[3,4,†] iD, Michelle Huth[1,‡] iD, Fabian Titz-Teixeira[2,‡] iD, Petra van der Lelij[1] iD, Julia Ramesmayer[1], Henry F Thomas[1], Meryem Ralser[3], Laura Santini[1] iD, Elena Galimberti[1], Mihail Sarov[5], A Francis Stewart[5,6], Austin Smith[3,4] iD, Andreas Beyer[2,7,*] iD & Martin Leeb[1,**] iD

## Abstract

In the mammalian embryo, epiblast cells must exit the naïve state and acquire formative pluripotency. This cell state transition is recapitulated by mouse embryonic stem cells (ESCs), which undergo pluripotency progression in defined conditions *in vitro*. However, our understanding of the molecular cascades and gene networks involved in the exit from naïve pluripotency remains fragmentary. Here, we employed a combination of genetic screens in haploid ESCs, CRISPR/Cas9 gene disruption, large-scale transcriptomics and computational systems biology to delineate the regulatory circuits governing naïve state exit. Transcriptome profiles for 73 ESC lines deficient for regulators of the exit from naïve pluripotency predominantly manifest delays on the trajectory from naïve to formative epiblast. We find that gene networks operative in ESCs are also active during transition from pre- to post-implantation epiblast *in utero*. We identified 496 naïve state-associated genes tightly connected to the *in vivo* epiblast state transition and largely conserved in primate embryos. Integrated analysis of mutant transcriptomes revealed funnelling of multiple gene activities into discrete regulatory modules. Finally, we delineate how intersections with signalling pathways direct this pivotal mammalian cell state transition.

**Keywords** exit from naïve pluripotency; haploid ES cells; naïve to formative transition; signalling; systems biology
**Subject Categories** Chromatin, Transcription & Genomics; Methods & Resources; Stem Cells & Regenerative Medicine

**The EMBO Journal (2021) 40: e105776**

## Introduction

Mouse embryonic stem cells (ESCs) can self-renew in defined conditions in a state of naïve pluripotency (Smith, 2017). The ESC exit from naïve pluripotency provides an amenable experimental system for dissection of a cell fate decision paradigm (Buecker *et al*, 2014; Kalkan *et al*, 2017). Naïve pluripotency is under control of a gene regulatory network (GRN) containing the core pluripotency transcription factors (TFs) *Pou5f1*, *Sox2* and naïve-specific TFs such as *Nanog*, *Esrrb*, *Klf4* and others (Chen *et al*, 2008; Dunn *et al*, 2014; Niwa 2018). In defined cell culture conditions that include inhibitors against Mek1/2 (PD0325901) and Gsk3 (CHIR990201, CH; collectively termed "2i"), ESCs can be homogenously maintained in the naïve state (Ying *et al*, 2008). Within 24–36 h after withdrawal of 2i, ESCs transit into formative pluripotency, entirely losing naïve identity (Kalkan *et al*, 2017). During this transition, the naïve GRN is extinguished and expression of formative factors such as *Otx2*, *Pou3f1*, *Dnmt3a/b* and *Fgf5* is initiated. A similar transition is evident during peri-implantation development, where the TF network maintaining naïve pluripotency dissolves between embryonic day (E) 4.5 and E5.5 (Boroviak *et al*, 2014; Acampora *et al*, 2016; Mohammed *et al*, 2017). The speed of the naïve to formative GRN transition is notable because (i) the cell cycle is around 12 h

1   Max Perutz Laboratories Vienna, University of Vienna, Vienna Biocenter, Vienna, Austria
2   Cologne Excellence Cluster Cellular Stress Response in Aging-Associated Diseases (CECAD), University of Cologne, Cologne, Germany
3   Wellcome - MRC Cambridge Stem Cell Institute, University of Cambridge, Cambridge, UK
4   Living Systems Institute, University of Exeter, Exeter, UK
5   Max Planck Institute of Molecular Cell Biology and Genetics, Dresden, Germany
6   Biotechnology Center, Center for Molecular and Cellular Bioengineering, Technische Universität Dresden, Dresden, Germany
7   Center for Molecular Medicine (CMMC), University of Cologne, Cologne, Germany
    *Corresponding author. Tel: +49 221 478 0; E-mail: andreas.beyer@uni-koeln.de
    **Corresponding author. Tel: +43 1 4277 74644; E-mail: martin.leeb@univie.ac.at
    †These authors contributed equally to this work as first authors.
    ‡These authors contributed equally to this work.
    §Present address: Center for Anatomy and Cell Biology, Medical University of Vienna, Vienna, Austria

long; (ii) all factors that are required to establish and maintain naïve pluripotency are expressed robustly in naïve cells; and (iii) the naïve pluripotency network is recursively self-reinforcing. The rapid dissolution of naïve pluripotency implies the existence of circuit-breaking mechanisms. In recent years, we and others have identified various factors promoting ESC differentiation using screens in haploid and diploid ES cells (Guo *et al*, 2011; Leeb & Wutz, 2011; Betschinger *et al*, 2013; Leeb *et al*, 2014; Li *et al*, 2018).

Robust assays employing ESCs expressing a Rex1 promoter-driven destabilised GFP reporter (Rex1::GFPd2) enable the dissection of the exit from naïve pluripotency in high resolution (Kalkan *et al*, 2017; Mulas *et al*, 2017). Rex1-GFP downregulation is initiated within 24h after 2i withdrawal (N24) and completed after 48h (N48). Nevertheless, the exact nature, mechanistic underpinnings and sequence of events during exit from naïve pluripotency remain only partially understood. In particular, we lack insight into how the different molecular components of the system co-operate to elicit proper cell fate transition. Here, we have driven a Rex1-GFP reporter screen to saturation, thus providing an extensive list of genes and pathways involved in the exit from naïve pluripotency. We utilised this information in a systems biology approach to explore regulatory principles of the exit from naïve pluripotency. To evaluate dependencies and causal relationships within the pluripotency and differentiation circuitries, we probed the response of the differentiation programme to a comprehensive series of exit factor gene knockouts. Through computational integration of molecular profiling data with regulatory networks and *in vivo* GRN trajectories, we expose the regulatory foundations of a cell fate choice paradigm at a pivotal junction in early mammalian development.

# Results

### Haploid ES cell saturation screen

Haploid ES cells are an efficient platform for insertional mutagenesis-based screens (Elling *et al*, 2011; Leeb & Wutz, 2011; Kokubu & Takeda, 2014). We previously reported a medium-scale screen comprising approximately $5 \times 10^4$ mutagenic events to identify factors regulating the exit from naïve pluripotency (Leeb *et al*, 2014).

We have now driven this approach to saturation by assaying approximately 1.2 million mutations in receptive genomic regions that cause delays in Rex1 downregulation in two independent Rex1-reporter cell lines (Fig 1A and B), utilising three different mutagenic transposon vectors in 35 independent screens (Fig 1C and D).

Stringent filtering resulted in a candidate list comprising 489 genes (Dataset EV1). These screens generated a candidate inventory of the machinery that mediates exit from naïve pluripotency. Reassuringly, the known exit from naïve pluripotency regulators *Tcf7l1*, *Fgfr2*, *Jarid2* and *Mapk1* (Erk2) (Kalkan & Smith, 2014) were among the highest ranked genes (Appendix Fig S1A). The candidate hit genes were enriched for processes involved in transcription regulation, epigenetics and signalling-related functions (Appendix Fig S1B and C; Dataset EV1), as well as RNA-binding functions in-line with emerging evidence of RNA regulatory mechanisms in cell fate control (Ye & Blelloch, 2014).

Many identified genes are not specific to pluripotency and have functions in common pathways and processes, implying that the exit from naïve pluripotency utilises widely expressed cellular machinery. Therefore, mechanisms mediating ESC transition might also be utilised in other differentiation processes.

### Establishment of a mutant ESC library for systematic transcriptional profiling

To characterise deficiencies in naïve exit in molecular detail, we generated KO ESC lines deficient for 73 selected genes, comprising top ranked genes from the mutagenic screen. We also included components from pathways and protein complexes for which multiple members were recovered, even if just below the cut-off threshold (e.g. the Paf complex member *Leo1*, the mTORC1 regulator *Tsc2* and the NMD component *Smg6*), and *Mbd3*, *Zfp281* and *L3mbtl3* as known players in the exit from naïve pluripotency (Betschinger *et al*, 2013). Three control genes were included that are either not expressed in ES cells (Nestin), expected to be neutral (Hprt) or whose ablation was expected to accelerate differentiation (c-Myc). Paired gRNAs were used to disrupt target genes in a diploid biparental Rex1::GFPd2 reporter ES cell line carrying a Cas9 transgene (henceforth termed RC9 cells) (Fig 1A) (Li *et al*, 2018). Following an efficient parallelised approach, we established passage matched and isogenic homozygous KO cell lines, thus maximising

---

**Figure 1. Establishment of exit factor deficient KO ESC lines informed by a haploid ESC saturation screen.**

A   Illustration of the Rex1-GFPd2 reporter cell line and its exit from naïve pluripotency. Rex1-GFPd2 (in short Rex1-GFP) expression is tightly linked to naïve pluripotency. Shutdown of GFP expression indicates commitment to differentiation.

B   FACS analysis of Rex1-GFP reporter levels throughout a 72h differentiation time course after 2i withdrawal.

C   Scheme of the screening strategy to identify candidate genes involved in the exit from naïve pluripotency. After random insertional mutagenesis using piggyBac transposon-based gene-trap vectors, haploid Rex1-GFP ESCs were released into differentiation. Cells maintaining GFP expression after exposure to differentiation conditions were isolated and the gene-trap insertion sites mapped.

D   The cumulative number of hits (red) and the cumulative number of novel hits (blue) in 35 independent insertional mutagenesis screens in haploid ESCs are shown.

E   Representative Rex1-GFP FACS plots showing the differentiation delays 24 h after 2i withdrawal of *Tcf7l1*, *Rbpj*, *Trim71*, *Smg5* and *Pten* KOs. A *Myc* KO served as a negative control. Blue indicates the Rex1-GFP FACS profiles for KO, and dashed lines indicate WT.

F   Differential expression of genes at N24 versus 2i in WT RC9 cells. Black dots show significance (FDR ≤ 0.05, H0: |log2FC| < log2(1.5)). Pluripotency genes are red dots, formative genes are orange dots, haploid screen hits are blue dots, and the 73 KO genes are green dots.

G   t-SNE projection of the 73 KOs in 2i, based on expression of the 3068 differentially expressed genes between N24 and 2i in WT. The strength of differentiation delay observed at N24 in the respective KOs are indicated by a colour gradient and measured as average naïve marker log2 fold change (log2 FC, based on expression levels of *Esrrb*, *Nanog*, *Tfcp2l1*, *Tbx3*, *Prdm14* and *Klf4*, *Zfp42*) in the respective KO at N24. Red: delayed differentiation; blue: accelerated differentiation.

H   Similar to (G) for KOs at N24.

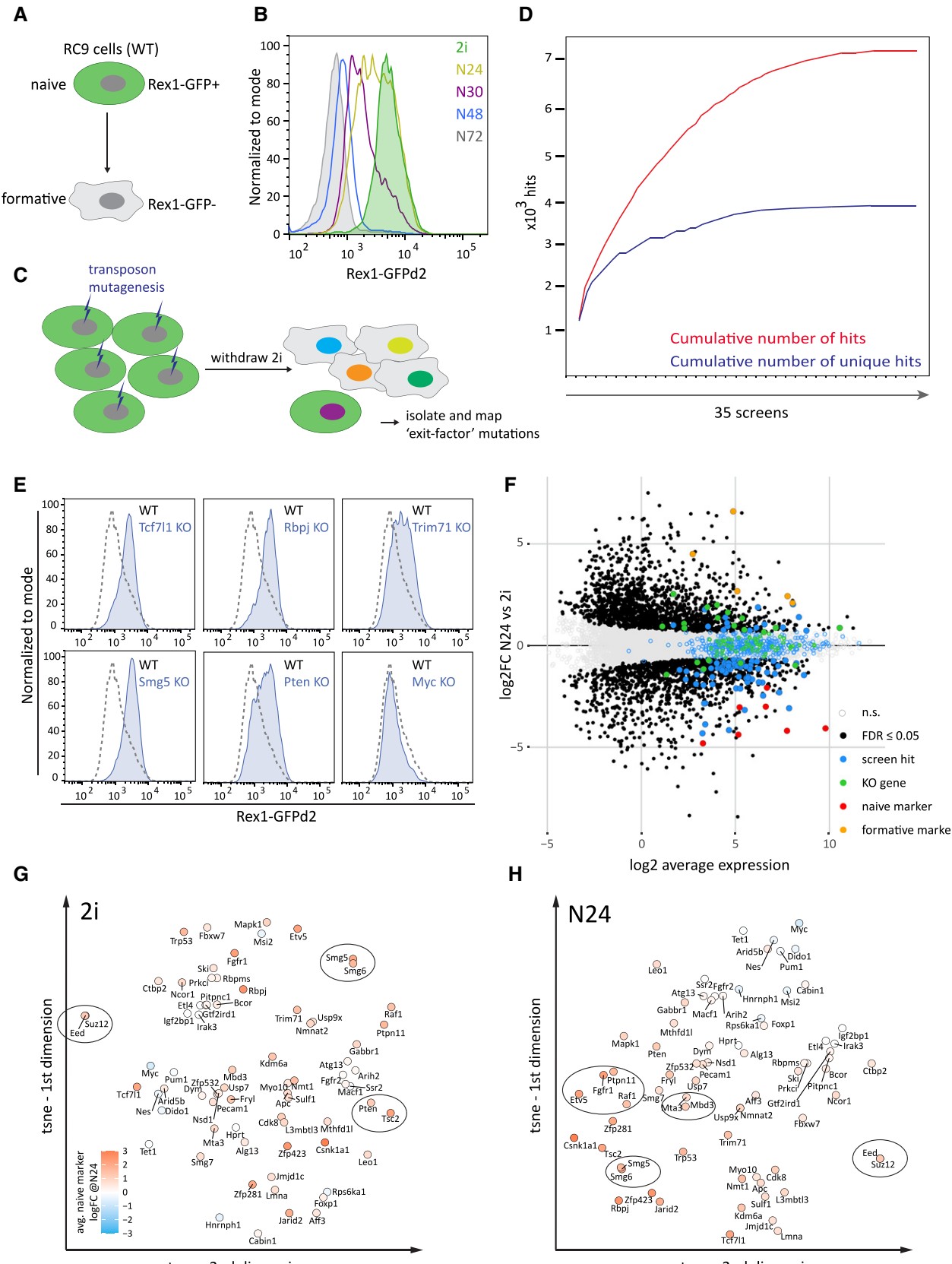

**Figure 1.**

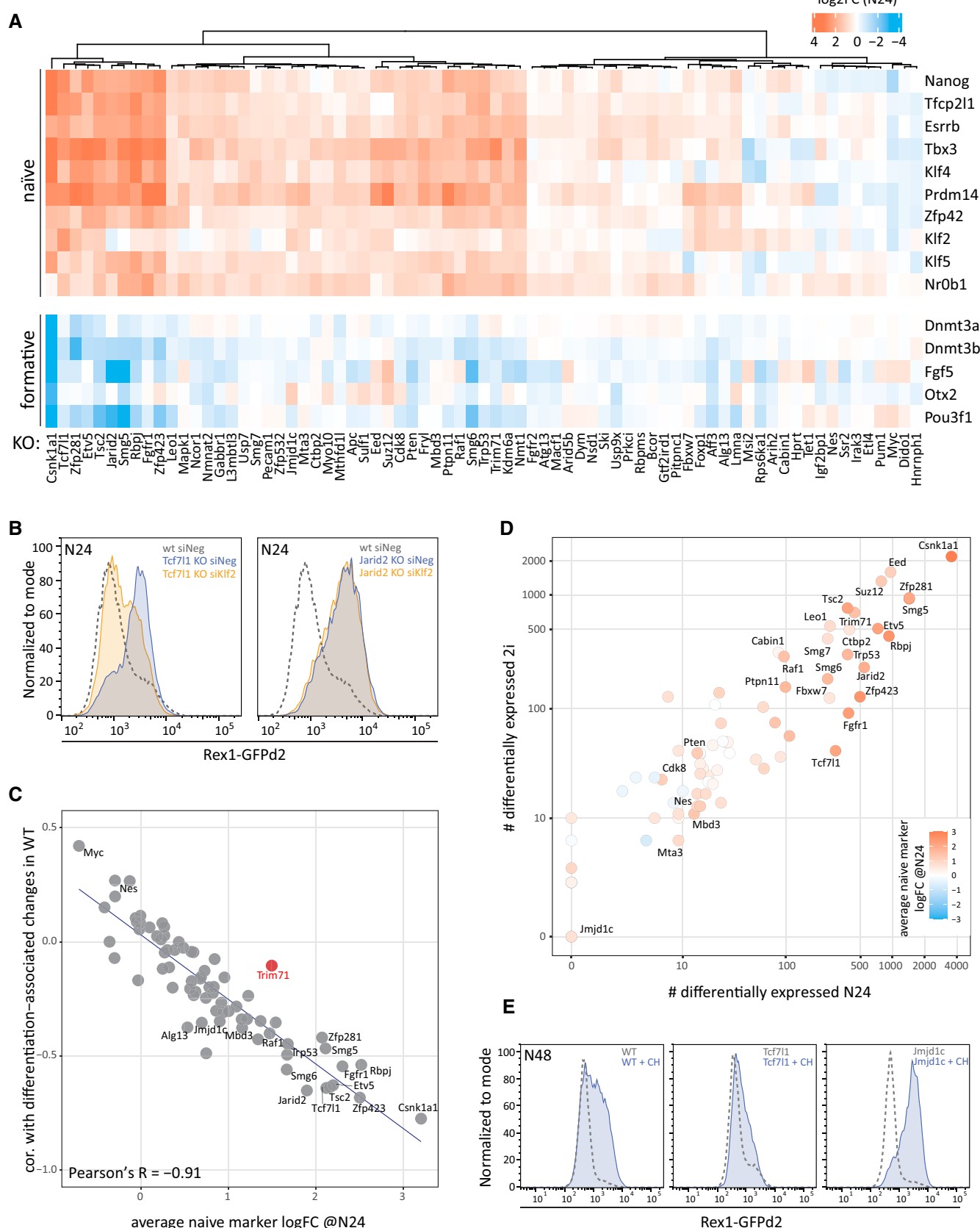

**Figure 2.**

**Figure 2. Systematic transcriptional profiling of a mutant ESC library.**

A  Naïve (top) and formative (bottom) marker gene expression changes at N24 compared with WT in all 73 KO ESCs. Clustering based on naïve marker gene expression shows a wide distribution of differentiation delay phenotypes.

B  Rex1-GFP FACS analysis of *Jarid2* and *Tcf7l1* KO ESCs after transfection with negative control or *Klf2*-specific siRNAs.

C  Plot showing a comparison of the log2 fold change (KO versus WT) of mean naïve marker gene expression at N24 to the extent of change of the global transcriptome between 2i and N24 (defined as the correlation of log2FCs in WT differentiation and log2FCs in KOs at N24). Each dot corresponds to one of the 73 KOs. Pearson's correlation is shown in the plot.

D  Comparison of the number of genes deregulated in 2i and at N24 in all KOs (FDR ≤ 0.05, H0: |log2FC| < log2(1.5)). Differentiation phenotypes are colour coded according to average naïve marker log2FCs in KOs at N24.

E  Rex1-GFP FACS analysis of WT, *Jmjd1c* and *Tcf7l1* KO cells at N48 cultured with and without the Gsk3 inhibitor CH.

comparability (Appendix Fig S1D and E). All KOs were validated by genomic PCR, followed by Sanger sequencing when required.

Full protein deficiency was validated for 14 KOs (*Eed, Suz12, Jarid2, Kdm6a, Smg5, Smg6, Smg7, Tsc2, Pten, Raf1, Tcf7l1, Leo1, Nmt1* and *Csnk1a1*) (Appendix Fig S2A). Only heterozygote clones could be generated for *Mapk1* (Erk2), resulting in reduced protein levels. Notably, increased levels of Erk1 in Erk2 heterozygous mutant cells failed to rescue the strong differentiation delay in the *Mapk1^het* KOs. For five further genes (*Alg13, Dido1, Msi2, Etv5, Jmjd1c*), we confirmed the absence of the corresponding transcripts or specific out of frame deletion of an exon by RT–qPCR or Sanger sequencing of RT–PCR products. Successful rescue experiments using 3xflag-tagged transgenes for six genes (*Rbpj, Etv5, Fgfr1, Jarid2, Mbd3* and *Tcf7l1*) established causality between the observed genotype and phenotype (Appendix Fig S2B and C). Thus, all tested knockouts showed the expected impact on RNA or protein expression. However, we cannot exclude the possibility of hypomorphic phenotypes in some cases.

ESC differentiation behaviour is highly dependent on cell density and timing of medium changes. To enable robust comparison of the differentiation of multiple KO ESC lines, we performed parallel differentiations batch wise in duplicates, always including WT ESCs and negative controls across seven experiments. At N24, we assayed the differentiation status by FACS analysis (Fig 1E) and extracted RNA for transcriptome analysis.

**The exit machinery is already poised in 2i**

Batch-corrected RNA-seq data (Appendix Fig S3A and B) comprising 14 replicates of WT ES cells identified 3068 differentially expressed genes (DEGs) between 2i and N24 (H0: |log2FC| < log2 (1.5), FDR ≤ 0.05; Fig 1F, Appendix Fig S4A–C and Dataset EV1). Interestingly, most of the 489 genes identified in the haploid screens including the 73 genes (Dataset EV6) selected for KO did not change significantly in expression between 2i and N24 and were not present in the list of DEGs (Fig 1F and Appendix Fig S4C), with only 21% of screen hits showing differential expression at N24 (6% up-, 15% downregulated). This implies that the exit machinery is already embedded in the ground state and ESCs are poised for rapid decommissioning of naïve identity and entry into differentiation (Kalkan & Smith, 2014).

To facilitate interrogation of the KO gene expression datasets, we developed an interactive online tool (GENEES—Genetic Network Explorer for Embryonic Stem Cells—http://shiny.cecad.uni-koeln.de:3838/stemcell_network/).

Using t-Distributed Stochastic Neighbour Embedding (t-SNE), we visualised similarities between KOs based on expression of the DEG

in 2i and at N24 (Fig 1G and H). We observed clustering of members of the same complex or pathway: *Eed-Suz12* (PRC2), *Ptpn11-Raf1-Fgfr1-Etv5* (Fgf/ERK), *Smg5-Smg6* (NMD; nonsense mediated decay), *Mta3-Mbd3* (NuRD) and *Pten-Tsc2* (mTORC1 signalling) (Fig 1G and H). The transcription profiles of the KO ESCs clustered by genotype, but mainly by culture condition (2i and N24) (Appendix Fig S4A). This is consistent with the observation that despite manifesting differentiation delays at N24, all of the KO ESCs ultimately departed from the naïve state during longer differentiation time courses, as measured by loss of Rex1-GFP. Furthermore, even KOs that showed extensive Rex1-GFP downregulation delays at N24 displayed transcription profiles that were globally adjusted towards differentiation. Therefore, the knockout of a single gene is not sufficient to permanently block exit from naïve pluripotency in culture, in accord with the finding that ternary depletion of *Tcf7l1*, *Etv5* and *Rbpj* is required for sustained self-renewal in the absence of 2i or LIF (Kalkan *et al*, 2019).

**The exceptional role of *Csnk1a1* and the involvement of compensatory mechanisms**

At N24, the *Csnk1a1* KO clustered with 2i samples (Appendix Fig S4A), indicating a special behaviour for this mutant. However, at N48 most Rex1-GFP expression was downregulated, illustrating a strong delay but not a block in differentiation (Appendix Fig S5A). siRNA treatment as well as treatment with Epiblastin A, a chemical inhibitor of Csnk1a1 (Ursu *et al*, 2016), delayed the exit from naïve pluripotency without apparently affecting proliferation within the duration of the assay (Appendix Fig S5B–D). Csnk1a1 is a serine threonine kinase and a component of the beta-catenin destruction complex. Although KO of another destruction complex member *Apc*, or of the downstream repressor *Tcf7l1* resulted in the upregulation of similar gene-sets (Appendix Fig S5E), we observed stronger differentiation defects and larger amplitude of gene deregulation in two independently derived *Csnk1a1* mutants. However, these mutants also exhibited markedly impaired cell cycle profiles in N2B27-based media (Appendix Fig S5F). Upon continuous culture (~5 passages), proliferation was restored in *Csnk1a1* KO cells and differentiation potential was regained, suggesting upregulation of compensatory mechanisms and a likely effect of proliferation rate on differentiation kinetics, complicating mechanistic characterisation. A second case of phenotype adaptation was observed in *Pum1* mutants. *Pum1* KOs showed pronounced differentiation delays during early passages (Appendix Fig S5G), as also seen for acute Pum1 depletion by siRNA and in previously generated CRISPR KO ESCs (Leeb *et al*, 2014). However, the phenotype was lost in later passages, and

*Pum1* KO cells showed WT-like Rex1-GFP expression levels at N24 (Fig 2A and Appendix Fig S5G).

### Robust feedback wiring in the naïve TF network

Interestingly, the transcriptome data revealed that exit factors do not, in general, reduce naïve transcription factor (TF) expression in the ground state. However, *Rbpj* KO resulted in moderate but significant increases in *Klf4*, while the aforementioned *Csnk1a1* KO ESCs showed limited upregulation of both *Klf4* and *Tbx3* in 2i (Fig EV1A). *Ctbp2* KO significantly upregulated Nr0b1 in the ground state. Other KOs had no significant effects on factors of the naïve network in 2i. These data are consistent with robust feedback wiring in the naïve TF network (Dunn *et al*, 2014; Niwa, 2018) and neutralisation of most differentiation factors in 2i in culture (Martello & Smith, 2014). In contrast, we observed a more extensive impact on formative markers. Several KO cell lines showed lower baseline expression in 2i of *Otx2, Fgf5, Dnmt3a/b* and *Pou3f1* (Oct6) (Fig EV1B). In-line with recent results, we noted that depletion of several Fgf/ERK components resulted in reduced *Dnmt3a/b, Pou3f1* and *Fgf5* expression in 2i (Kalkan *et al*, 2019). Although Fgf/ERK signalling is effectively inhibited in 2i (Ying *et al*, 2008), our data suggest that either residual pathway activity or potential moonlighting functions of pathway components mediate poised expression of the formative pluripotency programme in 2i.

Clustering based on the expression of ten naïve pluripotency marker genes showed that the downregulation of the naïve pluripotency TFs during formative differentiation is defective across multiple KOs (Fig 2A). Although overall expression of the naïve TFs was highly correlated, *Klf2* appeared to be an exception. *Klf2* downregulation was notably impaired in *Tcf7l1* KO ESCs, whereas it was unaffected by several KOs, including *Jarid2,* despite a comparable extent of deregulation of most other naïve marker genes. This indicates that *Klf2* expression can be uncoupled from the core naïve network. Forced *Klf2* expression stabilises self-renewal (Hall *et al*, 2009; Qiu *et al*, 2015). *Klf2* depletion destabilises mouse ESC identity (Yeo *et al*, 2014) and increases the speed of Rex1-GFP downregulation upon 2i withdrawal (Fig EV1C). The differentiation delay in *Tcf7l1* KO ES cells is partially dependent on *Klf2* (Fig 2B), consistent with direct regulation (Martello *et al*, 2012). In contrast, *Klf2* depletion in *Jarid2* KOs did not restore differentiation timing, indicating that separable gene networks contribute to dismantling naïve pluripotency.

### Effects of exit gene depletion on global gene regulation

We used two measures to gauge differentiation delays in KOs compared with WT cells: first, the average naïve marker change defined throughout this manuscript as the log2 fold change of a set of seven naïve TFs, *Esrrb, Nanog, Tfcp2l1, Tbx3, Prdm14, Klf4* and *Zfp42* at N24 at N24 compared with WT; second, the extent of global transcriptome adjustments between 2i and N24 in each KO compared with changes occurring in WT. Generally, the delayed extinction of the naïve TF network in KOs at N24 was accompanied by reduced gene expression change between 2i and N24 ($R = -0.91$; Fig 2C). *Trim71* KOs formed an exception. They exhibited a relatively strong impact on naïve TF expression levels, while displaying only modest global transcription changes. Thus, the RNA-binding protein Trim71 appears to be focused on the regulation of naïve pluripotency TF genes, although the ensuing exit-delay in *Trim71* KO cells is modest.

A general correlation was observed between the numbers of genes deregulated in 2i and at N24 (Fig 2D). Mutants with the strongest impact on the transcriptome (e.g. *Csnk1a1, Eed, Suz12, Zfp281* and *Smg5*) showed deregulation of several thousand genes both in 2i and at N24. *Fgfr1* and *Tcf7l1* KOs were two exceptions to this correlation with relatively few genes deregulated in 2i but several hundred at N24. We surmise that the effect of these two genes in 2i is largely masked because they are in the same pathways as the inhibitor targets Mek1/2 (PD0325901) and Gsk3 (Chiron 990201) (Ying *et al*, 2008; Martello & Smith, 2014).

Defective differentiation could simply equate with the extent of overall gene deregulation. We therefore mapped the average deregulation of naïve marker genes onto Fig 2D. Most KOs showing the strongest increase in naïve marker gene expression correlated with large-scale gene deregulation (both in 2i and at N24). However, there were several exceptions: for example, KO of *Eed, Suz12, Smg7, Leo1* and *Ctbp2* affected a substantial number of genes, but caused naïve marker deregulation that was weaker than expected assuming a direct correlation between the number of deregulated genes and the differentiation phenotype (Figs 2D and EV1D). Vice versa, KO of *Mbd3, Cdk8* or *Pten* affect naïve marker downregulation, despite a relatively mild impact on the number of DEGs in the respective KOs.

### Transcriptome analysis reveals a genetic interaction between *Jmjd1c* and *Tcf7l1*

The putative histone 3 lysine 9 (H3K9) demethylase *Jmjd1c (Kdm3c)* KO is a case of particular note. No genes were significantly deregulated at N24 (Fig 2D), and a mild global neutralisation of differentiation-induced transcriptome changes was observed (Fig 2C). Neither the role of *Jmjd1c* in pluripotency regulation nor its mode of action is known. We used multiple regression analysis to determine the similarity of the *Jmjd1c* KO to all other KO RNA-seq profiles. At N24, the *Jmjd1c* KO transcriptome was most similar to the *Tcf7l1* KO profiles (Fig EV1E), suggesting a potential functional connection between *Jmjd1c* and *Tcf7l1*. Indeed, Jmjd1c has recently been reported as a high-confidence protein interactor of Tcf7l1 (preprint: Moreira *et al*, 2018). The 2i component Chiron 99021 is a specific inhibitor of Gsk3 and phenocopies deletion of *Tcf7l1*, which is the downstream repressor in ESCs (Wray *et al*, 2011; Shy *et al*, 2013). Accordingly, Chiron 99021 delays the differentiation of WT ESCs, but had little additional effect on *Tcf7l1* KOs, without affecting growth or survival of treated cells (Fig 2E and Appendix Fig S5H). *Jmjd1c*-deficient ESCs did not show a discernible phenotype at N48, in-line with minimal deregulation of the naïve TF network. Addition of Chiron 99021 had a stronger than expected effect on *Jmjd1c* KOs and resulted in a synthetic enhanced delay phenotype (Fig 2E), suggesting a cooperative activity of *Jmjd1c* and the Wnt/Tcf7l1 axis in the exit from naïve pluripotency.

### Relative quantification of differentiation delays *in vitro*

To quantify differentiation delays, we obtained RNA-seq data from a 2h-resolved WT ESC differentiation time course (Fig 3A and Appendix Fig S6A). We then compared the expression patterns of a

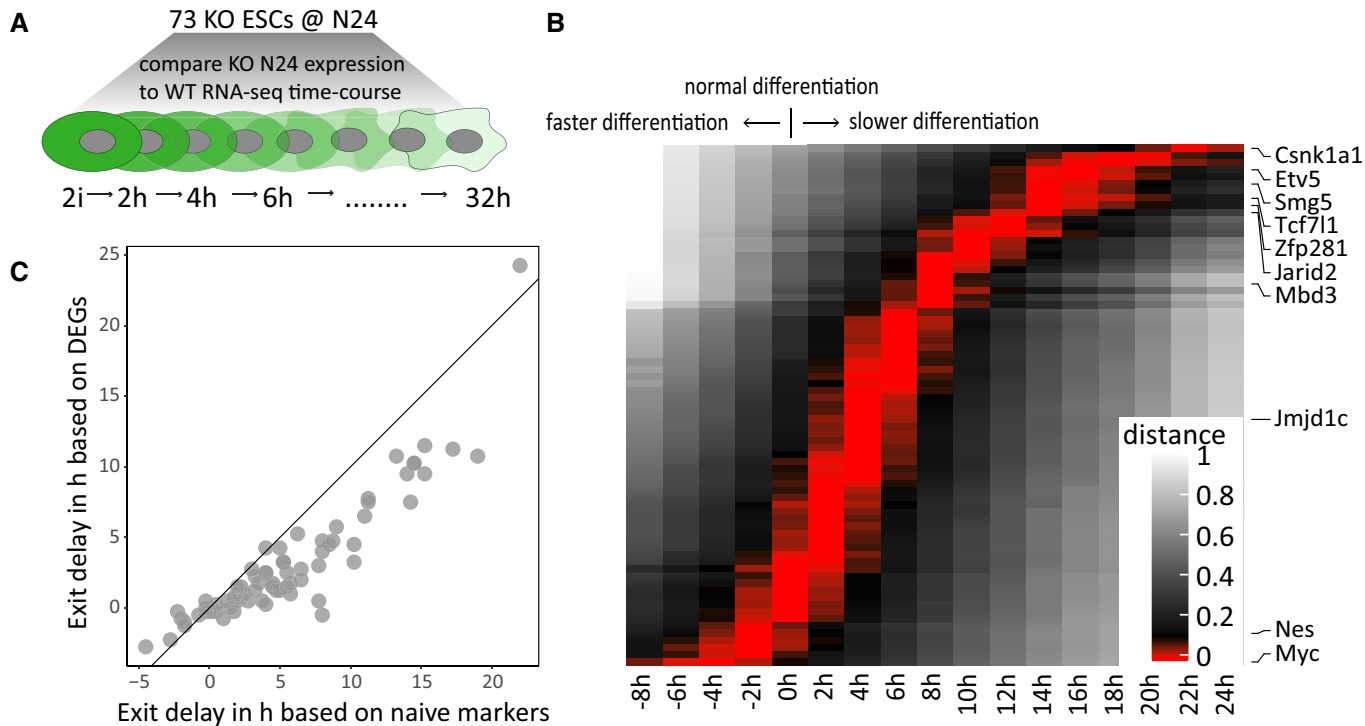

**Figure 3. Relative quantification of differentiation delays *in vitro*.**

A Scheme illustrating comparisons of the 73 KO gene expression profiles at N24 with an *in vitro* differentiation time course of WT ESCs with a 2-h resolution.

B Heatmap showing differentiation delays of the 73 KO lines quantified by average naïve marker gene expression. Red bars indicate the closest correlation to that specific time point. Positive values indicate delayed differentiation. Negative values indicate more rapid differentiation. Each line corresponds to one KO. Selected KOs are indicated. See Dataset EV2 for the full hierarchy of the 73 KOs.

C Plot showing a comparison of the differentiation delays calculated using naïve markers as in (B) to differentiation delays calculated using all 3068 genes differentially expressed in WT differentiation (N24 vs. 2i) (see Appendix Fig 6B).

set of naïve markers in a given KO at N24 with the expression of the same gene-set along the WT 2i to N32 trajectory. This enabled us to "position" each KO along the trajectory and thus quantify the differentiation delay with a precision of about 4h (Fig 3B, Dataset EV2). In a complementary approach, we used all 3068 DEGs between 2i and N24 as an alternative reference set. This yielded similar, but on average slightly less pronounced differentiation delays (Fig 3C and Appendix Fig S6B), supporting the hypothesis that the naïve TF network is regulated in concordance with but partially independently from the rest of the transcriptome during exit from naïve pluripotency.

**The regulatory programme for the *in vivo* pre- to post-implantation epiblast transition is preserved in most mutants**

To explore the extent to which our *in vitro* data captures *in vivo* regulation, we first compared global transcriptomes of WT and KO ESCs with single-cell transcriptome data from the *in vivo* pre- (E4.5) and post-implantation (E5.5 and E6.5) mouse epiblast (Mohammed *et al*, 2017) (Fig 4A and Dataset EV3). As expected (Boroviak *et al*, 2014), WT cells in 2i showed greater similarity (fraction of identity, FOI) to the E4.5 than to the E5.5 epiblast (Gong & Szustakowski, 2013). At N24, in contrast, higher FOI to the E5.5 epiblast was observed. We could not detect a clear difference between FOIs of

cells in 2i or at N24 when compared to the E6.5 epiblast. This underscores that 24 h of 2i withdrawal models the E4.5 to E5.5 epiblast transition (Kalkan *et al*, 2017).

Interestingly, some KO ESCs showed expression features in 2i that indicated increased similarity to the *in vivo* epiblast (Fig 4B). Among those were KOs showing strong *in vitro* differentiation defects, such as *Zfp281, Rbpj, Fgfr1 and Etv5*, but also *Trim71, Fbxw7* and *Smg7*, which showed only modest differentiation delays in culture. PRC2 mutants (*Eed, Suz12* and *Jarid2* KOs) showed an opposing pattern and lost similarity to the E4.5 epiblast in 2i.

At N24 several KOs, such as *Csnk1a1, Zfp281, Fgfr1, Etv5, Rbpj* and *Smg5*, retained strong similarity to the E4.5 epiblast (Fig 4C). Strikingly, *Zfp281* and *Csnk1a1* KO profiles at N24 showed similarity to the E4.5 epiblast on par with WT cells cultured in 2i (Fig 4D). The strong *Zfp281* phenotype is consistent with an overt differentiation delay phenotype (Mayer *et al*, 2020). Overall, there was a good correlation between the *in vitro* differentiation delay and the similarity with pre-implantation epiblast cells, and both the 2i and N24 KO transcriptomes that were more similar to the E4.5 epiblast exhibited stronger exit-delay phenotypes *in vitro* (Figs 4C and D, and EV2A). Together, this indicates that similar transcriptional networks are regulated by similar mechanisms during *in vivo* and *in vitro* transitions to formative pluripotency, with the exception of the *Jarid2* KO, which at N24 showed a similarity to the E4.5 epiblast that was

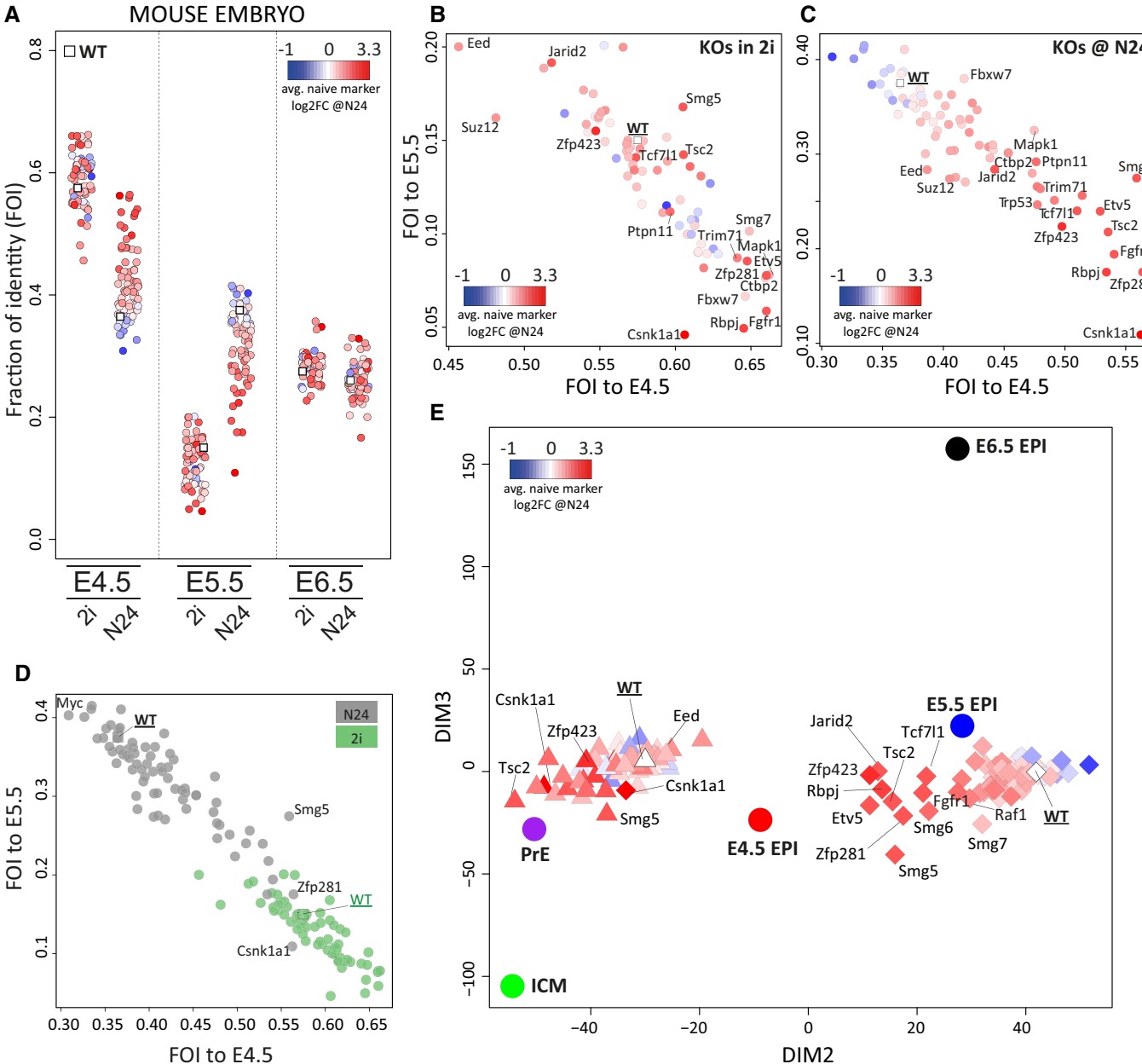

**Figure 4. Comparing KO expression profiles to the *in vivo* pre- to post-implantation epiblast transition.**

A  Fractions of identity (FOI) between 2i and N24 expression data and mouse E4.5, E5.5 and E6.5 epiblast, computed using all expressed genes (log2FPKM > 0) for all KOs. Each dot represents one KO. Average naïve marker log2FC in KOs at N24 is indicated by a colour gradient.

B  FOIs of WT and KO cells in 2i compared with the E4.5 and E5.5 mouse epiblast, computed using all expressed genes (log2FPKM > 0); average naïve marker log2FCs in KOs at N24 are indicated by a colour gradient.

C  FOIs of WT and KO cells at N24 compared with the E4.5 and E5.5 mouse epiblast, computed using all expressed genes (log2FPKM > 0); average naïve marker log2FCs in KOs at N24 are indicated by a colour gradient.

D  FOIs of WT and KO cells in 2i (green) and at N24 (grey) compared with E4.5 and E5.5 mouse epiblast, computed using all expressed genes (log2FPKM > 0).

E  PCA showing all KOs in 2i (triangles) and at N24 (diamonds) and embryo-derived data sets (ICM, primitive endoderm [PrE], E4.5, E5.5 E6.5 epiblast). Colour gradient indicates naïve marker log2FC at N24 of the respective KO.

smaller than expected based on its strong *in vitro* differentiation defect (Figs 4B and C, and EV2A).

We then performed principal component analysis (PCA) for genes variably expressed in embryo development, using averaged values for inner cell mass (ICM), primitive endoderm (PrE) and

E4.5, 5.5 and 6.5 epiblast (Mohammed *et al*, 2017). The PCA separated the 2i and N24 transcriptomes into two clusters with proximity to PrE/E4.5 and E5.5 epiblast, respectively (Fig 4E). Notably, *in vitro* differentiation defects were reflected by closer proximity of mutant N24 profiles to the E4.5 epiblast in the PCA.

We found that overall KOs showing strong *in vitro* differentiation phenotypes also exhibited increased similarity at N24 to the E4.5 epiblast and a concomitant reduction in the similarity to the E5.5 epiblast (Fig EV2B). This indicates a delay along the normal naïve to formative differentiation axis. Such a behaviour was most prominently detectable in KOs for *Zfp281, Etv5, Fgfr1, Rbpj* and *Trim71*. However, KOs for *Csnk1a1, Zfp423* and *Jarid2* deviated from this pattern, indicating that they might have disengaged from the normal embryonic developmental trajectory.

### Correspondence to primate embryogenesis

Despite morphological and timing differences between rodent and primate peri-implantation development, embryos of both orders appear to transit through similar pluripotency states (Smith, 2017; Rossant & Tam, 2018). To examine a potential role for the set of 73 exit factors in primate naïve to formative epiblast transition, we compared transcriptional profiles of our KO series and cells of the macaque *in vivo* naïve and post-implantation epiblast (Nakamura *et al*, 2016). In general, ESCs in 2i were more similar than cells at N24 to the pre-implantation macaque epiblast (Fig EV2C). Correspondingly, N24 cells were closer to the macaque post-implantation epiblast. Levels of similarity of KOs in 2i to the pre-implantation epiblast and at N24 to the post-implantation stage were well correlated between mouse and macaque (Fig EV2D and E). These observations suggest that the overall GRN redeployment and the activity and role of exit factors during the naïve to formative transition are conserved between rodents and primates.

### Identification of an extended naïve pluripotency network

ESC differentiation requires fine-tuned coordination between extinction of the naïve and initiation of the formative transcription networks. To date, only incomplete inventories of the genes that functionally define these two states have been made. These genes cannot be defined based simply on differential expression between 2i and N24, because not all DEGs will be functionally linked to the naïve GRN. To identify those genes that show specific linkage to the core naïve pluripotency network, we trained regression models to predict gene expression changes across all KOs at N24 as a function of naïve marker log fold changes. The 496 genes whose expression

was thereby identified to be tightly associated ($R^2 \geq 0.65$) with one or several of the seven core pluripotency markers (*Nanog, Esrrb, Tbx3, Tfcp2l1, Klf4, Prdm14* and *Zfp42*) were defined as "Naïve-Associated Genes" (NAGs; Dataset EV3). Identification as NAG can be achieved by positive or negative association with naïve TF expression. Thus, we identified 278 NAGs that were downregulated (downNAGs—similar to e.g. *Nanog*) and 218 NAGs that were upregulated (upNAGs—similar to *Pou3f1*, Figs 5A and EV3A) during WT differentiation. Importantly, NAGs largely differ in identity from the top differentially expressed genes in WT differentiation (Fig 5A). Naïve pluripotency TFs not represented in the defining TF set were correctly identified as NAGs, including *Klf5, Nr0b1* and *Nr5a2*. *Klf2* is not one of the NAGs being only weakly associated ($R^2 = 0.49$) with the naïve network, in-line with our earlier observation that *Klf2* expression can be uncoupled from the naïve TF network. Of further note, *Klf2* expression is barely detectable in marmoset or human pre-implantation epiblast cells (Fig EV3B).

The NAG showing the strongest association with the naïve core network is *Pdgfa*, which is relatively highly expressed in ES cells (FPKM ~80). *Pdgfa* has no known role in ESC self-renewal, but functions in segregation of the primitive endoderm (Artus *et al*, 2013). To examine whether the link between *Pdgfa* and the naïve transcription factor network is maintained *in vivo*, we utilised GRAPPA, a tool to visualise single-cell expression data from pre-implantation development (Boroviak *et al*, 2018). Indeed, *Pdgfa* is uniformly expressed in the E4.5 epiblast (Fig EV3C). Its cognate receptor *Pdgfra*, in contrast, is neither expressed in ES cells nor the naïve epiblast (FPKM in WT ESCs < 0.5), but specifically marks the neighbouring primitive endoderm at E4.5 (Plusa *et al*, 2008).

We surveyed the expression of NAGs at the transition from naïve to post-implantation pluripotency in the single-cell RNA-seq data sets from E4.5, E5.5 and E6.5 epiblast cells (Mohammed *et al*, 2017). We detected a clear enrichment of NAGs within genes that separate the pre- from the post-implantation epiblast in the differentiation state resolving dimension of a principal component analysis (PCA) (Figs 5B and EV3D), highlighting that NAGs are indicators for the naïve epiblast state also *in vivo*. Strikingly, NAGs showed a strongly correlated fold change behaviour *in vitro* and *in vivo* during the epiblast transition from E4.5 to E5.5 (Fig 5C, E and F). Underscoring the special role of NAGs, this effect was not observed for a matched set of top differentially expressed genes in ESC

---

**Figure 5.   Identification of an extended naïve pluripotency network.**

A   Venn diagrams showing the overlap between upNAGs and top-upregulated genes (left) and downNAGs and top-downregulated genes (right).

B   The x-axis displays all 3[rd] dimension (Dim3) genes, separating mouse naïve from post-implantation epiblast (the inset shows the full PCA plot, see also Fig EV3D) in order of extent of contribution to Dim3. The contribution of the 496 NAGs (218 upNAGs, 278 downNAGs; orange line) to Dim3 and a size-matched set of top differentially expressed genes (218 top-upregulated, 278 top-downregulated based on log2FC between 2i and N24; green) are plotted in a cumulative manner.

C   Log2FC in mouse ESC differentiation (N24 versus 2i) compared with log2FC in mouse *in vivo* transition from pre- to post-implantation epiblast (E5.5 versus E4.5). Selected gene groups are plotted from left to right: i) NAGs, ii) all 3068 differentially expressed genes between 2i and N24 and iii) top differentially expressed genes in mESC differentiation (218 top-upregulated, 278 top-downregulated). Rho values indicate the level of correlation between *in vivo* and *in vitro* differentiation of given gene groups. Alpha indicates the angle between the x-axes and the orthogonal regression. Genes with |log2FC| > 0.1 in both the x- and y-axes are highlighted in orange and blue, respectively.

D   Log2FC in mouse ESC differentiation (N24 versus 2i) compared with log2FC of orthologues in macaque *in vivo* transition from pre- to post-implantation epiblast. Gene groups from left to right are identical to (C).

E   Frequency of co-regulation of NAGs or the matching set of top differentially expressed genes and differentially expressed genes in mouse and macaque *in vivo* pre- to post-implantation epiblast differentiation. Co-regulation was defined as showing a |log2FC| > 0.1 in both conditions. Two-tailed chi-square test (CI 95%) was used to compute significance levels between the expected and the observed number of modulated genes.

F   Heatmap derived from one-way hierarchical clustering of log2FCs of NAGs (left) and top differentially expressed genes (right) in the *in vitro* exit from naïve pluripotency model (N24 versus 2i), mouse pre- to post-implantation transition and macaque pre- to post-implantation transition.

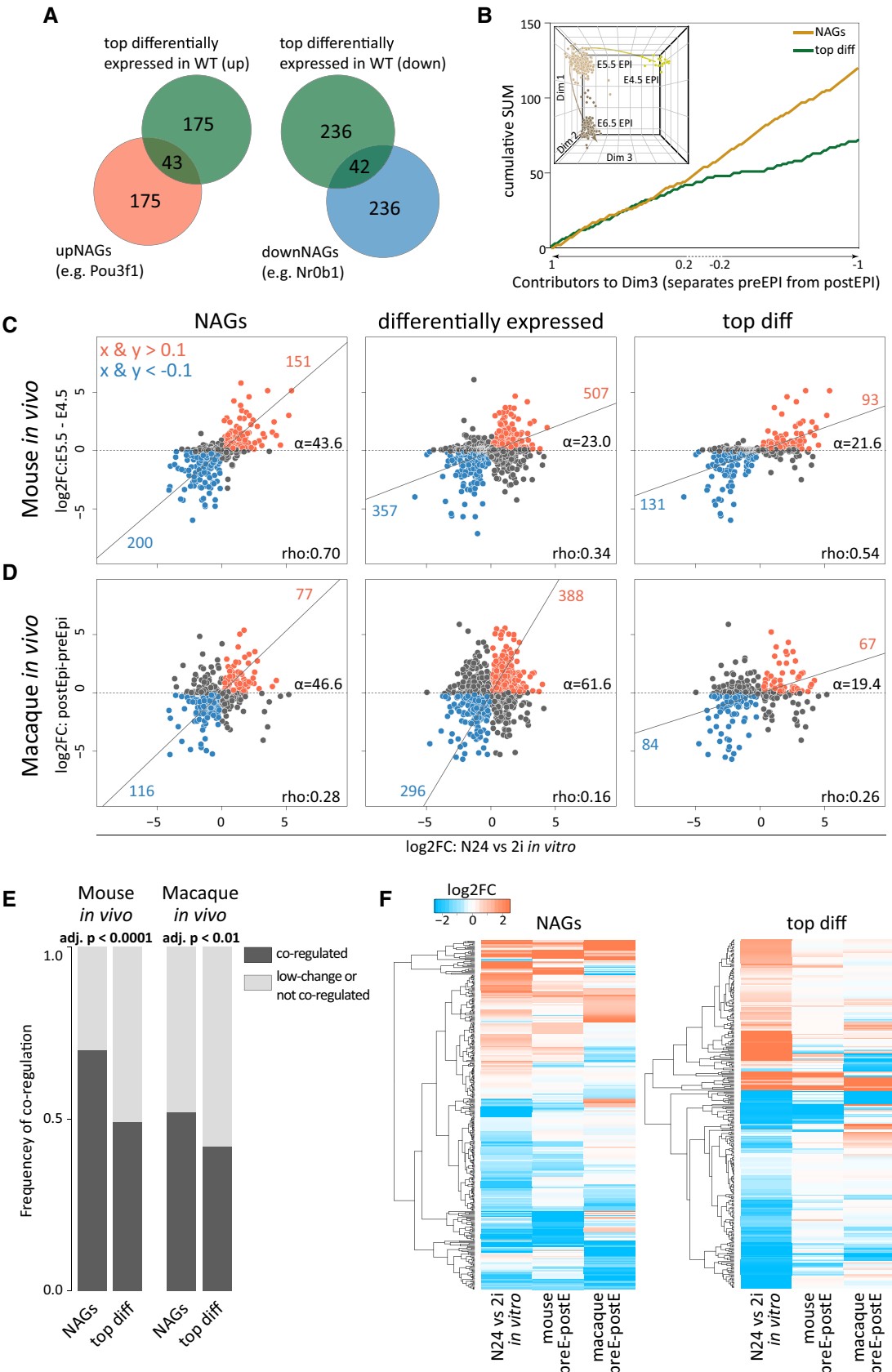

**Figure 5.**

differentiation *in vitro*. Furthermore, NAG orthologues showed significantly co-regulated expression dynamics during macaque pre- to post-implantation epiblast differentiation *in vivo* (Fig 5D–F), which further underscores their relevance. We thus propose that NAGs constitute an integral component of the naïve transcriptional network with potential functional relevance *in vivo* across mammalian species.

**Deregulation of signalling cascades is a hallmark of differentiation delay**

We next analysed the extent of deregulation of five key signalling pathways known to be active in pluripotent cells: LIF/Stat3, mTORC1, Wnt/β-catenin, Fgf/ERK and Notch (Smith *et al*, 1988; Lowell *et al*, 2006; Watanabe *et al*, 2006; Kunath *et al*, 2007; Ying *et al*, 2008; Molotkov *et al*, 2017). Changes in the activity of a signalling pathway are not necessarily reflected in expression changes of pathway member transcripts. Thus, in order to quantify pathway activities, we employed "expression footprints". We identified pathway-specific marker gene sets, each containing 50 genes, reporting pathway activity changes. These marker sets were determined using N24-derived KO transcription profiles of ESC lines deficient for key signalling pathway components. Thereby, we defined specific mTORC1 (affected by *Tsc2 KO*), Wnt/β-catenin signalling (*Tcf7l1 KO*), Fgf/ERK (*Fgfr1* and *Ptpn11 KOs*) and Notch (*Rbpj KO*) pathway targets (Appendix Fig S7A, Dataset EV4). Overlaps with available Tcf7l1 ChIP profiles (Martello *et al*, 2012) supported the reliability of this approach. For LIF signalling, we compared RC9 ESCs grown in 2i in the presence and absence of LIF for 24 h. This resulted in a list of LIF-sensitive genes including known targets such as *Socs3, Gbx2, Junb, Tfcp2l1, Klf4* and *Klf5* (Martello *et al*, 2012; Ohtsuka *et al*, 2015) (Dataset EV4). In summary, the expression footprints present non-overlapping sets of marker genes whose expression state is indicative of the activity of the respective signalling pathway.

Using the expression footprints, we then asked whether we could detect a preferential deregulation of one or more of these signalling footprints in specific KOs either in 2i (for LIF) or at the N24 time point (for all other pathway profiles) (Fig 6A, Dataset EV4). Surprisingly, we detected mis-regulated LIF target genes in several KO ESC lines cultured in 2i in the absence of LIF (Fig 6A). Activation of such

a "LIF-like" profile in 2i was closely correlated with the extent of differentiation delay observed at N24 (Appendix Fig S7B). Notably, the presence of LIF together with 2i before induction of differentiation slows down naïve state exit (Appendix Fig S7C), suggesting a causative role of increased LIF likeness for the observed differentiation delays (Dunn *et al*, 2014; Mulas *et al*, 2017). *Tsc2, Ptpn11, Raf1, Mapk1* and *Trim71* KO ESCs showed the greatest similarity to the LIF profile in 2i; in contrast, *Tcf7l1* KO ESCs, despite showing a pronounced differentiation defect, lack the "LIF footprint" (Appendix Fig S7B). *Tsc2* depletion resulted in the strongest activation of a LIF footprint of all KOs, suggesting that a component of the LIF response is mediated through Akt (Appendix Fig S7B) (Watanabe *et al*, 2006; Niwa *et al*, 2009). Addition of a JAK inhibitor to several KOs showing a LIF-like profile resulted in no or only minor amelioration of the differentiation defect. This suggests that these KOs do not directly activate Jak/Stat signalling, but that a LIF-like expression profile reflects a consolidated naïve network that is resilient to dismantling.

*Tsc2* deficiency leads to constitutive activity of the mTORC1 pathway, which has previously been associated with an exit from naïve pluripotency phenotype (Villegas *et al*, 2019). KOs including *Pten, Csnk1a1, Zfp281, Eed* and *Suz12* exhibited deregulated Tsc2 responsive genes. Surprisingly, deletion of *Etv5*, a core downstream TF in the Fgf/ERK pathway, also increased the set of Tsc2-responsive genes. Genes downstream of the Notch pathway were most strongly affected in a group of mutants deficient for mRNA homeostasis (*Smg5* and *Smg6*) and transcriptional regulators, including *Zfp423, Jarid2, Mbd3* and *L3mbtl3*. The latter suggests an interaction between the Notch pathway (*Rbpj*) with the Polycomb and NuRD complexes to modulate network rewiring during the exit from naïve pluripotency. Co-operativity between Rbpj and the Polycomb-associated protein L3mbtl3 has been reported in *Drosophila* and *C. elegans* (Xu *et al*, 2017). As expected, *Raf1, Fgfr2, Mapk1* and *Etv5* KO ESCs showed an Fgf-signalling footprint at N24. *Mbd3, Mta3, Nsd1* and *Ctbp2* KOs showed Fgf/ERK target deregulation similar to reference KOs, indicating an involvement of chromatin regulators in ERK target gene control (Fig 6A). The KOs for the beta-catenin destruction complex member *Apc* and the previously discussed *Jmjd1c* showed the expected similarities with the *Tcf7l1* profile. Interestingly, several KOs with strong phenotypes showed footprints similar to *Tcf7l1* KOs at N24. This is in-line with evidence that β-

**Figure 6. Signalling pathways and gene expression modules regulated by multiple KO genes.**

A  Relationship between the KO expression profiles at N24 (or in 2i for the LIF pathway) to the expression footprints of the five signalling pathways, as indicated (see Appendix Fig S7A for details on derivation of pathway footprint genes). Relative pathway activity is set to −1 (red, indicating relative inactivity of pathway in pathway-defining KO) for the Fgf/ERK (defined by *Fgfr1* & *Ptpn11* KOs), Wnt (*Tcf7l1* KO) and Notch (*Rbpj* KO) pathways and + 1 (green, indicating relative activity in pathway-defining KO or upon LIF supplementation) for mTORC1 (*Tsc2* KO) and LIF pathways. Pathway activity is indicated by a colour gradient (green for activity, red for inactivity). All tiles with an absolute pathway activity smaller l0.3l are greyed out. The size of the tiles corresponds to the Spearman correlation of pathway footprints between each KO and the corresponding representative KOs. The colour gradient on the sidebar at the left indicates average log2FC of naïve marker gene expression for the corresponding KO at N24 (strength of differentiation phenotype).

B  Multiple regression coefficients of pathway activity profiles across all 73 KOs were used to predict the differentiation delay phenotype (defined by average naïve marker log2FC at N24). Error bars indicate the standard error of the coefficients.

C  The absolute correlation value (Pearson's) between average expression change of cluster genes and expression change of naïve marker genes is plotted against the enrichment of NAGs in all clusters. Constitutive clusters are indicated by triangles, induced clusters are shown as circles. Adjusted *P*-values (Fisher's exact test) are indicated as colour gradient for upNAGs (e.g. formative genes; shades of red) and downNAGs (e.g. naïve TFs; shades of blue).

D  GO enrichment analysis for constitutive KO clusters. Dot-size scales with log2 enrichment. Multiple testing corrected *P*-values (Fisher's exact test) are colour coded in shades of red.

E  As for (D) except GO enrichment analysis is for the N24-induced clusters.

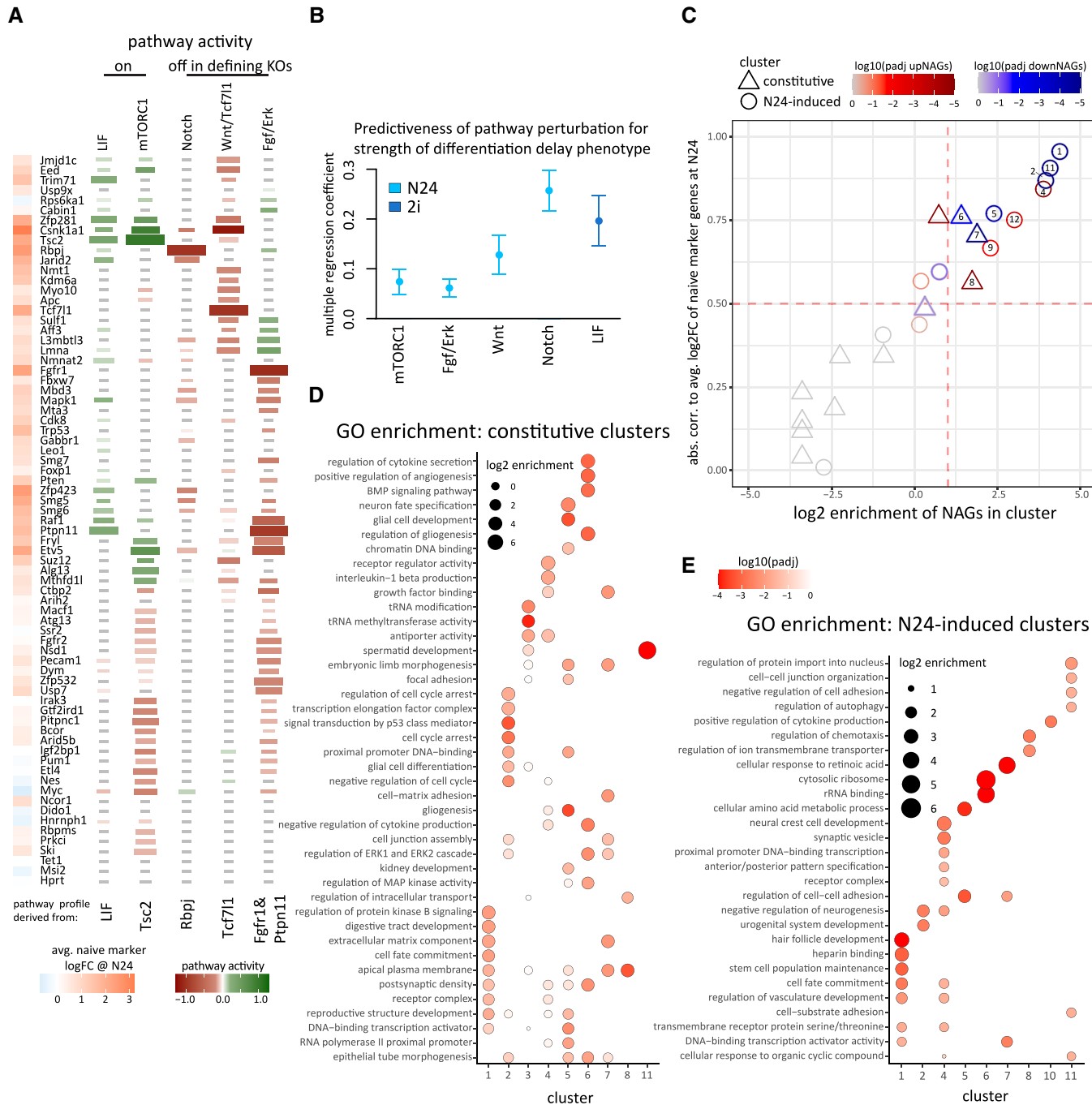

**Figure 6.**

catenin constitutes or regulates a major differentiation switch during the exit from naïve pluripotency and is influenced by multiple exit KO genes (Wray *et al*, 2011; Hoffman *et al*, 2013; Neagu *et al*, 2020). *Rbpj* and *Tcf7l1* KOs did not show a correlation with each other or with *Tsc2* KO, consistent with independent but cooperative mechanisms (Kalkan *et al*, 2019).

We then asked to what extent the alteration of any of the five signalling pathways was predictive for the strength of the differentiation defect. We found that whereas aberrant Fgf/ERK or mTORC1 activity were unreliable predictors, deregulation of Notch/Rbpj or

Wnt/Tcf7l1 at N24, or deregulation of LIF target genes in 2i correlated with differentiation delay (Fig 6B). Together these data indicate that many KOs with a delay phenotype have altered activity of at least one of the five key signalling pathways. Strikingly, however, there was no single signalling pathway perturbed downstream of all KOs. We observed that KOs showing multiple pathway footprints are rare, but most KOs showing a delayed naïve marker gene downregulation at N24 deregulate at least one specific pathway. *Trp53*, *Ncor1* and *Usp9x* constitute exceptions, which show appreciable differentiation defects without clear deregulating of any of the tested

signalling cascades. We conclude that signalling pathways are harnessed in an independent fashion and that most KO genes contribute directly or indirectly to at least one pathway activity that promotes the exit from naïve pluripotency.

### Identification of gene-networks downstream of multiple KO genes

To explore common regulatory targets of multiple KO factors, we classified deregulated genes into two groups: (i) deregulated by a KO in both 2i and at N24, termed "constitutive KO response"; (ii) not, or only weakly, deregulated by a KO in 2i, but significantly deregulated at N24, termed "N24-induced KO response". Genes in both groups are affected by knockouts, but genes in the second group deviate from wild-type expression only under conditions of differentiation. We identified all genes that showed either type of behaviour in at least one KO (Appendix Fig S7D; Dataset EV5). Genes deregulated exclusively in PRC2 core-components (*Eed* or *Suz12* KOs) or *Csnk1a1* KOs were excluded from further analysis, because the deregulation of thousands of genes in these KOs had a disproportionate impact on the resulting gene lists. In total, the constitutive and the N24-induced KO response groups contained 1,886 and 716 genes, respectively. Genes belonging to the N24-induced KO response group were strongly enriched for factors that are dynamically regulated during normal differentiation, whereas genes of the constitutive response group were not (Appendix Fig S7E). To characterise the constitutive KO response genes, we grouped them based on their log2 fold changes between KO and WT at N24. This resulted in twelve clusters, ranging in size from 5 to 496 genes (Fig EV4A and Dataset EV5). Analysis of the N24-induced response genes yielded 12 clusters ranging from 17 to 107 genes (Fig EV4B and Dataset EV5).

Gene expression patterns in 9 out of 12 N24-induced clusters showed a strong correlation with the phenotypic strength across all KOs, indicating that they contain genes relevant for the naïve to formative transition (Fig 6C). All phenotype-correlated N24-induced clusters showed enrichment of up (N24-induced clusters i4, i9, i12)- or downNAGs (N24-induced clusters i1, i2, i5, i11).

Induced cluster 1 was enriched for genes involved in cell fate commitment and stem cell population maintenance and contained genes that are associated with signalling and transcriptional activation (Fig 6E, Dataset EV5, Appendix Figs S8–S20). Naïve marker genes were detected in N24-induced cluster 1 (i1), with the notable exception of *Klf2*. *Klf2* was part of cluster i5, within which more than half the genes were bound by Tcf7l1 (Martello *et al*, 2012) (Fig EV4D). This provides further support for the notion that Wnt/Tcf7l1 controls *Klf2* expression, independently of the rest of the naïve network.

Cluster i4 was enriched for functional terms overlapping with cluster i1, indicating that these two clusters contain pivotal cell fate switch genes with opposing expression patterns during differentiation. Consistently, formative factors were detected in cluster i4, which was further enriched for several GO terms relating to neuroectodermal differentiation, indicating that i4 contains genes of the formative programme and the ectodermal "default" programme of ESC differentiation (Ying *et al*, 2003) (Fig 6E, Dataset EV5, Appendix Figs S8–S20). Average expression of genes in both clusters (downregulation in cluster i1, upregulation in cluster i4) showed

strong correlations with the extent of differentiation defects across all KOs (Fig 6C).

Interestingly, cluster i11 with the second highest NAG enrichment and phenotype correlation is enriched for terms pertaining to cell adhesion, indicating the importance for cell–cell interaction for proper regulation of the exit from naïve pluripotency. Further enrichments for cell adhesion in clusters i5 and i7 underscore the importance of controlling cell adhesion properties for maintaining proper differentiation behaviour. Cluster i8, although neither strongly correlated with the exit defect nor enriched for NAGs, contained pivotal cell fate switch genes such as *Tcf7l1*, *Fgf4* and *Tfe3* (Betschinger *et al*, 2013).

A correlation with the differentiation defect was less pronounced in constitutive clusters and only detectable in constitutive clusters c6, c7 and c8, with an overall weaker effect than observed for N24-induced clusters. Consistently, only five out of twelve constitutive clusters showed an enrichment for NAGs, including phenotype associated c5, c6 (downNAGs) and c8 (upNAGs). Constitutive cluster 6 genes were enriched for Bmp signalling and Fgf/ERK signalling-related terms and constitutive cluster 7 for Fgf/ERK signalling and extracellular matrix related terms (Fig 6D and Dataset EV5). Overall, in contrast to the N24-induced KO response, constitutive clusters showed a less cluster-specific enrichment for GO terms (Fig 6D and E). In summary, constitutive and N24-induced clusters define crucial genetic modules that are co-regulated by multiple exit factors.

### Wiring between regulatory pathways and gene expression clusters

To identify regulatory pathways potentially controlling the gene expression clusters from above, we measured signalling pathway activity using the pathway-specific genes defined before across the 2h-resolved 32h long WT differentiation time course (Fig 7A). mTORC1 and LIF signalling were downregulated in parallel, with slightly slower kinetics for LIF signalling. Notably, the *Tcf7l1*-axis of Wnt signalling showed an early activation whereas Fgf/ERK signalling appeared to initiate strong activation only just before the 20h time point. This is in-line with a reported sequential requirement of first Wnt and then Fgf signalling to transit cells to a formative state (Neagu *et al*, 2020). We then utilised pathway activities and cluster-specific expression levels across all KOs to build a model predicting the impact of pathway activity on cluster-gene expression (Figs 7B and EV5AB). We only considered those pathways-to-cluster connections in our model where activity changes of the regulatory pathways preceded the expression changes of their putative target clusters over the 32 h WT differentiation time course (Fig EV5C and D). In our model, the Notch and mTORC1 pathways showed high connectivity to N24-induced clusters (7 out of 12 clusters, each), whereas Wnt/β-catenin signalling was specifically correlated with clusters containing high levels of downNAGs (cluster i1, i5 and i11) and the only constitutive cluster containing an appreciable number of (down)NAGs (const c6), indicating a major role for *Tcf7l1* in downregulation of the naïve GRN.

Together, this shows that N24-induced clusters which contain genes dynamically regulated during the exit from naïve pluripotency are largely under the control of five key signalling pathways gating differentiation in a cooperative manner by co-regulation of the majority of differentiation-specific gene modules.

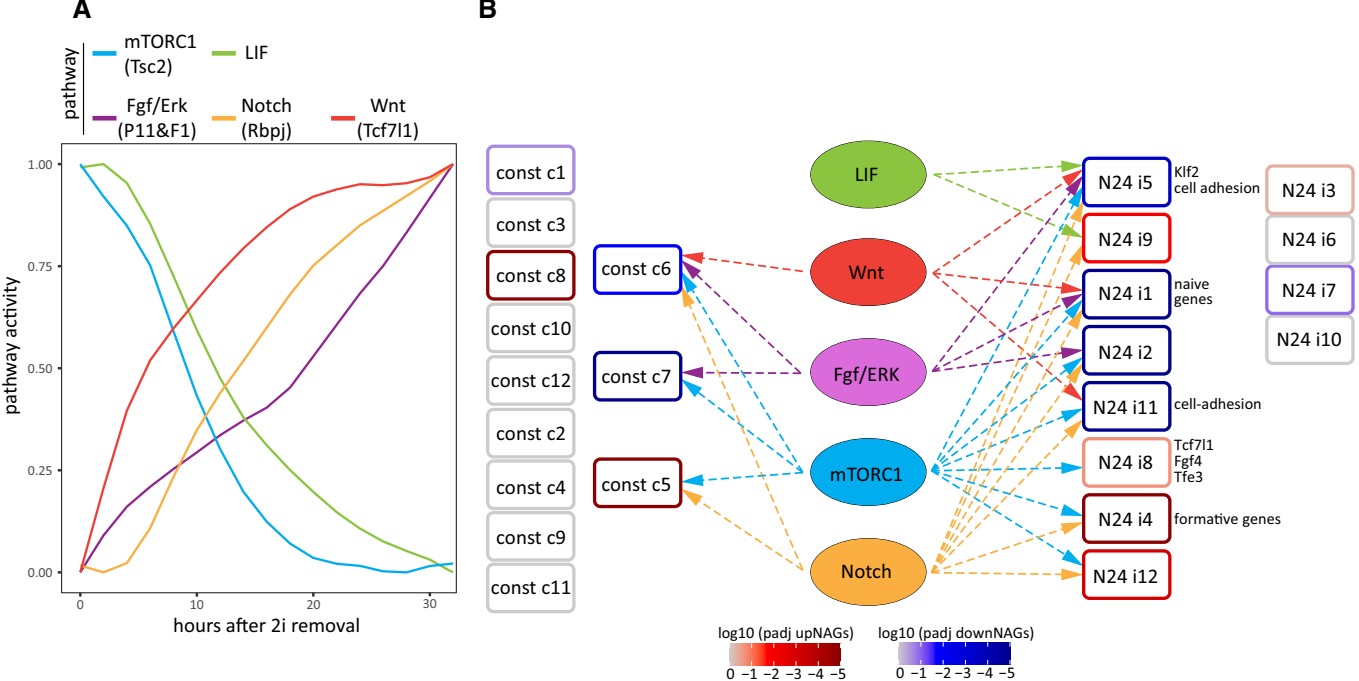

**Figure 7.  Wiring between regulatory pathways and gene expression clusters.**

A  Pathway activities (average expression of the 50 specific pathway-defining genes) across a WT differentiation time course for the mTORC1, LIF, Fgf/Erk, Notch and Wnt pathways. Pathway activity is defined as the average log2FC of pathway-defining genes relative to the mean expression throughout the time course.

B  Visualisation of connections between pathways (central ellipses) and gene clusters (rectangles), based on multiple regression models. The colour gradient of the clusters indicates adjusted *P*-values of enrichment for up- or downNAGs. Connecting arrows link pathways to clusters that fit time course validated regression models (see Fig EV5B–D). Connecting arrows are coloured by source pathway. Unconnected clusters are listed separately (constitutive, left; N24-induced, right).

## Discussion

Ordered progression through pluripotency requires the shutdown of the naïve TF network and concomitant large-scale rewiring to establish the formative GRN (Buecker *et al*, 2014; Kalkan & Smith, 2014; Kalkan *et al*, 2017). Following Harald Weintraub's pioneering work with muscle differentiation and Thomas Graf's work on haemato-poiesis, the ability of certain transcription factors to change cell identity without a developmental context gained central prominence with somatic cell reprogramming to pluripotency (Lassar *et al*, 1986; Xie *et al*, 2004; Takahashi & Yamanaka, 2006). However, most of these studies involved forced expression of selected transcription factors to achieve synthetic cell state transitions *in vitro*. Here, we utilised the remarkable properties of ESCs to recapitulate the pivotal embryonic transition from naïve to formative pluripotency. We show that this developmental cell state transition is not based on the action of few master transcription factors. Although crucial for defining lineage trajectories, transcription factors do not initiate cell fate decisions. Instead, the robust stability of naïve pluripotent stem cells is sustained and dismantled by multiple crosstalking components and by at least four diverse signalling inputs.

Our dissection of an authentic cell state transition indicates that the exit from naïve pluripotency is largely guided along a trajectory constrained by signalling pathways and funnelled through a handful of genetic modules. These adaptations of cellular networks establish the opportunity for TFs to initiate and consolidate cell identity. In the specific case of the exit from naïve pluripotency, expression of Oct4 and Sox2, which are key TFs for both the naïve and formative states, supports the maintenance of both the naïve and formative regulatory networks to ensure lineage fidelity during differentiation. Transcription factor circuits are crucial to maintain cell identity and to ensure proper lineage identity, but our data suggest that lineage-specific TFs do not stand at the top of cell fate decision hierarchies.

Several signalling cascades have been implicated in ESC self-renewal and naïve pluripotency exit, including pathways controlled by LIF, Akt/mTORC1, Wnt, Fgf/ERK and Notch. We derived transcription footprints of these pathways and scrutinised the extent to which they were deregulated by the different KOs to find that virtually all KOs showing a strong differentiation defect also showed deregulation of at least one of these five pathways. Notably, however, we observed segregation in terms of pathway deregulation. Thus, KOs that induce, e.g. a *Tcf7l1*-like profile were less likely to show a signature of another pathway. We also confirm a previously reported sequential activity of Wnt and Fgf/ERK signalling during the naïve to formative transition (Neagu *et al*, 2020). Our data suggest that the exit from naïve pluripotency involves co-ordinated activity changes of independent but functionally partially redundant pathways under the control of separable genetic networks. The observation that single gene depletion is not sufficient to prevent exit from the naïve state is consistent with this interpretation. Our analyses exposed discrete pathway and GRN features that mediate timely and robust mammalian cell state

transitions. The finding that multiple sub-networks with distinct expression patterns can be identified indicates the existence of several independent but cooperative molecular programmes that control differentiation.

By examining co-regulation with the core naïve network across all KOs at N24, we identified the NAGs. NAGs obey strikingly similar expression dynamics *in vitro* and *in vivo*. Moreover, in the macaque pre- to post-implantation transition, orthologous NAGs show similar behaviour as during mESC exit from naïve pluripotency. Therefore, this cohort of genes constitutes a layer of the pluripotency network that is tightly linked to, and likely acts in conjunction with, the core naïve TF network across mammalian species (Dunn *et al*, 2014). We suggest that collective modulation of NAG expression will propel naïve cells into formative differentiation and, conversely, that collective NAG deregulation will delay proper differentiation in exit factor KOs. The control of NAG expression appears to be interwoven with the signalling pathways involved in controlling the naïve to formative transition. The N24-induced clusters enriched for NAGs showed high levels of deregulation in KOs exhibiting a strong differentiation delay and an overall high connectivity to all tested pathways. The functional significance of NAGs likely extends beyond self-renewal and naïve identity. Indeed, most NAGs are not transcription factors. The growth factor Pdgfa is a case in point. *Pdgfa* is the most strongly associated NAG, but has no known activity on ESCs, which do not express the cognate receptor, Pdgfra. Within the ICM, Pdgfa expression is restricted to the naïve epiblast, whereas Pdgfra is present exclusively in the primitive endoderm. This reciprocal expression pattern is consistent with the known paracrine action of Pdgfa to promote primitive endoderm segregation (Artus *et al*, 2013). Thus, conserved linkage of NAGs to the naïve TF network may mediate paracrine communication to other lineages within the blastocyst as well as consolidate naïve epiblast fate.

Whereas most naïve TFs were tightly co-regulated across the 73 KOs, *Klf2* expression was uncoupled from other members of the core network. Interestingly, while *Klf2* is highly expressed in mouse ESCs and naïve epiblast, it is very lowly expressed in primate blastocysts, and also in the porcine pluripotent compartment (Ramos-Ibeas *et al*, 2019). *Klf2* may therefore be a rodent-specific addition to the naïve TF network. A further significant species difference is the very low expression of *TCF7L1* in primate naïve cells, which underlies the differential responsiveness of mouse and human naïve cells to GSK3 or Wnt pathway inhibition (Bredenkamp *et al*, 2019; Rostovskaya *et al*, 2019). *Klf2* is a genomic target of Tcf7l1 (Martello *et al*, 2012). Together this suggests a rodent-specific linkage between *Klf2* and *Tcf7l1* as regulatory module to sustain and extinguish the naïve to formative transition, respectively.

For most KOs, the degree of failure in naïve TF downregulation and the neutralisation of global differentiation-related transcription changes are comparable. However, the relatively weak *Trim71* phenotype despite specific upregulation of some naïve TF genes suggests that upregulation solely of core naïve factors must reach a certain threshold to stably maintain naïve pluripotency, consistent with negative feedback constraints within the network (Chambers & Tomlinson, 2009; Dunn *et al*, 2014; Niwa, 2018).

The remarkable advantages of ESCs as an experimental venue include the utilisation of well-defined culture conditions to recapitulate key features of *in utero* developmental progression (Nichols &

Smith, 2012; Boroviak & Nichols, 2014; Kalkan *et al*, 2017; Smith, 2017). Our high-resolution transcriptome analysis confirms that differentiation over 24 h *in vitro* mimics the peri-implantation transition from naïve to formative epiblast *in utero*. These data provide rich resources for further studies. In particular, many strong differentiation delay KO transcriptome profiles retained transcriptome resemblance to the pre-implantation epiblast. This relationship held true in 2i and at N24 and indicates that the KO of these genes perturb the cell state transition machinery operative *in vivo*. In general, we surmise that stalling the activity of the exit machinery is critical to *in vitro* capture and self-renewal of naïve stem cells, consistent with empirical requirements for signal inhibition in both mouse and human (Ying *et al*, 2008; Bredenkamp *et al*, 2019). Interestingly, deficiency for some differentiation regulators increases similarity of 2i profiles to the *in vivo* pre-implantation epiblast. Therefore, we propose that some of the mechanisms driving exit from naïve pluripotency contribute to the lack of complete identity between *in vivo* E4.5 epiblast and ES cells in 2i. This is in-line with ESCs representing an engineered interruption in developmental progression and suggests that state-of-the-art self-renewal culture conditions may not yet capture the naïve state perfectly (Smith, 2017).

Our analyses provide a comprehensive inventory of genetic factors and regulatory networks governing the major cell fate transition from pre- to post-implantation epiblast. We present evidence that mammalian *in vivo* development employs similar regulatory networks operating with similar mechanisms (Boroviak & Nichols, 2017; Rossant & Tam, 2017). These programmed cell state transitions in development do not rely upon instructions delivered by one or two master transcription factors. Rather, a cloud of activity involving multiple co-ordinated and partially redundant inputs serves to destabilise an existing, multiply stabilised GRN. Thereby, a consolidated cell state can be rapidly dismantled by a combination of signals, triggering transition to the next developmental waystation.

# Materials and Methods

### Cloning

Guide RNAs (gRNAs) were designed using the computational pipeline described below, or by http://crisprscan.org. To clone the gRNAs for ESC KO of the 73 selected genes, we used RecET recombineering to insert the guide RNA protospacer encoding sequence between the U6 promoter and the universal sgRNA scaffold in pBR322-U6-cm-ccdB-sgRNA-amp, as described previously (Baker *et al*, 2016). In brief, the plasmid was linearised at the point of insertion by BstZ17I digestion, purified using the Invitrogen Charge Switch PCR Purification Kit and dissolved in water; 80-mer oligonucleotides containing the protospacer sequence, flanked by 30 bp of homology to the target plasmid were dissolved in water. *E. coli* GB05 transformed with the pSC101-Prha-ETgA-tet plasmid (Baker *et al*, 2016) were cultured to OD600 ~0.5 in deep-well 96-well plates and then induced for 1 h for expression of the ETgA operon with L-rhamnose. Electroporation with 200 ng (0.1 pmol) of the linearised vector and 50 pmol of oligonucleotide, using BTX 96-well electroporation plates (MOS96, 2 mm gap,) and the BTX

ECM630 electroporator with HT-200 adapter, was described previously (Sarov et al, 2012). After 1 h of recovery time, 100 µl (1/10th) of the culture was transferred to a new plate with 900 µl of LB media plus 100 µg/ml ampicillin. After overnight selection, the saturated cultures were plated for single colonies on L-agar + 100 micrograms/ml ampicillin. A single colony for each construct was picked and the clones were grown under selection in 96-well plates, plasmid DNA was isolated, and Sanger sequenced from both directions using the M13F and M13R primers (flanking the U6-protospacer-sgRNA scaffold cassette, Dataset EV6). Sequence analysis confirmed the correct engineering of the plasmids. Alternatively, annealed oligonucleotides (Dataset EV6) were cloned into a BsaI site of a gRNA expression vector (Addgene Plasmid number 41824) (Mali et al, 2013) and correct insertion was determined by Sanger sequencing with the SP6 primer (Dataset EV6). For generating rescue cell lines, indicated coding sequences were cloned into a pCAG-3xFLAG-empty-pgk-hph vector (Betschinger et al, 2013) after PCR amplification. Correct insertion was detected by restriction digest and Sanger sequencing with the 3xFlag_seq primer (Dataset EV6).

## Cell culture

Diploid (biparental) ES cells were routinely cultured in gelatin-coated 25 cm$^2$ flasks in DMEM supplemented with 15% FCS (batch tested, Biowest), 1× Penicillin-Streptomycin (Sigma), 0.1 mM NEEA, 1 mM sodium pyruvate, 1 mM L-glutamine, 0.05 mM 2-mercaptoethanol (Gibco), 10 ng/ml LIF (batch tested, in-house), 1.5 µM CHIRON and 0.5 µM PD0325901 (termed ESDMEM-2i) on gelatin-coated plates. The basal medium during all differentiation assays and for haploid ES cell culture (N2B27) consisted of a 1:1 mixture of DMEM/F12 and Neurobasal medium supplemented with 1× B27 (Gibco), 0.5× N2 (homemade), 0.1 mM NEAA, 1 mM L-glutamine, 1× Penicillin-Streptomycin and 0.05 mM 2-mercaptoethanol. Haploid ES cells were routinely cultured in N2B27 supplemented with 3 µM CHIR99201, 1 µM PD0325901 and 10ng/ml LIF and regularly sorted based on FSC/SSC parameters by FACS (Leeb et al, 2015). Haploid Rex1::GFPd2-IRES-BSD (hRex1-GFPd2) (Leeb et al, 2012) and haploid Rex1::mKO2-IRES-BSD/pOct4-GFP-IRES-Puro ESCs (generated in this study from a double reporter mouse) were derived from activated oocytes in 2i/LIF medium and used for screens. Biparental diploid ES cells expressing a destabilised GFP transgene (GFPd2) under control of the endogenous Rex1 promoter and carrying an EF1alpha-driven Cas9 cassette targeted to the Rosa26 locus (RC9 cells, (Li et al, 2018)) served as the parental cell line for the KO lines generated in this study. A list of KO cell lines can be found in Dataset EV6.

## Transposon-based saturation screen for key players of the exit from pluripotency

The gene-trap vectors 5'-PTK-3', TNN, TNP (Horie et al, 2011) together with haploid Rex1::GFPd2 ES cells (hRex1-GFPd2, Leeb et al, 2014) and haploid Rex1::mKO2-IRES-BSD/pOct4-GFP-IRES-Puro ES cells (hOR, this study) were used in 35 independent experiments to drive the screen to saturation. Mutant pools were generated by electroporation of 10$^7$ haploid ESCs using a Gene-Pulser Xcell (270 V, 500 µF, ∞ Ω, Bio-Rad) with 0.5 µg gene-trap

plasmid and 10 µg hyperactive transposase (hyPBase) (Yusa et al, 2011). 24 h after electroporation, selection was started for 4 days using 1µg/ml puromycin. Thereafter, cells were plated at a density of 10$^4$ cells/cm$^2$ in N2B27 medium in the absence of LIF or inhibitors to allow differentiation. After 7–10 days in differentiation conditions, GFP-positive cells were sorted and replated at a density of 10$^4$ cells/cm$^2$. After culture in N2B27 medium for a further 7–10 days, GFP-positive ESCs were sorted, expanded for 48 h in 2i/LIF medium and DNA was isolated using the PureGene kit (Qiagen) according to the manufacturer's instructions. Sequencing libraries were prepared using an optimised Splinkerette PCR protocol (Leeb et al, 2014) with a modified and improved set of adapters and primers (see Dataset EV6). Adapters were annealed at 50 µM each in T4 DNA ligase buffer by incubating at 97.5°C for 150 s followed by a temperature decrease of 0.1°C/5 s for 775 cycles. Ready-to-use adapters were stored at −20°C. Genomic DNA was isolated using the PureGene kit (Qiagen) according to the manufacturer's recommendations and quantified on the Qubit 2 fluorometer (Life Technologies) using the Qubit dsDNA Broad Range Assay (Life Technologies, Q32853). 2 µg of genomic DNA was diluted to a total volume of 120 µl with Low TE buffer (10 mM Tris–HCl, pH 8.0, 0.1 mM EDTA) and sheared to 250 bp on a Covaris S-series Sample Preparation System with the following settings: Duty Cycle—20%, Intensity—5, Cycles per Burst—200, Time—60 s, Temperature—4–7°C. After shearing, DNA was isolated with the QIAquick PCR purification kit (Qiagen) and quality assessed using the Agilent Bioanalyzer with the DNA high sensitivity kit (Agilent). End repair was performed using the NEBNext End Repair Module (E6050S) by mixing 100 µl sheared DNA, 11.6 µl buffer and 6 µl enzyme followed by incubation at 20°C for 60 min. After cleanup with the PureLink PCR Purification Kit (K3100-01), 42 µl of the end repaired DNA was A-tailed with the NEBNext dA-Tailing Module (E6053S) with 5 µl buffer and 3 µl enzyme at 37°C for 60 min. Following PureLink purification of the A-tailed DNA, the annealed adapters were ligated using the NEBNext Quick Ligation (E6056S) kit at 20°C for 30 min. Products were cleaned up using Ampure XP beads at a 1:1 ratio. Libraries were amplified using Kapa HIFI (Roche) with 1 µM PB5_pr1 and 1 µM SplAP1 during 18 cycles of 20 s 98°C, 20 s 63°C and 40 s 72°C with an initial denaturation at 95°C for 2 min and a final extension at 72°C for 5min. Samples were cleaned up using Ampure XP beads at a ratio of 0.8. A second round of PCR was conducted with 24 µl of PCR1 using the Kapa HIFI polymerase with 0.6 µM SplAP2 and 0.6 µM index primer for 13 cycles at an annealing temperature of 60°C with all other settings as in PCR1. Final libraries were quantified using the KAPA SYBR Fast qPCR Mix with primers SYB7 and SYB5 (Dataset EV6) at 0.5 µM each, and a final quality control was conducted with the Agilent Bioanalyzer DNA High Sensitivity Assay. Integration sites were mapped as in (Leeb et al, 2014) and described below.

## Generation of KO ESCs

Direct comparison between KOs necessitates a similar culture and passaging history. We therefore established an experimental pipeline to parallelise the generation of multiple KO ESC lines per week. 2 × 10$^5$ RC9 cells were transfected in a 6-well format with a set of two gRNA-containing vectors (Dataset EV6; (1 µg each), together with 0.5 µg of pCAG-dsRed by Lipofectamine2000 (3 µl per reaction).

12–16 h after transfection, medium was replaced. To enrich for transfectants, cultures were sorted for GFP/dsRed double-positive cells on a BD FACS Aria III 48h after transfection and plated at clonal density in 6-cm dishes (100–800 cells per dish) in ESDMEM-2i. Approximately 1 week after sorting, 48 colonies per gene KO were picked into 96-well plates, trypsinised and split into one expansion-plate and one plate for PCR genotyping after boiling lysis and proteinase K treatment (see PCR genotyping). Identified KO clones were expanded to 12-well or 6-well plates after 3 days of growth, and the remaining plate was frozen by adding 100 µl 2× freezing medium (80% FCS/ 20% DMSO) to 100 µl cell suspension as a backup. Clones were frozen from 12- or 6-well plates in freezing medium containing 50% FCS and 10% DMSO. The 73 cell lines generated by this strategy and analysed by RNAseq are listed in Dataset EV6.

### PCR genotyping

Crude DNA lysates were generated by pelleting half of a picked colony in a PCR plate at 500 g at 4°C, followed by two washes with PBS. 25 µl of water were added and cell pellets were boiled for 5min at 95°C. After cooling, Proteinase K was added at a final concentration of 3 µg/µl and incubated at 65°C for 1 h followed by inactivation at 95°C for 10min. Successful KO generation was confirmed by PCR, employing a three-primer strategy. Specific reverse primers for the deletion event and a possible wild-type allele were used in combination with a common forward primer (Dataset EV6) in separate PCRs. Cell lines with indels caused by a NHEJ event around the exonic gRNA1 target site were detected by Sanger sequencing of the "wild-type"-like band. Verification of the KO events was performed by the same PCR strategy on DNA, isolated using the PureGene DNA isolation kit (Qiagen) according to the manufacturer recommendations. For genotyping PCR, we used OneTaq (NEB) or JumpStart RedTaq (Sigma) PCR Master Mix according to the manufacturer's recommendations for cycling and a standard annealing temperature of 55°C and 35–40 cycles.

### qPCR genotyping

Cells grown in ESDMEM-2i were harvested in RLT buffer and isolated with the RNeasy mini kit (Qiagen) according to the manufacturer's recommendations. cDNA was transcribed using the Sensi-FAST cDNA Synthesis Kit (Bioline). Expression of exonic regions upstream of the deletion, the deleted exon and an intronic region downstream of the mutated exon was determined for the tested KO by qPCR using the Sensifast SYBR No Rox-Kit (Bioline). Primers are listed in Dataset EV6.

### Immunoblotting

Whole cell fractions were isolated using RIPA buffer, and protein concentrations were determined using a Bradford Assay (Bio-Rad). 20 µg whole cell lysate were separated on 8-12% SDS–PAGE gels (depending on the molecular weight of the target proteins) and subsequently blotted on 0.2 µm nitrocellulose membranes (Amersham). Membranes were blocked in TBS or PBS containing 5% milk and 0.1% Tween-20 or 3% BSA and 0.1% Tween-20 for 1h. Primary antibodies were incubated overnight at 4°C, at a dilution of 1/1,000 with the following exceptions: Anti-Gapdh 1/25,000, Anti-Tubulin

1/5,000, Anti-Vinculin 1/5,000, Anti-Smg5 1/200 and Smg7 1/2,000. Secondary goat anti-mouse IgG HRP (1:10,000) or goat anti-rabbit IgG HRP (1:15,000) were incubated at RT for 1h in blocking solution. Antibody binding was detected using the ECL select detection kit (Amersham). A list of antibodies can be found in Dataset EV6.

### Parallel differentiation

ES cells were differentiated in seven batches, always including WT controls, control KO ESCs (no phenotype expected) and a range of ESCs with expected weak to strong phenotypes. Where applicable two independent KO clones were used, other KO ESCs were cultured in replicates before initiation of differentiation. 24 h before starting the differentiation assay, the medium was changed from ESDMEM-2i to N2B27-based 2i/LIF medium. Cells were then trypsinised, counted and plated in 2i (without LIF) at a density of $10^4$ cells/cm$^2$ in two replicate wells of a 6-well plate for differentiation, one 6-well as undifferentiated control and 2 wells of a 12-well plate (undifferentiated control and differentiated sample) to monitor differentiation by FACS. 12h after plating, cells were washed twice with PBS and medium was changed to unsupplemented N2B27. For the undifferentiated controls, medium was replaced with fresh 2i. 24 h later, at N24, cultures were harvested in RLT buffer and stored at −80°C before isolation of RNA using the RNeasy Mini Kit (Qiagen). mRNA was isolated from 1 ug total RNA by poly-dT enrichment using the NEBNext Polya mRNA Magnetic Isolation Module according to the manufacturer's instructions. Samples were eluted in 15 µl 2x first strand cDNA synthesis buffer (NEBnext, NEB). After chemical fragmentation by incubating for 15 min at 94°C, the sample was directly subjected to the workflow for strand specific RNA-Seq library preparation (Ultra Directional RNA Library Prep, NEB). For ligation, custom adapters were used (Dataset EV6). After ligation, adapters were depleted by an Ampure-XP bead purification (Beckman Coulter) adding beads in a ratio of 1:1, followed by an index PCR (15 cycles) using Illumina compatible index primer. After double Ampure-XP beads purifications (with beads added in a ratio 1:1), libraries were quantified on a Fragment Analyzer run with a NGS Assay Kit (Agilent) and loaded on a HiSeq 3000 flowcell with 50 cycles single end sequencing by pooling the samples based on molarity aiming for 30 mio. reads per sample.

### 2h resolved differentiation time course

Medium was changed from ESDMEM-2i to N2B27-2i/LIF 24h before initiating the assay. Cells were trypsinised, counted and plated at a density of $10^4$ cells/cm$^2$ in 6-well plates in 2i (without LIF). 12h later, medium was changed to unsupplemented N2B27 (or 2i for the undifferentiated controls), and cells were harvested in buffer RLT every 2 h for the next 32h (until N32). RNA was isolated using the RNeasy Mini Kit (Qiagen). All libraries were generated from 500 ng RNA using the TruSeq Stranded mRNA Library Prep Kit (Illumina) and analysed on a HiSeq4000 machine to generate paired end reads at 75 bp read length.

### LIF signature

$1x10^4$ cells/cm$^2$ were plated in N2B27-2i and N2B27-2iLIF and grown for 24 h after which RNA was isolated using the RNeasy Mini

Kit (Qiagen). Libraries were made from 500ng high quality RNA using the Quantseq 3′ mRNA-Seq Library Prep Kit FWD for Illumina (Lexogen) according to the manufacturer's instructions. Libraries were quantified using qPCR according to the Quantseq Library Prep Kit and with the included primers, pooled and quality checked using an Agilent Bioanalyzer DNA High Sensitivity Assay (Agilent). Multiplexed samples were sequenced on a HiSeq4000 machine to generate single end reads of 50bp read length.

## RNAi experiments

For RNAi FlexiTube, siRNAs against Klf2 were used with AllStars Negative Control siRNAs (Qiagen). 20 ng siRNAs/$4 \times 10^4$ cells were transfected in 2i using DharmaFect (Dharmacon). 12 h later, the medium was changed to N2B27 after two PBS washes. At N24 and N32, Rex1-GFPd2 expression was determined by flow cytometry (see below).

## Flow cytometry

Cells were dissociated in 0.25% trypsin/EDTA, and trypsin was neutralised with DMEM supplemented with 15% FCS. After passing through a 40 μm mesh, Rex1-reporter activity was measured using a BD Fortessa machine. High-throughput measurements were performed in 96-well plates using the HTS unit of the BD Fortessa. Data were analysed using FlowJo software.

## Data analysis

### Statistical analysis

Established standard tests (e.g. for differential expression testing, gene set enrichment) were used throughout. Correction for multiple testing was performed whenever necessary. Details of individual statistical tests are provided in the following sections.

## Transposon mutagenesis screen in haploid murine embryonic stem cells

Integration mapping was performed as previously described (Leeb *et al*, 2014). Most gene-trap insertions were found in intronic sequences, consistent with the relative length distributions of genomic features. However, after normalising to available TTAA sites per region, TTAA sites in 5′UTRs and promoters (defined as < 500 bp upstream of the TSS) were overrepresented, whereas TTAA sites in introns were relatively depleted. 7,760 independent integrations passing cut-off criteria were mapped to 3,469 genes. Of those, 232 Genes (2,474 independent integration sites) were hit in five or more independent screens, indicating that those are integrations causative for the detected differentiation defect. To complement the candidate list and correct for gene-specific biases, we further developed a strategy to assign statistical strength to candidate genes by assessing the number of integrations per gene in relation to available TTAA transposon integration sites. We utilised the number of different TTAA sites within each gene to calculate the probability of having integrations in $k$ or more different locations by chance. All TTAA sites in mm10 were assessed using bowtie, and the analysis was restricted to genes classified as protein-coding, long non-coding RNAs and micro-RNAs. We computed the probability

that a gene is hit $k$ times given that it contains $n$ TTAA transposon integration sites assuming a binomial distribution (while $k$ is the observed number of integrations).

$$P[X \geq i] = \sum_{i=k}^{n} P[X = i] = \sum_{i=k}^{n} B(i|p_0, n) = \sum_{i=k}^{n} \binom{n}{i} p_0^i (1 - p_0)^{n-i}$$

$p_0$ was then calculated by $\frac{All\,hit\,TTAA-sites\,in\,genes}{All\,TTAA-sites\,in\,genes}$

We applied this test to genes with at least 26 TTAA sites, as genes with fewer TTAA sites would be significant ($P$-value < 0.05) with a single integration. Candidate gene lists between both analyses showed overlapping results and top hits were largely identical. This resulted in a list of 421 significant genes (adjusted $P$-value < 0.01 and hit in at least two independent screens).

## Guide RNA design criteria

To automate the design of sgRNAs as far as possible, we used the R package CRISPRseek (Zhu *et al*, 2014) to score the efficiency and likely off-target binding of candidate sgRNA sequences. For a selected gene, the algorithm proceeds as follows:

Retrieve the sequence of the first protein-coding exon.

1. Sequence has to be at least 50nt long.
2. Find sgRNAs within the sequence and score them.
3. If no appropriate sgRNA could be found, move to next exon.
4. Still in the first third of the coding sequence?
5. Find a second sgRNA in an intron at least 5kb, but no further than 30kb downstream.
6. Score the sgRNAs.

A minimal efficiency of 0.2 and an off-target score < 100 was required.

## PCR primer design strategy for simplified knockout validation

The surrounding regions of the sgRNAs (min distance 20 bp) were used as input for PRIMER3 primer predictions. Primers were designed to have a length of 18–27 bases with an optimum of 20 and to have melting temperatures between 57 and 63°C with an optimum of 60°C. All primers were checked for off-target binding sites by using Blastn. Blastn was called with the parameters:

-task megablast -use_index false -word_size 7 -perc_identity 65 –evalue 30000 -max_target_seqs 50 -num_threads 4 –outfmt 7.

## RNA quantification, RNAseq

Quality control was performed using fastQC (version 0.11.5). Transcripts were mapped to the mm10 mouse reference genome and counts determined using STAR (version 2.5.3). FPKM values were calculated using DESeq2 (version 1.24.0).

## RNA quantification, QuantSeq

Transcripts from RC9 cells in 2i and 2i/LIF medium were measured using QuantSeq (Lexogen) and quantified as described. Reads were trimmed with bbduk (version 35.92). Quality control was performed using fastQC (version 0.11.5). Transcripts were mapped to the mm10 mouse reference genome using STAR (version 2.5.3). After

indexing with samtools (version 1.3), reads in genes were counted using HTSeq-count (version 0.6.0). Counts were subjected to differential expression analysis.

## Differential expression analysis

Differential expression analysis was carried out using limma (version 3.40.6), after transforming transcript counts using the included voom function. We then fitted a linear model to sample measurements from each combination of knockouts (including RC9 control) and conditions (2i and N24). RNAseq batches were included in the model as a confounding factor and batch-corrected using removeBatchEffect for visualisation. Contrasts were fitted to determine the expression differences between (i) each knockout sample in 2i and the RC9 control, (ii) each knockout sample in N24 and the RC9 control, and (iii) the difference between the log fold changes calculated in (i) and (ii) (interaction effect). We also fitted a contrast to determine changes during normal differentiation, i.e. the differential expression between RC9-N24 versus RC9-2i. All fitted contrasts were tested for differential expression using moderated t-statistics through the limma function treat (H0 |log2FC| < log2(1.5)). $P$-values were corrected for multiple testing using the Benjamini–Hochberg method (FDR). For subsequent analyses, the more stringent $P$-values and FDR values as calculated by treat were used (unless noted otherwise). Differential expression analysis of 2i/LIF versus 2i RC9 measurements was carried out separately, but in an identical manner.

## GO enrichment analysis

Mouse gene GO annotations were extracted from the R package org.Mm.eg.db (version 3.4.0). GO terms that included between 5 and 500 genes were selected for further analysis. For each gene list of interest, significance of functional enrichment of GO terms compared with the background list was determined using Fisher's exact tests. The background was restricted to genes whose transcripts were detected at a median count above 5. To reduce redundancy of functional categories, we clustered GO terms that differed in 5 or fewer genes of interest (using the R base function hclust on the L1-distance of the binary membership matrix). The smallest GO term by total annotations was selected independently from Fisher test results as the primary, i.e. most specific term to represent each cluster. We then carried out correction for multiple hypothesis testing using the Benjamini–Hochberg method on all primary GO terms to obtain adjusted $P$-values.

Pathway enrichment was performed in the same way as described above for GO terms with Reactome pathway annotations taken from the reactome.db R package (version 1.68.0).

## Naïve marker dependency of N24 KO expression patterns (NAGs)

To quantify the similarity of each gene's log2 fold change across KOs at N24 with the log2 fold change of seven core pluripotency markers (*Nanog, Esrrb, Tbx3, Tfcp2l1, Klf4, Prdm14 and Zfp42*), multiple regression analysis was performed. That is, each gene's DE pattern was modelled as a linear combination of marker DE patterns, quantifying the strength of association as their shared variance (multiple-$R^2$). NAGs were defined as having a naïve marker $R^2$ of $\geq 0.65$.

## Cluster analysis (constitutive and N24-induced)

Based on the differential expression analysis of all KOs (excluding *Suz12, Eed & Csnk1a1* KOs) in both N24 and 2i, and their interactions with the RC9 control, we defined two main categories of KO-responsive genes: (i) the constitutive knockout response: genes that were significantly changed (adj. $P \leq 0.05$) in the same direction in both N24 and 2i, in at least one knockout condition. Genes from the N24-induced KO response category were excluded. (ii) N24-induced knockout response: genes that were significantly changed in both N24 and the knockout:RC9 interaction term, in at least one knockout condition. To determine whether genes in the N24-induced knockout response subset were overall more correlated with naïve pluripotency markers, we checked the distribution of naïve marker multiple-$R^2$ values in the N24-induced knockout response set compared with both the background distribution and genes classified as part of the constitutive KO response.

To determine whether distinct functional clusters of genes that may be activated in all, single or sub-groups of knockouts, we clustered genes in either of the categories (constitutive and N24-induced response) based on their $KO^{N24}$ versus $RC9^{N24}$ log fold changes, using the R base package hclust with the "Ward.D2" method. We determined the total within-cluster variance at different numbers of clusters. The number of clusters was chosen such that the model improved strongly by increasing the number of clusters up to that value, but only marginal gains result from increasing it further. Thereby, 12 gene clusters in the constitutive response and 12 clusters in the induced response were identified. GO enrichment analysis was carried out as described above for member genes of each cluster. Here, we required at least five genes of a cluster to be enriched in a GO term to be considered.

## Differentiation pathway correlation analysis

We defined downstream responses of key differentiation pathways using the most specific responses to the knockouts of their upstream regulators: *Tsc2* (mTORC1), *Ptpn11 & Fgfr1* (Fgf/Erk signalling), *Tcf7l1* (Wnt signalling) and *Rbpj* (Notch signalling). In addition, LIF-specific changes were defined as the response of genes in RC9 cells in the 2i/LIF compared with the 2i condition. To ensure specificity, we required that the log fold change of each response gene for a given regulator (in N2B27 medium) exceeded that observed in the remaining conditions by a factor of two. In the case of Ptpn11 and Fgfr1, this criterion had to be met by the lower of the two log fold changes for each gene. Additionally, the genes were required to satisfy a FPKM (CPM in case of LIF) cut-off of at least 1 in either the KO or the WT. From all remaining, significantly expressed genes (FDR $\leq 0.05$) we then selected, by absolute log fold change in descending order, the top 50 most specific genes for each pathway (Appendix Fig S7A).

Pathway activity was then determined across all KOs at N24. For each pathway, the log2 fold changes of the 50 pathway-specific response genes from the associated regulator knockout ($KO^{N24}$ versus $RC9^{N24}$) were selected. We then correlated these log fold changes to the log fold changes of the same genes in all other knockouts at N24 (and in 2i for the LIF pathway footprint, which was derived by the contrast 2i versus 2i/LIF). A positive correlation indicates that the pathway is disrupted in a manner similar

to knocking out its regulator, whereas a negative correlation indicates the opposite. In the case of LIF-specific genes, a positive correlation indicates an effect similar to that of adding LIF. *P*-values of correlation values were calculated based on Fisher's Z transform. *P*-values were then corrected for multiple testing using the Bonferroni method.

To determine how predictive the calculated correlations were of the differentiation phenotype (mean naïve marker log2FC versus RC9 at N24), we carried out multiple linear regression and extracted the coefficients of the model to establish the relative contribution of each pathway signature to naïve marker deregulation.

### Quantification of differentiation delay in knockouts

Raw RNA-seq data were aligned to the mm10 genome, and read counts were obtained using the STAR aligner (version2.5.2b). Gene expression for the following biotypes was quantified: protein_coding, misc_RNA, miRNA, scaRNA, scRNA, snoRNA, snRNA and sRNA. Read counts were normalised using DESeq2 (version 1.22.2) and subsequently the log2fc between all samples and the mean expression of the 2i samples was calculated. This resulted in expression profiles for each gene over 32 h of differentiation.

Gaussian process regression was used to smoothen the profiles of each gene (R package tgp version 2.4-14). The expression profiles of 73 knockouts at N24 were mapped onto the differentiation axis from the time course to quantify differentiation delays per KO, i.e. we aimed to quantify how many hours the KO expression pattern is "behind" the expected differentiation in WT cells. This "mapping" to the time axis was done first computing the log2FCs between N24 versus 2i for each KO. The resulting log2FC profiles were compared with the log2FCs at each time point during the WT differentiation. We assumed that the time point at which the difference between the WT profile and the KO profile is minimised best reflects the molecular differentiation state of the respective KO.

We used the Euclidean distance for this purpose. To make distances better interpretable between the knockouts the distances were set to:

$$\frac{\text{Euclidean distance} - \min \text{Euclidean distance}}{\max \text{Euclidean distance} - \min \text{Euclidean distance}}$$

where the maximum and minimum distances refer to the respective maximum and minimum Euclidean distances of the respective KO across all time points (i.e. the worst and the best matching time point). Thus, the best matching time point will get a distance score of 0.

In order to further increase the time resolution, additional time points were imputed via linear interpolation. Here, we interpolated expression profiles every 15 min between the two neighbouring time points of the time point with the smallest Euclidean distance. Timing of knockouts was repeated on the interpolated time points, and the new minimal distance was used to quantify the differentiation delay.

### Prediction of cluster expression by pathway activity

Pathway activities for each pathway in all KOs were calculated using following formulas:

$$
\begin{aligned}
\text{directed } FC(KO) \\
= FC\left(\frac{KO}{WT}\right) * \begin{cases} LIF \begin{cases} +1 \; same\;direction\;as\;in\;\frac{LIF}{2i}comparison \\ -1\;opposite\;direction\;as\;in\;\frac{LIF}{2i}\;comparison \end{cases} \\ Tsc2,P11F1,Tcf7l1,Rbpj \begin{cases} +1\;same\;direction\;as\;in\;\frac{rep,KO}{WT}\;comparison \\ -1\;opposite\;direction\;as\;in\;\frac{rep,KO}{WT}\;comparison \end{cases} \end{cases}
\end{aligned}
$$

$$directed\,avg.\log FC_{KO} = \frac{\sum_n^{PW\,specific\,marker\,genes} directed\,FC(KO)_n}{\sum_n^{PW\,specific\,marker\,genes} directed\,FC(rep.KO)_n}$$

$$pathway\,activity_{KO} = directed\,avg.\log FC_{KO} * \begin{cases} 1\,for\,LIF,Tsc2 \\ -1\,for\,P11F1,Tcf7l1,Rbpj \end{cases}$$

The combinations of pathway activities and average cluster expression changes over all KOs were used to build a multiple regression model to predict the expression changes in each cluster by the five pathway activities. Connections between pathway activities and cluster expression changes not passing a significance cut-off (adj. *P*-value $\leq 0.01$) of the multiple regression models were excluded from each model. The remaining models were tested in the time course data. Connections from clusters where prediction of expression changes based on pathway activities did follow or contradict the observed expression changes in the time course were excluded.

### Embryo comparison

Sequencing data were obtained from the European Nucleotide Archive (Toribio *et al*, 2017) from single-cell mouse embryo profiling studies (Deng *et al*, 2014; Mohammed *et al*, 2017). The macaque FKPM expression data set was provided by (Nakamura *et al*, 2016). Orthologues genes mapping in 1-to-1 fashion were used during comparative analysis. Alignments to gene loci were quantified with htseq-count (Anders *et al*, 2014) based on annotation from Ensembl 87 (Toribio *et al*, 2017). Principal component and cluster analysis were based on log2 FPKM values computed with the Bioconductor package DESeq (Anders & Huber, 2010), custom scripts and FactoRmineR packages (Lê *et al*, 2008). Highly variable genes were calculated by fitting a non-linear regression curve between average log2 FPKM and the square of coefficient of variation. Specific thresholds were applied along the x-axis (log2FPKM) and y-axis (coefficient of variation) to identify the most variable genes. Fractional identity between the bulk RNAseq (2i, N24) and the mouse, macaque embryo or human capacitation data set were computed using R package DeconRNASeq (Gong & Szustakowski, 2013).

### GENEES (Genetic Network Explorer for Embryonic Stem Cell Differentiation) Application

To both visualise and interrogate gene expression analysis of all KO ES cells, we have set up the online tool GENEES (Genetic Network Exploration of Embryonic Stem Cell Differentiation). It is accessible at [http://shiny.cecad.uni-koeln.de:3838/stemcell_network/]. The app integrates multiple visualisations and summary statistics to make all information directly available for exploratory analysis. The app has 3 different tabs for inspection of single genes, pre-computed clusters and custom gene sets. Single genes can be inspected in the

"Genes tab", while previously described constitutive clusters and induced clusters can be further inspected in the "Pre-computed clusters" tab. In the "Select custom geneset" tab, a set of genes of interest subjected to analysis can be defined and analysed.

## Data availability

The data sets produced in this study are available on the Gene Expression Omnibus database under: GSE145653 (https://www.ncbi.nlm.nih.gov/geo/query/acc.cgi?acc = GSE145653).

**Expanded View** for this article is available online.

## Acknowledgements

We thank Johanna Stranner, Thomas Sauer and Andy Riddell for help with flow cytometry, Meng Li and Kosuke Yusa for sharing RC9 cells, Andreas Dahl for NGS-sequencing support and Christa Bücker for critical comments on the manuscript. ML is funded by a WWTF-VRG grant (VRG14-006). This study was funded by an FWF/DFG DACH grant to AB and ML (FWF grant number: I 3786; DGF grant number 398882498). AB and MG were supported by the BMBF (Sybacol). RS received support by the Cologne Graduate School of Ageing Research. LS is part of the FWF doctoral programme SMICH and supported by an Austrian Academy of Sciences DOC Fellowship. Wellcome and the Medical Research Council provide core support for the Wellcome-MRC Cambridge Stem Cell Institute. AS received funding from the Medical Research Council (MR/P00072X/1) and is an MRC Professor. This project was initiated within the EU FP7 integrated project SyBoSS (Grant agreement: 242129) co-ordinated by AFS and AB including partners ML and AS.

## Author contributions

Conceptualisation of study: ML, AB, AS and AFS. Study design: ML and AB. Experiments: ML, AL, AB, RS, MG and FT-T. Wet-laboratory experiments: AL, MH, JR, PvdL, HFT, LS, EG, ML and MS. Bioinformatic analyses: RS, FT-T, MG, GGS, MR and AL. Manuscript writing: ML, AB, AS, AFS, AL, RS, MG, MH, FT-T and GG. Funding: ML, AB, AFS and AS.

## Conflict of interest

The authors declare that they have no conflict of interest.

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
