## [Review Process File · The EMBO Journal]

Cooperative genetic networks drive embryonic stem cell transition from naïve to formative pluripotency

Martin Leeb, Andreas Lackner, Robert Sehlke, Marius Garmhausen, Giuliano Stirparo, Michelle Huth, Fabian Titz-Teixeira, Petra van der Lelij, Julia Ramesmayer, Henry F. Thomas, Meryem Ralser, Laura Santini, Elena Galimberti, Mihail Sarov, Francis Stewart, Austin Smith, and Andreas Beyer

DOI: [10.15252/embj.2020105776](https://doi.org/10.15252/embj.2020105776)

Corresponding authors: Martin Leeb (martin.leeb@univie.ac.at) , Andreas Beyer (andreas.beyer@uni-koeln.de)

Review Timeline:

Submission Date:	1st Jun 20
Editorial Decision:	14th Jul 20
Revision Received:	16th Nov 20
Editorial Decision:	18th Dec 20
Revision Received:	18th Jan 21
Accepted:	19th Jan 21

Editor: Daniel Klimmeck

Transaction Report:

Dear Dr Leeb,

Thank you for the submission of your manuscript (EMBOJ-2020-105776) to The EMBO Journal. Please accept my apologies for the extended duration of the peer-review process due to protracted referee input and detailed discussions in the team. Your manuscript has been sent to three reviewers, and we have received reports from all of them, which I enclose below.

As you will see, the referees acknowledge the potential novelty of your results and value as a resource to the field, although they also express a number of issues that will have to be conclusively addressed before they can be supportive of publication of your manuscript in The EMBO Journal. In more detail, referee #3 states that the claims made on network cooperation and decommissioning-based cell identity loss during pluripotency exit are not sufficiently supported by the results (ref#3, pt.6). Further, this referee asks you to address potential sequentiality of differentiation events (ref#3, pt.1,2). Referee #2 agrees that cooperative requirement of exit factors is not supported and asks you to consolidate the singular role of Csnk1a1 (ref#2, pt.5). In addition the referees raise a number of points related to data presentation, controls required, discussion of the findings as well as preceding literature, which would need to be conclusively addressed to achieve the level of robustness and clarity needed for The EMBO Journal.

I judge the comments of the referees to be generally reasonable and given their overall interest, we are in principle happy to invite you to revise your manuscript experimentally to address the referees' comments.

Please let me know any time if you have additional questions or need further input on the referee comments.

In light of the criticism raise by the reviewers, I would appreciate if you could contact me during the next weeks via e.g. a call to discuss your perspective on the comments and potential plan for revisions.

Please see below for additional instructions for preparing your revised manuscript.

Thank you for the opportunity to consider your work for publication. I look forward to your revision.

Kind regards,

Daniel Klimmeck

Daniel Klimmeck, PhD
Editor
The EMBO Journal

Before submitting your revision, primary datasets (and computer code, where appropriate) produced in this study need to be deposited in an appropriate public database (see <https://www.embopress.org/page/journal/14602075/authorguide#datadeposition>).

The accession numbers and database should be listed in a formal "Data Availability" section (placed after Materials & Method) that follows the model below (see also <https://www.embopress.org/page/journal/14602075/authorguide#availabilityofpublishedmaterial>). Please note that the Data Availability Section is restricted to new primary data that are part of this study.

Data availability

Our journal also encourages inclusion of *data citations in the reference list* to directly cite datasets that were re-used and obtained from public databases. Data citations in the article text are distinct from normal bibliographical citations and should directly link to the database records from which the data can be accessed. In the main text, data citations are formatted as follows: "Data ref: Smith et al, 2001" or "Data ref: NCBI Sequence Read Archive PRJNA342805, 2017". In the Reference list, data citations must be labeled with "[DATASET]". A data reference must provide the database name, accession number/identifiers and a resolvable link to the landing page from which the data can be accessed at the end of the reference. Further instructions are available at <https://www.embopress.org/page/journal/14602075/authorguide#referencesformat>

- a point-by-point response to the referees' comments, with a detailed description of the changes made (as a word file).

- a word file of the manuscript text.
 - individual production quality figure files (one file per figure)
 - a complete author checklist, which you can download from our author guidelines (<http://emboj.embopress.org/authorguide>).
 - Expanded View files (replacing Supplementary Information)
- Please see out instructions to authors
<https://www.embopress.org/page/journal/14602075/authorguide#expandedview>

The revision must be submitted online within 90 days; please click on the link below to submit the revision online before 12th Oct 2020.

Link Not Available

Referee #1:

In this extensive study, the authors assess the regulatory genetic networks that govern the cell state transition from naïve to formative state of pluripotency in mouse embryonic stem cells (mESCs). By performing genome-wide saturated loss-of-function screens in haploid mESCs, they reveal dozens of genes that exhibit a delay phenotype during this cell fate change, when mutated. They focus on and further analyze 73 such genes, generating KO cell lines and performing RNAseq analyses during their transition from naïve to formative pluripotency. Using the transcriptome datasets they generated, along with several previously published ones, they classify these 73 regulator genes according to: 1) their regulation of naïve- and formative pluripotency-associated genes during naïve-to-formative pluripotency transition, 2) the stage of normal development that their mutants are delayed at, 3) the resemblance of the transcriptomes of their mutants to the in vivo developmental stages in mouse and primate peri-implantation development, 4) their regulation of 5 pluripotency-associated signaling pathways and the transcriptional response gene networks downstream of these pathways. They also utilize the transcriptome data of the mutant mESC lines to define an extended naïve pluripotency network that shows a high similarity between the in vitro and in vivo epiblast transitions. This work addresses an interesting question in cell biology regarding cell fate changes during early development, in an unbiased way, and using a robust cellular model. Therefore, the manuscript is suitable for publication either as a Research Article or as a Resource manuscript.

Enclosed are a few minor points that should be addressed to improve the manuscript:

- 1) In Supplementary Figure 1c, the authors show enrichment of a number of GO terms related to differentiation of specific tissues. It would be nice if they could discuss it shortly to explain the overlap of genes related to specific tissue differentiation pathways and the naïve-to-formative pluripotency transition.
- 2) In Supplementary Figure 1f, the authors list the *Csnk1a1* KO as one of the KOs with full protein deficiency, although a residual band is visible in the KO lane. Given their observation on this mutant regaining its differentiation potential after several passages, the authors should clarify whether this reversal of phenotype is due to residual protein in their mutant line.
- 3) There are several mistakes in the numbering of the Figures in the main text or figure legends. (e.g., Figure S1a referral in P9, line 138; Figure 6d referral in P22, line 447; missing legend of Supplementary Figure 4i) The authors should make sure that the right figures are referred to throughout the text. Similarly, some Figure panels are not mentioned in the text (e.g., Supplementary Figure 4c and Supplementary Figure 5a).

Referee #2:

Summary:

The authors began with a mutagenesis screen to identify factors that prevent *rex1-egfp* downregulation following 2i withdrawal, which normally induces state transition. Following that, they made a number of CRISPR mutant ESC lines [Fig 1] for a more in-depth investigation of these factors regulating state transition. They investigated how these CRISPR mutations affect expression of specific ESC marker genes, as well as global transcriptome [Fig 2]. They also studied these CRISPR mutants using a differentiation time-course to quantify differentiation/transition delay [Fig 3]. They showed that the state transition regulatory program is largely preserved in these CRISPR mutant lines [Fig 4]. They proposed to extend the list of naive pluripotency associated genes from an original core 7 to 300+ NAGs [Fig 5, 6]. Finally, they defined a list of key (but not exhaustive) 5 signaling pathways regulating the naïve to formative ESC state transition [Fig 7].

Comments:

1. The original screen was performed in haploid ESCs, but all subsequent follow-up work (CRISPR mutants generation & characterization) seems to have been done on diploid ESCs. Reasons/rationales for this discrepancy unclear, which assays used which type of cell and why?
2. While a good effort was made to generate 73 ESC KO lines, it seems that validation of absence of protein (or even RNA) was only attempted for one quarter of these. Additionally, it is also unclear whether CRISPR disruption resulted in homozygous or heterozygous KO—one would normally assume homozygous, but protein still present (at reduced level) for at least one tested gene *Mapk1* [Sup Fig 1]. Please clarify/ show evidence.
3. CRISPR genome editing may cause other unintended off-targeted mutations on both small (few bp deletions) and large (chromosome rearrangements/ deletions) scales. What attempts have the authors made to check for/ rule out these?
4. Rescue-by-reintroduction experiments [Sup Fig 1G/H] are nice in theory, but what level is rescue protein expressed at relative to WT?
5. The authors repeatedly make the point that no single gene KO blocks exit from naive

pluripotency -- except for *Csnk1a1*? This point needs more clarification.

6. More data needs to be shown related to gene expression clustering of the CRISPR mutants, (for example visualization of clustering by media condition? Mutated gene? in addition to clustering by batch shown in Fig S1). The reviewer is aware that there is a web app, but this is critical data and should be in the paper too.

7. In Fig 2A the authors provide evidence that their CRISPR KOs did not show reduced naive transcription factor expression - data is qPCR? Is this also corroborated by protein expression/other functional assays?

8. In addition to *rex1-gfp* expression the authors also quantify differentiation/transition delay by expression of a defined gene set & global transcriptomic profile [Fig 2]. It is promising that they correlate (Fig 2C), but can the authors demonstrate any other functional consequences of this differentiation/transition delay on ESC behavior/properties?

9. Similarly, for the differentiation/transition time-course defined in Fig 3, are there any other non-RNA-level based quantifiable features (e.g. Rex1 protein level?) that the authors can use as additional validation?

10. In the final figures, the authors highlight many signaling pathways, and even more pluripotency-related genes/pathways, but the implications of these findings/ categorizations not made clear (it reads like a data dump).

a. For example, of the 5 signaling pathways, how do they differently affect pluripotency state transition?

b. Similarly, of the 1800+ genes/12 clusters in the constitutive KO response group and 720 genes/12 clusters in the N24 group, which were the most important / most predictive of delay severity, etc?

Referee #3:

- general summary and opinion about the principle significance of the study, its questions and findings

The manuscript of Lackner et al aims to identify the genetic networks that guide the transition between two pluripotent states in culture, from the naive pluripotent state of mouse embryonic stem cells (ESCs) to the formative pluripotent state these cells differentiate into upon withdrawal of ESC self-renewal factors. The authors perform a transposon-based mutagenesis screen in haploid ESCs similar to one they performed earlier (Leeb et al 2014, Cell stem cell 14), but now driven to saturation. They identify 489 candidate genes whose mutagenesis causes a delay in downregulation of the naive marker *Rex1* in differentiation conditions. For 73 selected genes, they construct diploid KO ESC lines, and determine their transcriptomes after 24 hrs of differentiation. They correlate these data with a fine-grained time-course ESC differentiation sequence to determine the extent of differentiation delay, group the mutants into categories, and link them to signalling pathways and common gene modules.

The authors have generated a massive and valuable dataset, and performed several ingenious analyses. Nonetheless, the conclusions are somewhat underwhelming and we learn relatively little

of note about pluripotency transitions. During the presentation of the results the authors discuss various more or less interesting observations, such as a link between *klf2* and *tfc7l1*. However, despite this link having been noted before, they don't perform in depth analyses of these observations. The conclusions are very broad and general and it is often not explained how they are supported by the data. This already starts with the title: "Cooperative genetic networks drive a mammalian cell state transition": nowhere in the manuscript is cooperation between different genetic networks in driving differentiation investigated. Or: "Our data support a cell fate transition model in which extinction of cell identity induced by the interplay of signalling and the chromatin modification machinery precedes the initiation of a novel cell fate." No criteria or assays to determine extinction of cell identity or initiation of a novel cell fate are presented so this is in fact an opinion rather than a conclusion.

This lack of insightful conclusions is surprising because interesting questions can be addressed using the newly generated data. It seems to me that the authors adhere to a pre-conceived model of the naive-formative transition, or exit from naive pluripotency as they frequently describe it, as a single monolithic event and simply state without any reasoning that all their observations support this model. Thus, they conclude that signalling pathways and gene modules are all harnessed in parallel to create a vastly complex 'exit machinery' that breaks down the naive pluripotency network although their data fits much better with a model in which the pathways/modules act sequentially and/or independently, during different phases of the transition.

There is indeed plenty of evidence that there are multiple steps and cell biological functions involved in the naive-formative transition, including recent evidence for an intermediate 'rosette' pluripotent state controlled by the sequential action of Wnt and Mek signals during the naive-formative transition (Neagu et al 2020 Nature cell biology 22, 534), the induction of transcription factors that rewire enhancer networks (e.g. Matsuda et al 2017 Development 144, 1948; Yang et al 2014, Cell reports 7, 1968; Buecker et al 2014 Cell stem cell 14, 838), adjustments in adhesive and morphologic properties (Veluscek et al 2016 Stem cells 34, 1213; Kalkan et al 2017 Development 144, 121), and stepwise action of signalling pathways and transcription factors (the earlier mentioned Neagu et al; Kalkan et al 2019 Cell stem cell 24, 785; Nett et al 2018 Embo reports 19, e45642). The authors should try to find evidence for these steps and functions in their data, and find gene (modules) that operate them. For example, is there a category of KOs that predominantly affects adhesive/morphological properties? Are there categories predominantly associated with the naive-rosette or rosette-formative transition? Can a sequence of events be generated that occurs during the naive-formative transition? By breaking up the transition into discrete events, such analyses would do much more to unravel and assign functions to the gene networks guiding the naive-formative transitions. This study could then constitute an important contribution to our understanding of pluripotency. In its current form, however, it is mostly a collection of data that is of potential interest and in need of further analysis.

- specific major concerns essential to be addressed to support the conclusions

1. As described above, the authors should try to find evidence for a sequence of distinct events or states in the naive-formative transition, and find gene (modules) that operate them. This could then be incorporated into fig 7b, which is not very informative as it is now.
2. In the introduction the authors state as their goal to explore regulatory principles of the exit from naive pluripotency. What principles have been identified?
3. In general, number of replicates and statistical measures are not indicated in many figures.
4. What state is N24? Earlier studies show that N24 cells contain many Rex1+ cells that are not formative. Why was this time point chosen? What was done to accommodate the heterogeneity of

the cell population?

5. From 26 genes the deletion was verified by western or rtPCR/sequencing or rescue; how about the other 57 genes? How were the 73 KO genes selected?
6. p7 line 92-98: Most of the genes identified in the screens did not change between 2i and N24. The authors conclude from this that ESCs are poised for rapid decommissioning of naive identity, but the reasoning behind this is not explained. One thing that should be discussed from this is whether the idea that induction of an 'exit machinery' by differentiation signals is required to exit from the ground state is correct. It further seems to me that genes that are necessary for (part of) the naive-formative transition would be present in naive cells and only downregulated after the process in which the gene is involved is completed. That there is no change in expression at N24 suggests that the transition is not yet complete, so not particularly rapid. Alternatively, and referring to point 4, the cell population is heterogeneous and part of the cells have not yet switched to formative, maintaining expression of the identified genes. Either way, the data do not support the author's conclusion.
7. p9 l146: The authors argue that the lack of effect of 'exit factors' on naive factors in 2i indicates the neutralisation of 'differentiation drivers' in 2i. These differentiation drivers would have been hit in the screen. How do the authors define exit factors and differentiation drivers, and which of their hits belong to each of these categories? How do they think neutralisation of differentiation drivers occurs, and is there any direct evidence for this?
8. I applaud the authors for creating a webtool to access the transcriptome data but I regret to say that I was unable to figure out how to use it or what I could do with it.
9. Fig S2c: this figure is supposed to show increased rex1 downregulation following Klf2 depletion, but it could well be argued that it shows the opposite.
10. Fig 2b: It is claimed that the differentiation delay of Tcf7l1 KO ESCs is dependent on Klf2 because Rex1 downregulation speeds up upon Klf2 KD. However, the figure doesn't show the effect of Klf2 KD on Rex1 in Tcf7l1 KO ESCs in 2i. Klf2 KD could cause downregulation of Rex1 without affecting differentiation speed. Also, the interaction between Tcf7l1 and Klf2 was already reported in Leeb et al (2014) CSC 14, 385 and in Martello et al (2012) CSC 11, 491; which novel insight is gained with the data here?
11. p12l212: stronger than expected effect of CHIR on Jmjd1c KO, suggesting cooperation between Jmjd1c and Tcf7l1. Could the authors elaborate on how this could work? Perhaps Tcf7l1 relies to some extent on Jmjd1c to repress target genes? Is there any evidence for this?
12. Fig 2e: CHIR delays differentiation of jmjd1cKO much stronger than that of WT. However, ESCs in CHIR without other factors for 48 hrs may experience a lot of cell death, and it may be preferentially the naive cells that die. The effect of the Jmjd1c KO (and of other KOs) might therefore be predominantly on survival rather than differentiation. Have the authors considered this effect? Can they account for this by quantifying numbers of input and output cells to determine the extent of cell loss, or demonstrate that cell death/survival plays no role?
13. p12l219: " We then compared the expression patterns of specific gene-sets in a given KO at N24 to the expression of the same gene-set along the WT 2i to N32 trajectory." Which specific gene-sets were used? How were they selected? Are they the same for each KO?
14. Fig 4b-e, S4a: WT cells are not indicated in many panels. In contrast to what is stated in the main text, the N24 samples do not seem to show higher FOI to E5.5 than to E4.5. Seems we should look to 4c-e but these figures are not clear, one cannot identify how each KO line shifts from 2i to N24. Axes are different, which makes any comparisons even harder. Even not all KO lines that are mentioned in text are labelled in figure.
15. Fig 5a: Authors should compare top 374 instead of 250 differentially expressed genes with the 374 NAGs. Since NAGs are downregulated, the authors should compare them with the 374 most downregulated genes. What does the 1-0.25 at x-axis mean?
16. p17l321 and Fig 5b-d: " Strikingly, NAGs showed a near identical fold change behaviour in vitro

and in vivo during the epiblast transition from E4.5 to E5.5." Superficially yes, but the behavior of the individual genes look quite different and rho-values have been omitted, suggesting that the fold change is only by accident similar and doesn't reflect conserved behavior.

17. p20l394: "We conclude that signalling pathways are harnessed in parallel...". The data do not demonstrate this because they are perfectly compatible with a mechanism in which the signalling pathways act sequentially. See also the general summary.

18. p21l416: how was clustering of the constitutive KO response genes performed? Why did it result in 12 clusters? Is there biological relevance to the 12 clusters or could there just as well have been 11, 14 or 21 clusters?

19. p22l452: "In summary, constitutive and N24 induced clusters define crucial genetic modules that are co-regulated by multiple exit factors."; p23l465: "Together, this shows that at least 24 distinct transcription modules can be defined, whose expression is deregulated upon exit gene depletion and which are largely under the control of five key signalling pathways gating differentiation." Of course, that is how the clusters were defined. The cluster analyses are potentially interesting but the authors should take the analysis further and not stop at self-evident conclusions. Fig 6c,e show potential biological functions and it should be further investigated which of these functions could play a role in the naive-formative transition. These can then be correlated to the time course data and an outline generated for a functional sequence of events in the transition, as also described under 1.

20. p23l458: "Adding further evidence to the pathway to cluster relationships, we found that in most cases activity changes of the regulatory pathways preceded the expression changes of their putative target clusters over the 2h differentiation time course." I assume this is demonstrated in figs S7a-f, but the authors should present these figures and explain how they demonstrate this.

21. p25l498: "Notably, however, we observed segregation in terms of pathway deregulation. Thus, KOs that induce e.g. a Tcf7l1-like profile were less likely to show a signature of another pathway. This suggests that ESC differentiation requires coordinated activity changes of independent pathways under control of separable genetic networks." It was recently demonstrated that WNT signals regulate a different step in the naive-formative transition than other pathways (Neagu et al 2020 Nature cell biology 22, 534). This argues that the pathways are segregated because they act during different phases of the transition. The authors should discuss their results in light of these earlier findings.

22. p25l502: "The observation that single gene depletion is not sufficient to prevent exit from the naïve state is consistent with this interpretation." The authors should discuss the possibility that a single KO is not sufficient to completely block a pathway, e.g. there is redundancy in Tcf factors, as well as in Mapk factors.

- minor concerns that should be addressed

23. First sentence of introduction: "Mouse embryonic stem cells (ESCs) can self-renew in defined conditions in a ground state of naïve pluripotency." Can the authors explain here what the salient differences are between ground state and naive pluripotency? Is there a non-ground state of naive pluripotency?

24. How was pathway analysis in Fig S1b performed?

25. What do colors in bars of fig S1bc mean? Why different colors in the same bar?

26. Fig S1f: ERK2 heterozygote looks like it upregulated ERK1.

27. Fig S1g: what is EV sample? Jarid2 is missing.

28. Fig S1h: Why is the rescue for Etv5, Rbpj not complete?

29. Fig 1e: legend says 30h, text/figure 24h differentiation.

30. Fig S1i: how is batch correction done?

31. S1m: 2i sample is missing.
32. S1n: Pum1 KO in 2i sample is missing.
33. p9 line 138: wrong reference S1a.
34. Combinations of six, seven or ten naive TFs are used in various analyses, e.g. to determine the developmental delay. What are the reasons to use different combinations for different analyses?
35. Fig S2a,b: are these log fold change? log2? Perhaps it would be more informative to plot normalized expression values. Error measures, number of replicates and individual sample values are lacking.
36. Fig 2d lacks color scale. Pten points to dot with little naive TF change while main text (p111194) claims big delay.
37. Fig 2d, S2e: I don't get the apparent absence of any change in the Jmjd1c KO at 2i or N24. Discrepancy between these figures.
38. Fig 4a: what is the purpose of this figure? What do the colors mean?
39. What is Fraction of identity and why is it the right tool to compare RNA-seq with scSeq embryo data?
40. Fig S4f-i: WT cells missing.
41. Fig S5a is not referenced anywhere.
42. Fig S6a,b,e: N24 instead of N2.
43. Table S5: what is the NAGs column and the values in it?
44. p211427: wrong fig reference.
45. Some apparently incorrect references, e.g. 51
46. p211429: why is that intriguing? Explain the relevance of these observations.
47. Fig 7B: many gene interactions indicated that are not supported by evidence, e.g. Fgf5-Dnmt3b. What is the basis for these diagrams? Authors should limit themselves to actual evidence.
48. p451950 xxul.

RE: EMBOJ-2020-105772

"Cooperative genetic networks drive a mammalian cell state transition"

We are pleased to submit a revised version of our manuscript in which we provide additional data and analyses to address the key comments and suggestions of the three reviewers. We note that all three reviewers provide an overall positive assessment of our manuscript and are looking forward to receiving their feedback on our revised manuscript. We have made the following major changes in the manuscript:

- 1) We have further developed our analysis to identify potential functionality of identified clusters in Figure 6. To this end, we have identified those clusters (differentiation gene-modules) that show the strongest correlation with the strength of phenotype (strong cluster regulation in KOs with strong phenotypes) and have further refined our modelling approach to link clusters to signalling pathway activities. To illustrate these findings, we further provide a set of heatmaps showing expression of a set of strongly phenotype associated clusters (N24 induced and constitutive) and selected subsets of genes enriched in specific GO terms in the Appendix Fig S8-20.
- 2) We have restructured and improved data representation in Figures 6 and 7.
- 3) We have made a series of clarifications. For example, we have better described our strategy to ensure the genotype of the generated ES cells and have described the phenotype of the *Csnk1a1* KO ESCs in more detail.
- 4) We have furthermore addressed all reviewers' questions with new and additional data where required.
- 5) During the additional quality control steps, we have discovered that one WT sample in the analysis was wrongly assigned to the *Kdm6a* KO during data processing. To repair this issue, we had to replace the sample with an existing correct one. Because the existing RNAseq sample was available, but not part of the original normalization and batch correction, we had to renormalize, and batch correct all data. This necessitated recalculation of all results in the manuscript with the newest versions of all analysis packages. Although the fact that normalization is based on all samples in the analysis caused minimal changes in the basic data, no conclusions are affected. This is visible in the data presented and can be illustrated in a PCA plot showing the Top5 KOs based on phenotype strength, the old and new *Kdm6a* KOs and a set of 9 randomly selected KOs (using automated randomization). Only minimal differences could be detected (see plot below). As expected the only appreciable difference can be observed for one replicate of *Kdm6a*.

PCA of KOs in old and new analysis

Figure: PCA, comparing batch corrected individual RNAseq replicates before (filled shapes) and after renormalization after replacing one Kdm6a replicate (outlined shapes). KO's are indicated by different colours. RC9 WT cells are black. 2i samples are shown as circles; N24 samples as triangles. Note: Apparently filled and outlined shapes are resulting from near exact overlaps between 'old' and 'new' data.

In sum, we have addressed all reviewers' comments and made important improvements to our manuscript. We trust that you and the reviewers will consider this revised manuscript suitable for publication as a Research Article in the EMBO Journal.

Best regards,

Martin Leeb
(for all authors)

We thank all three reviewers for their comprehensive, fair and constructive evaluation of our manuscript and provide the following point to point response:

Referee #1

In this extensive study, the authors assess the regulatory genetic networks that govern the cell state transition from naïve to formative state of pluripotency in mouse embryonic stem cells (mESCs). By performing genome-wide saturated loss-of-function screens in haploid mESCs, they reveal dozens of genes that exhibit a delay phenotype during this cell fate change, when mutated. They focus on and further analyze 73 such genes, generating KO cell lines and performing RNAseq analyses during their transition from naïve to formative pluripotency. Using the transcriptome datasets they generated, along with several previously published ones, they classify these 73 regulator genes according to: 1) their regulation of naïve- and formative pluripotency-associated genes during naïve-to-formative pluripotency transition, 2) the stage of normal development that their mutants are delayed at, 3) the resemblance of the transcriptomes of their mutants to the in vivo developmental stages in mouse and primate peri-implantation development, 4) their regulation of 5 pluripotency-associated signaling pathways and the transcriptional response gene networks downstream of these pathways. They also utilize the transcriptome data of the mutant mESC lines to define an extended naïve pluripotency network that shows a high similarity between the in vitro and in vivo epiblast transitions. This work addresses an interesting question in cell biology regarding cell fate changes during early development, in an unbiased way, and using a robust cellular model. Therefore, the manuscript is suitable for publication either as a Research Article or as a Resource manuscript.

R: We thank the reviewer for his positive assessment and for noting that our manuscript is suitable for publication.

Enclosed are a few minor points that should be addressed to improve the manuscript:

1) In Supplementary Figure 1c, the authors show enrichment of a number of GO terms related to differentiation of specific tissues. It would be nice if they could discuss it shortly to explain the overlap of genes related to specific tissue differentiation pathways and the naïve-to-formative pluripotency transition.

R: Our GO and pathway analysis of screen hits indeed showed, among functions pertaining to signalling, chromatin organisation and general stem cell biology related terms, an enrichment for some tissue specific differentiation pathways (e.g. cerebellum morphogenesis [detected hits in category: *Usp9x*, *Dab1*, *Abl1*, *Gli2*, *Abl2*, *Ulk1*, *Ptpn11*, *Hspa5*, *Ldb1*], hindbrain development [*Trp53*, *Usp9x*, *Rere*, *Dlc1*, *Kdm2b*, *Dab1*, *Abl1*, *Gli2*, *Otx2*, *Pfdn1*, *Rnf7*, *Abl2*, *Ulk1*, *Ptpn11*, *Ctnnb1*, *Hspa5*, *Ldb1*] or skin development [*Tcf7l1*, *Fgfr2*, *Rbpj*, *Apc*, *Arrdc3*, *Rock2*, *Mreg*, *Ncoa3*, *Flnb*, *Sos1*, *Grhl3*, *Cbx7*, *Pdgfa*, *Clic4*, *Itga6*, *Ctnnb1*, *Lama5*, *Ldb1*, *Ngfr*]). However, we do not believe that these data indicate that there is indeed e.g. a skin specific or hindbrain specific group of genes required for the exit from naïve pluripotency. Closer inspection of the genes enriched in each term shows that multiple genes enriched for these GO terms have a rather general function (e.g. *Usp9x*, *Gli2*, *Ulk1*, *Ptpn11*, *Trp53*, *Kdm2b*, *Ctnnb1*, *Tcf7l1*, *Fgfr2*, *Rbpj*, *Apc*, *Sos1*, *Grhl3*, *Cbx7*, *Pdgfa*) and play roles in e.g. signalling, general germ layer specification and chromatin regulation. Identification as having a

role in tissue specific functions appears a function of the role of the identified genes in multiple differentiation steps.

We therefore interpret these data in a way that suggests an enrichment for screen candidates for multiple signalling and chromatin related functions. These are harnessed for the exit from naive pluripotency but may also be used for later differentiation processes. Although this could be interesting to explore in more detail, we propose to limit our statement regarding GO analysis of screen hits to:

"Many identified genes are not specific to pluripotency and have functions in common pathways and processes, implying that the exit from naïve pluripotency utilizes widely expressed cellular machinery. Therefore, mechanisms mediating ESC transition might also be utilised in other differentiation processes."

2) In Supplementary Figure 1f, the authors list the Csnk1a1 KO as one of the KOs with full protein deficiency, although a residual band is visible in the KO lane. Given their observation on this mutant regaining its differentiation potential after several passages, the authors should clarify whether this reversal of phenotype is due to residual protein in their mutant line.

R: We agree with the reviewer that the Csnk1a1 Western shown was not absolutely clear. The antibody we previously used in EV Fig 1F showed several background bands and only gave weak specific signals. We have now purchased a new antibody (Rabbit polyclonal Anti-Csnk1a1, Novus Biologicals, NBP1-18880) and were able to generate easier to interpret Western data (now shown in EV Fig 1F). Furthermore, to ascertain the proper genotype the Csnk1a1 KO (as well as all other KO ESCs) was frequently genotyped by PCR confirming homozygous deletion, with identical results before and after loss of the phenotype.

3) There are several mistakes in the numbering of the Figures in the main text or figure legends. (e.g., Figure S1a referral in P9, line 138; Figure 6d referral in P22, line 447; missing legend of Supplementary Figure 4i) The authors should make sure that the right figures are referred to throughout the text. Similarly, some Figure panels are not mentioned in the text (e.g., Supplementary Figure 4c and Supplementary Figure 5a).

R: We thank the reviewer for these corrections and apologize for these mistakes. We have now updated all Figure references which will make the manuscript better accessible and easier to read. WE have further split Additional data in Expanded view figures and Tables and an Appendix as of EMBO Journal manuscript guidelines.

Referee #2

Summary:

The authors began with a mutagenesis screen to identify factors that prevent rex1-egfp downregulation following 2i withdrawal, which normally induces state transition. Following that, they made a number of CRISPR mutant ESC lines [Fig 1] for a more in-depth investigation of these factors regulating state transition. They investigated how these CRISPR mutations affect expression of specific ESC marker genes, as well as global transcriptome [Fig 2]. They also studied these CRISPR mutants using a differentiation time-course to quantify differentiation/transition delay [Fig 3]. They showed that the state transition regulatory program is largely preserved in these CRISPR mutant lines [Fig 4]. They proposed to extend the list of naive pluripotency associated genes from an original core 7 to 300+ NAGs [Fig 5, 6]. Finally, they defined a list of key (but not exhaustive) 5 signaling pathways regulating the naïve to formative ESC state transition [Fig 7].

Comments:

1. The original screen was performed in haploid ESCs, but all subsequent follow-up work (CRISPR mutants generation & characterization) seems to have been done on diploid ESCs. Reasons/rationales for this discrepancy unclear, which assays used which type of cell and why?

R: It is correct that all experiments apart from the initial screen were performed in diploid, biparental ES cells. Despite the increased genetic complexity of diploid ESCs, we chose to do so to exclude any potential confounding effects introduced by using haploid ESCs in the initial genetic screens. The verification and detailed transcriptional analysis in biparental diploid cells allows drawing conclusions without any potential confounding impact by the haploid or parthenogenetic background. However, the high validation rate of the haploid screening results in diploid cells suggests that there is little systematic error introduced by using haploid ES cells.

2. While a good effort was made to generate 73 ESC KO lines, it seems that validation of absence of protein (or even RNA) was only attempted for one quarter of these. Additionally, it is also unclear whether CRISPR disruption resulted in homozygous or heterozygous KO-one would normally assume homozygous, but protein still present (at reduced level) for at least one tested gene Mapk1 [Sup Fig 1]. Please clarify/ show evidence.

R: The point of cell line validation is a very important one. We have taken several steps to ensure the reliability of our KOs, to the extent possible in a large-scale study. We believe these steps were not made sufficiently transparent in the initial submission. We have therefore now made these steps clearer in the manuscript and in the methods. In brief: All KOs were initially identified by PCR on genomic DNA, testing for either a deletion of at least one exon and the generation of an exon-intron fusion, or by identifying point mutations by Sanger sequencing. All KOs* in this study showed a homozygous phenotype (mutant band detectable, WT-amplicon or WT-sequence not detectable by PCR or Sanger sequencing, respectively). This strategy allowed us to unambiguously distinguish between homozygous and heterozygous clones. Genotype verification by PCR was again performed at the time of RNAseq analysis or soon before, after expansion of the respective clone to ensure reliability of RNAseq data. We have now made our strategy clearer in the text and methods. Large

deletions were also investigated and validated directly in the RNAseq datasets, but small indels cannot reliably be detected in mapped RNAseq data. Although it is not feasible (and in fact impossible due to AB availability) to investigate all clones on the protein level, we made a substantial effort to validate a large fraction of our KOs in secondary assays by Western and rescue experiments.

Further underscoring the validity of our datasets, mutants for components of related cellular processes clustered in t-sne or other clustering experiments (see Figure 1GH) and independent mutant clones clustered in replicate resolved dendrograms (now provided in Appendix Fig S3).

*Regarding Mapk1: For this KO, already initial genotyping revealed the heterozygous nature of this clone. Despite achieving multiple heterozygous clones for Mapk1 we were unable to obtain a single homozygous mutant ESC line, indicating a strong counterselection upon acute depletion of Erk2 protein or essentiality in our medium conditions [Note: Erk2 KO ESCs have been established, but cultured in serum/LIF medium, but Erk1/2 dKOs suffered from telomere shortening and growth arrest (Chen et al., 2015, PNAS, DOI: 10.1073/pnas.1516319112). The presence of 2i could therefore have possibly contributed to the stronger selection against homozygous mutants in our hands]. Regardless, the heterozygous clones showed a strong differentiation defect in Rex1GFP assays. Therefore we decided to include it in the analysis pipeline, while clearly stating that for this specific clone we knew that we were working with a heterozygous clone that showed reduced ERK2 levels.

3. CRISPR genome editing may cause other unintended off-targeted mutations on both small (few bp deletions) and large (chromosome rearrangements/ deletions) scales. What attempts have the authors made to check for/ rule out these?

R: We agree with the reviewer that there are potential off-target effects when using CRISPR/Cas9 (or any other nuclease-based genome engineering strategy). Our study is of a scale that no longer permits deep characterization of individual mutant cell lines, but we took steps to limit potential off-target effects: i) We designed gRNAs with low off-target scores. ii) For 44 out of 73 KOs, we have RNA-sequenced two independently derived clones, which will reduce the risk of off-target effects. iii) We performed rescue experiments to show causality of genetic deletions for the observed differentiation delays for a subset of mutants. We want to note that every single rescue experiment performed led to a reduction of the exit delay phenotype.

Only exome sequencing combined with CGH would provide the level of detail required to exclude any potential off-target effects that could overshadow the effect of the presumed mutation. Neither is feasible in the scale that would be required for our study.

4. Rescue-by-reintroduction experiments [Sup Fig 1G/H] are nice in theory, but what level is rescue protein expressed at relative to WT?

R: Rescue experiments were performed in pools of cells; therefore, we expect a range of expression levels in a heterogeneous population. We perform all transgenesis experiments in this study using transposons, which in our hands and conditions (100ng transposon, 500ng transposase, 200,000 cells, 3µl Lipofectamine 2000 in a reverse transfection protocol) results in relatively low levels of

transgene-expression. Regardless, the majority of 'rescued' KO cells regained differentiation potential, indicating that, at least for the tested factors, there are no strong dosage effects (at least for regaining the potential to exit naive pluripotency). Although rescue experiments were performed for a subset of genes only, every single rescue experiment was successful.

5. The authors repeatedly make the point that no single gene KO blocks exit from naive pluripotency -- except for Csnk1a1? This point needs more clarification.

R: We are sorry for being unclear on this point. We have further clarified the role of Csnk1a1 and portray additional data in Appendix Fig5 A-F.

Csnk1a1 mutant profiles assayed at N24 show a strong similarity to RNAseq profiles of Csnk1a1 mutants in 2i and naive marker downregulation is strongly impacted; at N24 Csnk1a1 are the only KOs that cluster together with 2i in PC1 on PCA analysis, despite showing substantial gene deregulation and deviation in PC2.

However, Csnk1a1 KO cells do initiate differentiation and are not locked in to the naive pluripotent state. We show near WT levels of RexGFP expression at N48 in data shown in Appendix Fig S5A. We furthermore want to point out that Csnk1a1 mutant ESCs show a pronounced proliferation defect (Appendix Fig S5F), making detailed mechanistic investigation very difficult and potentially prone to artefacts. This KO was the only one showing such a strong proliferation defect in our set of mutants and we decided to maintain it in the analysis as part of the b-catenin destruction complex and because it showed a strong phenotype.

Further complicating things, both the proliferation and the differentiation defect are lost upon passaging, suggesting the presence of compensatory and redundant pathways that can be activated upon Csnk1a1 loss. We have now also confirmed this effect in two further independent KO ESC lines and have made the (not replicated) observation that extended culture in EpiblastinA also restores differentiation potential measured by Rex1GFP downregulation, while acute addition of EpiblastinA results in a clear differentiation delay (Appendix Fig S5D).

We believe that the exceptionally strong differentiation delay observed at N24 in Csnk1a1 KO cells is, at least in part, due to the proliferation defect observed in these cells. We have now portrayed the Csnk1a1 KO in Appendix Fig S5A-F. Nevertheless, from a systems perspective, the Csnk1a1 KO provides a sample where strong differentiation defects can be observed. Therefore, we decided to transparently keep it in the analysis.

In response to the reviewer's comment we have rewritten the text to be more specific and clearer about the phenotype of Csnk1a1 KO cells and to avoid the impression that Csnk1a1 KOs are blocked in differentiation.

6. More data needs to be shown related to gene expression clustering of the CRISPR mutants, (for example visualization of clustering by media condition? Mutated gene? in addition to clustering by batch shown in Fig S1). The reviewer is aware that there is a web app, but this is critical data and should be in the paper too.

R: We agree that this is crucial data to show in the paper. We show t-SNE based clustering of mutants per condition in Appendix Fig S4A, before and after batch correction. In Fig 1GH we provide t-sne plots (separate plots for KOs in 2i and N24 to provide better resolution between KOs). These show the expected clustering of KOs deficient for components of the same pathway or complex (*Mta3-Mbd3*; *Tsc2-Pten*; *Smg5-Smg6*; *Fgf/ERK* components). We now provide further data in the Appendix Fig S3, showing two dendrogram (based on top1000 most variant genes and all DEGs of the N24 vs 2i in WT contrast) of all replicate-samples before merging of replicates. Overall, excellent clustering was observed.

7. In Fig 2A the authors provide evidence that their CRISPR KOs did not show reduced naive transcription factor expression - data is qPCR? Is this also corroborated by protein expression/other functional assays?

8. In addition to rex1-gfp expression the authors also quantify differentiation/transition delay by expression of a defined gene set & global transcriptomic profile [Fig 2]. It is promising that they correlate (Fig 2C), but can the authors demonstrate any other functional consequences of this differentiation/transition delay on ESC behavior/properties?

R: Figure 2A shows RNA-Seq data obtained at N24 for all KOs relative to WT cells. Transcription profiles are a good proxy for cell state. In our view, RNAseq data provides the most accurate readout of the exact cell state within the time-resolution required in our study. This enabled us to exactly quantify differentiation delays compared to WT cells for all KO samples based on the expression levels of a set of naive pluripotency marker genes at the N24 state in Revised Figure 3B.

In response to the reviewer's comment we have performed some additional experiments to ask whether the average naive marker expression level at N24 is a good proxy for delays in the downregulation of other known features of naive ES cells and functional commitment defects.

To this end we have obtained triplicate data measuring Rex1GFP levels at N24 and at N28 for 12 KO ESC lines that display various levels of differentiation delays based on RNAseq (REVFig 2B). Strong phenotypes are nearly completely Rex1GFP positive at N24. At N28, Rex1GFP expression is more variable, providing a better resolution for these KOs (these data also show that not even strong mutants are 'blocked' in differentiation. Overall RNAseq based measurements of naive marker genes and delays measured based on Rex1GFP downregulation defects are tightly correlated.

We furthermore performed triplicate commitment assays (Leeb et al., 2014) (REVFig 2CE) for the set of 12 cell lines. Our experiments show that Rex1-GFP regulation, commitment efficiency and the observed changes in average naive marker TF expression at N24 and are closely correlated for the set of KOs tested. Commitment assays are non-trivial and comparability is often confounded by cell line specific detachment issues during the long (~7-8 days) duration of the experiment. In this specific set of independently obtained experiments *Trim71*, *Ptpn11* and *Pten* mutants were lost from cultures due to strong detachment issues in 2i and in N2B27 medium. These three KOs show no aberrant growth in 2i and proliferate normally during early differentiation at N24.

In conclusion, we find that a delay in downregulating the naive network is strongly expected to result in functional differentiation defects, as suggested by our experiments. Furthermore, we want to point out that for several of the KOs in our study (e.g. Mbd3, Tcf7l1, Rbpj, Zfp281, Etv5, Jarid2, Eed, Suz12, Fbxw7, Kdm6a, Tsc2, Smg6) there exists evidence in the literature that establishes functional differentiation defects upon candidate gene depletion.

In conclusion, we are certain that for the purpose of our study a comprehensive assessment of the differentiation state by RNAseq is the ideal readout for cell state and differentiation delay measured by the average expression level of a set of naive transcription factors or by a more global assessment of how much of the gene expression changes that occurs in 2i was achieved by the mutants.

REVFig 1:

(A) Quantification of flow cytometry profiles of WT and KO in REX1-GFPd2 reporter line.

Quantifications are shown for N2i, N24, N28, N48 conditions. The gate for the REX1-GFPd2 signal is adjusted individually for each cell line so that 95 % of cells in 2i are GFP positive. Cells lines are ordered based on the strength of the differentiation delay observed at each timepoint. The red dashed line indicates the WT level. (N=3).

(B) Marker expression of the 11 closer investigated KO lines in the RNA-Seq data set. Log2FC vs WT for naive and formative marker genes are shown. Cells lines are ordered based on phenotypic strength.

(C) Schematic representation of commitment assay.

(D) Commitment assay quantification. (** no data available, * cell lines have attachment problems in the commitment assay and quantification may not represent real phenotypic strength). (N=3).

(E) Comparison of phenotypic strength of the 11 KO lines plus RC9-WT between Rex1-GFP based measurement by FACS at N28 and N24, colour code indicates average naive marker expression by RNA-seq. A good correlation between all three measurements can be observed (left). Comparison of phenotypic strength of the 11 KO lines plus RC9-WT between average naive marker expression and AP based measurement of commitment efficiency. R2 is shown in the panel. Overall increases in naive marker expression at N24 measured by RNAseq predict commitment defects (left). (** no data available for commitment assay, * cell lines have attachment problems in the commitment assay and quantification may not represent real phenotypic strength)

9. Similarly, for the differentiation/transition time-course defined in Fig 3, are there any other non-RNA-level based quantifiable features (e.g. Rex1 protein level?) that the authors can use as additional validation?

R: Using protein-based measurements would not provide sufficient resolution to detect fine grained differences in differentiation delays with the level of accuracy that RNAseq does. We have attempted time course proteomics ourselves and investigated existing resources (e.g. <http://www.stemcellatlas.org/>) but conclude that the required resolution to distinguish differentiation delays with precision is hard to achieve.

Rex1GFP reporter levels would unfortunately also not provide a resolution necessary for measuring strong differentiation delays, because Rex1GFP downregulation initiates only just before the N24 time point and until that time point very little quantifiable difference exists between KOs and WT cells (see also REVFig 2A). In contrast strong differences in the naïve mRNA expression can reliably be detected already after 2-4 hours of differentiation in WT cells.

Intrinsic validation for the 2h time-course itself comes from the PCA analysis where PC1 separates the time course RNA-seq profiles in the expected temporal order (2iL-2i-2h-4h-6h-...-30h-32h, Appendix Fig S6A). Furthermore, transcript levels of naive markers *Esrrb*, *Nanog* and *Tbx3*, and formative markers *Otx2*, *Dnmt3a* and *Pou3f1* follow the expected expression trajectories, showing the reliability of this dataset (REVFig 3).

Using entire transcriptomes and subsets of genes (e.g. naive TFs) is, in our opinion, the most accurate and statistically strongest method of measuring differentiation states and differentiation delays.

REVFig 2:

Representative plots of three naive and three formative marker genes, capturing known behavior of genes regulated during the exit from naive pluripotency.

10. In the final figures, the authors highlight many signaling pathways, and even more pluripotency-related genes/pathways, but the implications of these findings/ categorizations not made clear (it reads like a data dump).

a. For example, of the 5 signaling pathways, how do they differently affect pluripotency state transition?

R: We thank the reviewer for this suggestion and apologize for giving the impression of providing a data dump - this was obviously not our intention. We have rewritten this part to provide more context and a better explanation of our approach and description of our main findings.

Regarding the question how the 5 signalling pathways affect the pluripotency transition: The exit from naive pluripotency is under control of (at least) these 5 signalling pathways. The 5 pathways were selected based on genetic evidence from the haploid ES cell screen and the knowledge that the addition of LIF sustains self-renewal. Our data is compatible with a model in which signalling pathways control or are controlled by multiple exit factor genes. We find that virtually all KOs deregulate at least one of the signalling pathways at N24, whereas deregulation profiles are rather specific, and most KOs deregulate only one signalling pathway.

In Figure 7 we build a model to predict pathway to cluster wiring and discuss these findings now in more detail. As an example, we find that the Wnt/Tcf7l1 signalling axis is rather specifically linked to three gene modules (clusters) that contain a high proportion of downNAGs (e.g. several naive TFs). This underscored the role of Tcf7l1 in shutting down a naive pluripotency specific expression state.

Our data is consistent with a model in which gene modules are controlled by single or multiple signaling inputs. These inputs act in a cooperative manner to achieve a coordinated cell fate transition.

b. Similarly, of the 1800+ genes/12 clusters in the constitutive KO response group and 720 genes/12 clusters in the N24 group, which were the most important / most predictive of delay severity, etc?

R: We are very grateful to the reviewer for this suggestion. We have performed some additional analysis to address this interesting point. These data are now presented in Figure 6C and show that multiple clusters show a strong correlation with the phenotypic strength (e.g. they are deregulated specifically in KOs that show a strong differentiation delay). Overall these are also clusters that are enriched for NAGs. We have further added data to show heatmaps of KO based clustering of these 'phenotype-associated' clusters in the Appendix Fig S8-20, where we also show specific GO term associated genes from these clusters to visualize the behavior of the entire cluster vs the expression behavior of specific subgroups of genes. We believe that this analysis has strengthened our conclusions and makes the manuscript more accessible.

Referee #3:

Summary:

The manuscript of Lackner et al aims to identify the genetic networks that guide the transition between two pluripotent states in culture, from the naive pluripotent state of mouse embryonic stem cells (ESCs) to the formative pluripotent state these cells differentiate into upon withdrawal of ESC self-renewal factors. The authors perform a transposon-based mutagenesis screen in haploid ESCs similar to one they performed earlier (Leeb et al 2014, Cell stem cell 14), but now driven to saturation. They identify 489 candidate genes whose mutagenesis causes a delay in downregulation of the naive marker Rex1 in differentiation conditions. For 73 selected genes, they construct diploid KO ESC lines, and determine their transcriptomes after 24 hrs of differentiation. They correlate these data with a fine-grained time-course ESC differentiation sequence to determine the extent of differentiation delay, group the mutants into categories, and link them to signalling pathways and common gene modules.

The authors have generated a massive and valuable dataset, and performed several ingenious analyses. Nonetheless, the conclusions are somewhat underwhelming and we learn relatively little of note about pluripotency transitions.

R: We thank the reviewer for appreciating our "ingenious analyses" and our "valuable dataset". We are convinced that our response will highlight the fact that we make a number of important findings of relevance for pluripotency and cell fate research. We want to thank the reviewer for the in-depth and helpful comments on our manuscript; they have helped to significantly strengthen our manuscript.

During the presentation of the results the authors discuss various more or less interesting observations, such as a link between *klf2* and *tfc7l1*. However, despite this link having been noted before, they don't perform in depth analyses of these observations.

The conclusions are very broad and general and it is often not explained how they are supported by the data. This already starts with the title: "Cooperative genetic networks drive a mammalian cell state transition": nowhere in the manuscript is cooperation between different genetic networks in driving differentiation investigated. Or: "Our data support a cell fate transition model in which extinction of cell identity induced by the interplay of signalling and the chromatin modification machinery precedes the initiation of a novel cell fate." No criteria or assays to determine extinction of cell identity or initiation of a novel cell fate are presented so this is in fact an opinion rather than a conclusion. This lack of insightful conclusions is surprising because interesting questions can be addressed using the newly generated data. It seems to me that the authors adhere to a pre-conceived model of the naive-formative transition, or exit from naive pluripotency as they frequently describe it, as a single monolithic event and simply state without any reasoning that all their observations support this model. Thus, they conclude that signalling pathways and gene modules are all harnessed in parallel to create a vastly complex 'exit machinery' that breaks down the naive pluripotency network although their data fits much better with a model in which the pathways/modules act sequentially and/or independently, during different phases of the transition.

There is indeed plenty of evidence that there are multiple steps and cell biological functions involved in the naive-formative transition, including recent evidence for an intermediate 'rosette' pluripotent state controlled by the sequential action of Wnt and Mek signals during the naive-formative transition (Neagu et al 2020 *Nature cell biology* 22, 534), the induction of transcription factors that rewire enhancer networks (e.g. Matsuda et al 2017 *Development* 144, 1948; Yang et al 2014, *Cell reports* 7, 1968; Buecker et al 2014 *Cell stem cell* 14, 838), adjustments in adhesive and morphologic properties (Veluscek et al 2016 *Stem cells* 34, 1213; Kalkan et al 2017 *Development* 144, 121), and stepwise action of signalling pathways and transcription factors (the earlier mentioned Neagu et al; Kalkan et al 2019 *Cell stem cell* 24, 785; Nett et al 2018 *Embo reports* 19, e45642).

The authors should try to find evidence for these steps and functions in their data, and find gene (modules) that operate them. For example, is there a category of KOs that predominantly affects adhesive/morphological properties? Are there categories predominantly associated with the naive-rosette or rosette-formative transition? Can a sequence of events be generated that occurs during the naive-formative transition? By breaking up the transition into discrete events, such analyses would do much more to unravel and assign functions to the gene networks guiding the naive-formative transitions. This study could then constitute an important contribution to our understanding of pluripotency. In its current form, however, it is mostly a collection of data that is of potential interest and in need of further analysis.

Major concerns:

1. As described above, the authors should try to find evidence for a sequence of distinct events or states in the naive-formative transition, and find gene (modules) that operate them. This could then be incorporated into fig 7b, which is not very informative as it is now.

R: We agree with the reviewer, that this is a very important and interesting question. To address the reviewer's comment we performed some additional analysis to better explain and describe data in Figures 6 and 7:

1. We computed pathway activities over the WT time course, which shows that pathways show sequential changes, exactly as postulated by the reviewer. Interestingly we can detect in our dataset a sequential activation pattern of the Wnt/Tcf7l1 (activated first) and the Fgf/Erk cascade (activated later), in line with finding from the ten Berge lab (Neagu et al., NCB, 2020), who found a sequential activity of Wnt and Fgf/ERK during naive to formative transition in a rosette formation model. We report and discuss this finding now more prominently in Figure 7.

2. We connect pathways to clusters by a modelling approach and identify clusters (constitutive and N24 induced) likely regulated by signaling pathways.

3. We have better annotated the clusters and for example identified those clusters that most closely correlate with the phenotypic strength across KOs and those that are most significantly enriched for NAGs. We have also improved the results section describing functional enrichment within clusters (Figure 6)

As to whether sequential activity of gene modules can be comprehensively determined and whether sequential cell fate transition transitions can be identified and separated into (maybe even partially independent) discrete cell states remains an open question that will have to be addressed in future work. We feel that thoroughly solving that question would require substantial additional work (including experimental validation) that is beyond the scope of this publication.

We also want to refer to point 17, regarding our definition of parallel vs co-operative activity, which we believe has caused some misunderstanding.

2. In the introduction the authors state as their goal to explore regulatory principles of the exit from naive pluripotency. What principles have been identified?

R: As suggested by the EMBO Journal, we will summarize our key findings in a synopsis, together with a graphical abstract:

Synopsis:

Informed by a haploid transposon screen we performed transcriptional analysis of differentiation defective ES cells to identify molecular ground rules of cell fate transitions in a systems biology approach. We identified discrete gene modules and an extended naive pluripotency network under control of cooperative gene activities and five key signalling pathways to control the exit from naive pluripotency.

Key findings:

- The exit from naive pluripotency utilizes multiple cooperative molecular processes gated mainly by five signalling inputs.
- Genetic depletion of exit genes in ESCs leads to increased molecular similarity to the in vivo naive epiblast
- NAGs define an extended network which is active also in vivo
- Klf2 is only weakly wired into the naive pluripotency GRN
- Cooperative and potential redundant function results in incomplete differentiation blockages in exit factor KOs.

3. In general, number of replicates and statistical measures are not indicated in many figures.

R: All Figure legends and methods now contain the statistical information required.

4. What state is N24? Earlier studies show that N24 cells contain many Rex1+ cells that are not formative. Why was this time point chosen? What was done to accommodate the heterogeneity of the cell population?

R: At N24 a proportion of cells retains naive pluripotent characteristics (high Rex1-GFP expression and the ability to continue self-renewal in 2i) and another proportion of cells has initiated formative identity. This makes it the ideal time point because the mechanisms that are employed for formative commitment must be active and their molecular footprint detectable at this time point. Therefore, we work under the hypothesis that at this point in differentiation their function or failure can be detected best. Our RNAseq data shows that we can detect a continuum of exit-defects at the N24 time point. Importantly we do not only have the N24 time point for each KO, but also analyzed KOs in 2i, therefore we can detect dynamic changes from 2i to an early differentiation time point for each KO and compare it to expression changes observed in WT cells. Heterogeneity in cultures is a point that, frankly, cannot be comprehensively addressed in bulk RNAseq analysis.

5. From 26 genes the deletion was verified by western or rtPCR/sequencing or rescue; how about the other 57 genes? How were the 73 KO genes selected?

R: The point of cell line validation is a very important one. We have taken several steps to ensure the reliability of our KOs, to the extent possible in a large-scale study. We believe these steps were not made sufficiently transparent in the initial submission. We have therefore now made these steps clearer in the manuscript and in the methods. In brief: All KOs were initially identified by PCR on genomic DNA, testing for either a deletion of at least one exon and the generation of an exon-intron fusion, or by identifying point mutations by Sanger sequencing. All KOs* in this study showed a homozygous phenotype (mutant band detectable, WT band or sequence not detectable). This strategy allowed us to clearly distinguish between homozygous and heterozygous clones. Genotype verification by PCR was again performed at the time of RNAseq analysis or soon before, after expansion of the respective clone to ensure reliability of RNAseq data. We have now made our strategy clearer in the text and methods. Large deletions were also investigated and validated directly in the RNAseq datasets, but small indels cannot reliably be detected in mapped RNAseq data. Although it is not feasible to investigate all clones on the protein level, we made an effort to validate a substantial fraction of our KOs in secondary assays by Western and rescue experiments.

Further underscoring the validity of our datasets, mutants for components of related cellular processes clustered in t-sne or other clustering experiments (see Figure 1GH) and independent mutant clones clustered in replicate resolved dendrograms (Appendix Fig S3).

6. p7 line 92-98: Most of the genes identified in the screens did not change between 2i and N24. The authors conclude from this that ESCs are poised for rapid decommissioning of naive identity, but the reasoning behind this is not explained. One thing that should be discussed from this is whether the idea that induction of an 'exit machinery' by differentiation signals is required to exit from the ground state is correct. It further seems to me that genes that are necessary for (part of) the naive-formative transition would be present in naive cells and only downregulated after the process in which the gene is involved is completed. That there is no change in expression at N24 suggests that the transition is not yet complete, so not particularly rapid. Alternatively, and referring to point 4, the cell population is heterogeneous and part of the cells have not yet switched to formative, maintaining expression of the identified genes. Either way, the data do not support the author's conclusion.

R: At this point we need to respectfully disagree with the reviewer. The reviewer proposes a plausible model for regulating drivers and regulators for cell fate change, which is in fact completely compatible with the one we are proposing. ESCs can be wired for rapid naive-identity decommissioning already in 2i, while the factors that achieve this transition can be downregulated after having performed their function.

Our model is based on extensive analysis and supported by our data. To address this, we have now analyzed expression changes of exit screen candidate genes in more detail in revised Fig 1F and Appendix Fig S4C. In conclusion we find that most of the 489 genes identified in the haploid screens including the 73 KO genes are relatively highly expressed already in 2i; they did not change significantly in expression between 2i and N24 and were therefore not present in the list of DEGs (shown in revised Fig 1F and Appendix Fig S4C). Only 21% of screen hits showing differential expression at N24 (6% up-, 15% downregulated). This is in support of our hypothesis that the exit machinery is already embedded and expressed in the ground state, and that ESCs are poised for rapid decommissioning of naive identity and entry into differentiation. However, the downregulation of approximately 15% of genes around the time of commitment to differentiation could indicate that some exit-regulators are shut down (or at least downregulated) transcriptionally after completing their role in the exit from pluripotency, according to the reviewer's proposed model. In general, though we conclude that in support of our proposed mode of activity, exit factors are relatively highly expressed in 2i and expression levels remain largely constant during early differentiation. We have further addressed this question on the protein level using Western blotting (REVFig 6) using KO-factor specific antibodies. These data show that also on the protein level the large majority of tested factors retained similar expression levels during a 48h differentiation time course (we tested all factors for which we obtained antibodies, no further pre-selection was made and all results are shown). However, there are a few exceptions to this generalization: Pten and Cdk8 clearly increase expression levels, whereas Jarid2, Smg7 and Tcf7l1 are downregulated at N48 after an apparent transient increase at N24. Taken together exit factors show a range of expression behaviors, but the

largest proportion of exit factors is expressed in 2i and expression levels change little during the exit from naive pluripotency.

In further support of our model, depletion of exit factors increases the similarity of ESCs to the naive epiblast at E4.5 (Fig 4), giving rise to the hypothesis that the exit-factors identified in the haploid ESC screen are also contributing to some of the differences between the naive epiblast and ESCs in culture. This implies and gives rise to our hypothesis that exit genes are not only expressed in ESCs but are active in ESCs, whilst being constrained in activity by 2i.

REVFig 3 - Western blots showing expression of indicated proteins in 2i, 2i-LIF, at N24 and at N48 utilizing specific antibodies.

7. p9 l146: The authors argue that the lack of effect of 'exit factors' on naive factors in 2i indicates the neutralisation of 'differentiation drivers' in 2i. These differentiation drivers would have been hit in the screen. How do the authors define exit factors and differentiation drivers, and which of their hits belong to each of these categories?

R: We thank the referee for pointing out our misleading use of the word differentiation driver. We have no evidence at this point that the factors identified in the screen are 'differentiation drivers'. This would require an induction of differentiation phenotype upon overexpression in 2i, or at least increased speed of differentiation after OE and 2i withdrawal. One screen hit for which such a role has been shown is Otx2 (Buecker et al., 2014), we have no evidence that other factors have a similar role. We have removed the term 'differentiation driver' from the text and replaced it by 'differentiation regulator'.

How do they think neutralisation of differentiation drivers occurs, and is there any direct evidence for this?

R: The evidence behind this statement is that despite the presence of the largest part of the exit machinery in 2i, ESCs can be maintained in 2i and differentiation only initiates (very rapidly) upon 2i withdrawal. However, our results on the similarity to in vivo differentiation indicate that there is some activity of at least a set of exit factors already in 2i, leading to increased similarities of exit factor KO ESCs to the in vivo E4.5 epiblast. Further evidence supporting our statement comes from the fact that a set of exit-factor KOs reduce basal levels of formative gene expression (EV Fig 1B).

8. I applaud the authors for creating a webtool to access the transcriptome data but I regret to say that I was unable to figure out how to use it or what I could do with it.

R: We are very grateful for the feedback and are sorry to have used the reviewer as beta tester. We have completely overhauled the app and much simplified it. We have further added functions to map custom gene lists and show i) behavior of these genes in 2i and at N24 in all KOs. ii) expression behavior of these genes in the 32h TC as heatmap. iii) individual plots of expression kinetics for each gene, which, we believe, will be very helpful for our colleagues. We have also added more instructions to the documentation. We believe that the app now has all the functionality that will make it a great tool for the community.

9. Fig S2c: this figure is supposed to show increased rex1 downregulation following Klf2 depletion, but it could well be argued that it shows the opposite.

R We agree with the reviewer that the Figure did not optimally report the FACS results. Such shifts in the Rex1GFP negative population are often of technical nature. We have chosen another dataset of a replicate experiment to clearly show Rex1GFP downregulation after Klf2 KD, now shown in Fig EV1C.

10. Fig 2b: It is claimed that the differentiation delay of Tcf7l1 KO ESCs is dependent on Klf2 because Rex1 downregulation speeds up upon Klf2 KD. However, the figure doesn't show the effect of Klf2 KD on Rex1 in Tcf7l1 KO ESCs in 2i. Klf2 KD could cause downregulation of Rex1 without affecting differentiation speed. Also, the interaction between Tcf7l1 and Klf2 was already reported in Leeb et al (2014) CSC 14, 385 and in Martello et al (2012) CSC 11, 491; which novel insight is gained with the data here?

R: In Figure 2A and B we illustrate how our data can be used to detect that different KO factors can 'attack' the naive TF network at different places. The genetic interaction between *Tcf7l1* and *Klf2* is well established, as the reviewer correctly notes, and we cite the study making the original finding. Although naive TF factors are tightly linked in terms of expression, we can identify this interaction as outstanding even in a large KO repository. This however is not the main finding of this experiment.

The main experimental result in this Figure is that Klf2 KD cannot rescue the *Jarid2* KO dependent differentiation delays. A *Klf2* independent phenotype is predicted by our data and *Jarid2* KO is the only KO showing a very strong differentiation defect without showing an increase in *Klf2* levels. *Klf2* KD in *Tcf7l1* KOs was performed as control in this Figure. Data in this Figure constitutes also one layer of evidence for a less tight linkage of *Klf2* to the naive network, compared to the rest of the

naive core GRN. This finding is later corroborated by the fact that *Klf2* is not identified as NAG. We discuss possible evolutionary reasons for the less tight coupling of *Klf2* to the naive network in the discussion section. The special role of *Klf2* in the naive network is one of the main findings of our study.

11. p12I212: stronger than expected effect of CHIR on *Jmjd1c* KO, suggesting cooperation between *Jmjd1c* and *Tcf7l1*. Could the authors elaborate on how this could work? Perhaps *Tcf7l1* relies to some extent on *Jmjd1c* to repress target genes? Is there any evidence for this?

R: The reviewer raises the interesting question regarding the molecular functions of *Tcf7l1* and *Jmjd1c* and convergence thereof that lead to a synergistic interaction between these two factors. The role of *Tcf7l1* in the exit from naive pluripotency is well established. Nothing is known however about the function of *Jmjd1c* during the naive to formative transition. It is not even clear what exactly is the molecular role of *Jmjd1c* (supposed H3K9me3 demethylase). Gaining mechanistic insights into the interplay of these two proteins is the focus of an ongoing study in the Leeb lab and beyond the scope of the present manuscript. However, we want to note that a physical association

between *Tcf7l1* and *Jmjd1c* has recently been shown, which we cite in our manuscript.

A

B

REVFig 4:

(A) Quantification of cell number in WT and KO lines after 30h in plain N2B27 or N2B27 + Chiron.

2i samples serve as control. Measurements are normalized to the 2i mean (N=4).

(B) Flow cytometry profiles of WT and KOs in REX1-GFPd2 reporter line after a live/death-staining with Draq7. Profiles are shown in N2i and N2B27 or N2B27 + Chiron at the 30h timepoint. The Draq7 gate is set so that WT cells in 2i are Draq7 negative. The gate for the REX1-GFPd2 signal is adjusted individually

for each cell line so that 95 % of cells in 2i are GFP positive.

12. Fig 2e: CHIR delays differentiation of *jmjd1c*KO much stronger than that of WT. However, ESCs in CHIR without other factors for 48 hrs may experience a lot of cell death, and it may be preferentially the naive cells that die. The effect of the *Jmjd1c* KO (and of other KOs) might therefore be predominantly on survival rather than differentiation. Have the authors considered this effect? Can they account for this by quantifying numbers of input and output cells to determine the extent of cell loss, or demonstrate that cell death/survival plays no role?

R: This is an interesting question that we have now assessed in a quantitative manner for the specific case of *Jmjd1c/Tcf7l1*. We observed a small increase in cell numbers in *Tcf7l1* and *Jmjd1c* KO ESCs in N2B27 and in CHIRON alone, but without a significant level of cell death associated with any of the conditions. This is illustrated in REVFig 7AB by very low DRAQ7 levels, indicating that virtually all cells at this time in the analysis were viable, regardless of the Rex1GFP status at N30. We conclude that viability plays little role in 'generating' apparent exit defects by selective killing of differentiating/differentiated cells.

13. p121219: " We then compared the expression patterns of specific gene-sets in a given KO at N24 to the expression of the same gene-set along the WT 2i to N32 trajectory." Which specific gene-sets were used? How were they selected? Are they the same for each KO?

R: We are sorry for not making this experiment properly clear. Selected gene groups are the same for each KO. We have rewritten this part of the manuscript to avoid misunderstandings. In Fig 3B we have used the expression level of a set of naive markers during the 32h differentiation time course to anchor all KO samples to a specific timepoint in differentiation that corresponds to the expression levels of these naive markers best. Thereby we could accurately measure the timing of differentiation delays. In Appendix Fig S6B we have utilized all DEGs (WT N24 vs 2i) to assess the differentiation delay using this gene group.

14. Fig 4b-e, S4a: WT cells are not indicated in many panels. In contrast to what is stated in the main text, the N24 samples do not seem to show higher FOI to E5.5 than to E4.5. Seems we should look to 4c-e but these figures are not clear, one cannot identify how each KO line shifts from 2i to N24. Axes are different, which makes any comparisons even harder. Even not all KO lines that are mentioned in text are labelled in figure.

R: We thank the reviewer for pointing this out and agree with him that the data was hard to interpret without indicating the position of the WT. We have now updated Fig 4. It is now apparent that WT samples at N24 are more similar to the E5.5 and WT samples in 2i more similar to the E4.5 epiblast. Similarities in mutant ESCs vary, largely in accordance with the exhibited differentiation delay, but also some specific patterns can be observed, which we discuss in the manuscript. We have also labelled all KOs that are mentioned in the text. Furthermore, we provide all FOIs to all KOs in Dataset EV3.

15. Fig 5a: Authors should compare top 374 instead of 250 differentially expressed genes with the 374 NAGs. Since NAGs are downregulated, the authors should compare them with the 374 most downregulated genes. What does the 1-0.25 at x-axis mean?

R: Association with the naive network can be achieved by either positive or negative correlation with the set of naive marker genes. Therefore, NAGs are not necessarily downregulated during differentiation. We have updated Figure 5 and now show that NAGs (we have separated them into up and down-NAGs for clarification) are partially overlapping but largely different in identity to the top differentially expressed genes in WT in vitro differentiation (Fig 5A).

We agree with the reviewer that it is better to compare the group of NAGs to a size-matched group of top-differentially expressed genes and are grateful to her/him for pointing this out. We have therefore again performed all analyses in Fig 5 by comparing the results obtained for NAGs with those obtained from using the size-matched group of top-diff genes. Our conclusions remain the same (see Fig 5).

We have changed the axis in this plot to show the complete set of Dim3 genes. The x axis shows all genes that belong to Dim3, ranked according to their level of contribution to this Dimension in the PCA.

16. p171321 and Fig 5b-d: " Strikingly, NAGs showed a near identical fold change behaviour in vitro and in vivo during the epiblast transition from E4.5 to E5.5." Superficially yes, but the behavior of the individual genes look quite different and rho-values have been omitted, suggesting that the fold change is only by accident similar and doesn't reflect conserved behavior.

R: We agree with the reviewer. This was an unnecessary overstatement. We have rephrased this statement to better reflect the actual data to: "Strikingly, NAGs showed a strongly correlated fold change behavior in vitro and in vivo during the epiblast transition from E4.5 to E5.5". This statement is supported by the rho-value shown in the plots.

17. p201394: "We conclude that signalling pathways are harnessed in parallel...". The data do not demonstrate this because they are perfectly compatible with a mechanism in which the signalling pathways act sequentially. See also the general summary.

R: We completely agree with the reviewer and have probably failed to communicate our model appropriately. Regarding the reviewer's general introduction, we want to make clear that we completely agree with him and do not envisage the exit from naïve pluripotency as a singular monolithic event. Our data are very clearly showing that there are multiple decision-making structures acting in cooperativity during the exit from naïve pluripotency. In our manuscript we identify key upstream players of this process and the gene networks regulated by these.

We believe that our use of the word 'parallel' has caused some major misunderstanding, as also evident from the Reviewer summary above. In our view parallel indicates independence from each other (no intersection), but not necessarily having the same starting or end coordinates. Therefore, no temporal meaning or hierarchy between processes was implied. We used parallel synonymously

for co-operative (and potentially partially redundant). Our data provide evidence for such a co-operative activity (e.g. deficiency of factors relevant for various distinct molecular processes are required for naïve exit; multiple gene-modules need to be regulated for successful exit; pathway activity deregulation between KOs shows little overlap) and is in strong support of such a model. We are thankful to the reviewer to point out the ambiguous and not entirely clear meaning of the term parallel. Therefore, to clarify and to avoid misunderstanding we have changed the wording to 'co-operative(ly)' instead of parallel throughout the manuscript.

18. p211416: how was clustering of the constitutive KO response genes performed? Why did it result in 12 clusters? Is there biological relevance to the 12 clusters or could there just as well have been 11, 14 or 21 clusters?

R: To determine cluster number we used Ward's method to build a cluster hierarchy and the elbow method to determine where to set the cutoff on cluster number. The latter is based on looking at the change of total within-cluster variance by adding more clusters at each k. The experimental approach to determine cluster number is described in the methods and illustrated in REVFig 8.

A

B

REVFig 5:

(A) relation of change of within-cluster variance (y-axis) and the number of clusters (x-axis) for constitutive clusters. The change of within-cluster variance between $k-1$ clusters and k clusters indicates the improvement in clustering when using one more cluster. The used number of clusters is indicated by the dashed vertical line. (B) same as in (A) but for induced clusters

Therefore, the choice of clusters is based on scientific evidence. However, we freely admit that there is an arbitrary component in the choice of cluster number (as there is for setting most cut-offs). It could just as well be justified to select 10 or 11 clusters, but the key messages would have remained the same.

19. p22l452: "In summary, constitutive and N24 induced clusters define crucial genetic modules that are co-regulated by multiple exit factors."; p23l465: "Together, this shows that at least 24 distinct transcription modules can be defined, whose expression is deregulated upon exit gene depletion and which are largely under the control of five key signalling pathways gating differentiation." Of course, that is how the clusters were defined. The cluster analyses are potentially interesting but the authors should take the analysis further and not stop at self-evident conclusions. Fig 6c,e show potential biological functions and it should be further investigated which of these functions could play a role in the naive-formative transition. These can then be correlated to the time course data and an outline generated for a functional sequence of events in the transition, as also described under 1.

R: As mentioned before, the number of clusters was not chosen at random but based on scientific evidence (see response to point 18). Therefore, the conclusion that a certain number of clusters could be identified is a valid result by itself and not a self-evident conclusion.

Unfortunately building a hierarchical map of differentiation regulated or regulating gene modules is a non-trivial task that cannot be achieved within the frame of a revision. However, we have restructured analysis in Figure 6 and 7 and have e.g. identified which clusters show the strongest deregulation in KOs showing the strongest differentiation delays. We have further extracted more information on the molecular identity of several strongly phenotype associated clusters. Thereby we identified those clusters that contain large parts of the naive and formative programme and also, based on this reviewer's comment, identified several strongly phenotype associated clusters that contain GO terms pertaining to cell-adhesion related functions. This indicates that proper regulation of cell adhesion functions is a key function that has to be maintained for proper exit from naive pluripotency.

20. p23l458: "Adding further evidence to the pathway to cluster relationships, we found that in most cases activity changes of the regulatory pathways preceded the expression changes of their putative target clusters over the 2h differentiation time course." I assume this is demonstrated in figs S7a-f, but the authors should present these figures and explain how they demonstrate this.

R: We agree with the reviewer that the modelling-based approach to identify potential pathway-cluster wirings required better explanation. We therefore have adjusted the text and added a flow chart (Fig EV5A) to better communicate our experimental modelling approach. The flow chart

provides a step by step description of the strategy to model a pathway-cluster connection-map. We have also updated the text and Fig EV5A-D to describe approach and findings better.

21. p251498: "Notably, however, we observed segregation in terms of pathway deregulation. Thus, KOs that induce e.g. a Tcf7l1-like profile were less likely to show a signature of another pathway. This suggests that ESC differentiation requires coordinated activity changes of independent pathways under control of separable genetic networks." It was recently demonstrated that WNT signals regulate a different step in the naive-formative transition than other pathways (Neagu et al 2020 Nature cell biology 22, 534). This argues that the pathways are segregated because they act during different phases of the transition. The authors should discuss their results in light of these earlier findings.

R: We thank the reviewer for the suggestion to further illustrate and highlight this finding. We have indeed seen that the Wnt/Tcf3 axis is activated prior to Fgf/ERK signalling during the exit from naive pluripotency in our analysis. We show these data now more prominently in revised Figure 7A and discuss them appropriately in the text.

22. p251502: "The observation that single gene depletion is not sufficient to prevent exit from the naïve state is consistent with this interpretation." The authors should discuss the possibility that a single KO is not sufficient to completely block a pathway, e.g. there is redundancy in Tcf factors, as well as in Mapk factors.

R: We agree with the reviewer that such redundancy could exist and discuss this possibility now in the text. We interpret our data, however, to rather suggest independent cooperative mechanisms that result in a retention of at least some differentiation ability even in the strongest KOs.

E.g. CHIRON that should have a broader activity than just KO of Tcf7l1 cannot block differentiation. Mek1/2 inhibition alone by PD03 is also not sufficient to block differentiation. This indicates that even complete blockage of a signaling pathway cannot block differentiation. Our proposed way to interpret cooperative activity is to establish a 'functional redundancy'; not specifically by redundant regulation of the same genes (e.g. two equally active complex members that can replace each other) or by participating in the same molecular cascade, but by cooperative regulation of the same process (potentially by molecularly distinct mechanisms). Such a setup is potentially more robust than redundancy achieved by two homologous proteins. We believe that both modes of operation to achieve functional redundancy exist and are compatible with our model.

Minor concerns:

23. First sentence of introduction: "Mouse embryonic stem cells (ESCs) can self-renew in defined conditions in a ground state of naïve pluripotency." Can the authors explain here what the salient differences are between ground state and naive pluripotency? Is there a non-ground state of naive pluripotency?

R: The reviewer correctly points out the phrase 'a ground state of naive pluripotency' as an unnecessary tautology. Although strictly speaking "naive pluripotent" is further defining the ground state. There is only one naive ground state, and all naive cells are in the ground state, but not all ground states are necessarily naive. I leave linguistic and semantic finesse to others and we will use

either 'ground state' or 'naive pluripotency' to describe the ESC state in 2i medium and before entering a formative state. We have rephrased the first sentence of the introduction to:
"Mouse embryonic stem cells (ESCs) can self-renew in defined conditions in a state of naive pluripotency (Smith, 2017)."

24. How was pathway analysis in Fig S1b performed?

R: A description of GO enrichment is stated in the methods. In the revised version we describe pathway analysis in more detail.

25. What do colors in bars of fig S1bc mean? Why different colors in the same bar?

R: The colors represent the FDR of individual genes within the GO terms pertaining to the statistical strength of this genes in the haploid screen. Each rectangle of the bar represents a single gene.

26. Fig S1f: ERK2 heterozygote looks like it upregulated ERK1.

R: We thank the reviewer for noting this. ERK1 is indeed upregulated upon heterozygous ERK2 deletion; but without being able to rescue the strong phenotype of the ERK2 heterozygous mutant. We added a corresponding sentence to describe this finding to the text.

27. Fig S1g: what is EV sample? Jarid2 is missing.

R: EV stands for "Empty Vector". The reviewer is correct, we missed adding a Jarid2 Western panel. We added a Western Blot panel to show the rescue of Jarid2.

28. Fig S1h: Why is the rescue for Etv5, Rbpj not complete?

R: The incomplete rescue of Etv5 and Rbpj might results from the fact that we analysed a pool of transfected cells for these rescues.

29. Fig 1e: legend says 30h, text/figure 24h differentiation.

R: we corrected this mistake in the revised version

30. Fig S1i: how is batch correction done?

R: Batch correction is described in the methods. Briefly, we used the `removeBatchEffect` function of limma to visualize the data and included the batches in the experiment design for calculations.

31. S1m: 2i sample is missing.

R: The comparison shown here is between treated and untreated WT cells. Showing the 2i condition would not have additional benefit.

32. S1n: Pum1 KO in 2i sample is missing.

R: The comparison shown here is between treated and untreated WT cells. showing the 2i condition would not have additional benefit. Below we show that Rex1GFP profiles in 2i are near identical between RC9 and Pum1 KOs.

RevFig 6: Rex1-GFP FACS showing WT and Pum1 KO cells in 2i.

33. p9 line 138: wrong reference S1a.

R: In the revised version the correct reference is Appendix Fig S5G

34. Combinations of six, seven or ten naive TFs are used in various analyses, e.g. to determine the developmental delay. What are the reasons to use different combinations for different analyses?
check

R: We understand that this was suboptimal and are grateful to the reviewer for pointing out this discrepancy. We have now changed all analyses to use 7 naive marker TFs throughout the manuscript, apart from Figure 2A where we show 10 known naive markers and a set of formative markers.

35. Fig S2a,b: are these log fold change? log2? Perhaps it would be more informative to plot normalized expression values. Error measures, number of replicates and individual sample values are lacking.

R: Here we plotted log₂ fold changes of RNAseq data and selected only those genes for which significance was ascertained based on adjusted p-values. To be clearer, we generated a new panel with color-coded bars that show the adjusted p-value in Fig EV1AB. All details are given in the figure legend.

36. Fig 2d lacks color scale. Pten points to dot with little naive TF change while main text (p111194) claims big delay.

R: A color scale is now present in Fig 2D. The Pten label is now associated with the correct KO. Pten KO cells show a clear differentiation delay, similar to Mbd3 and Cdk8 KOs.

37. Fig 2d, S2e: I don't get the apparent absence of any change in the *Jmjd1c* KO at 2i or N24. Discrepancy between these figures.

R: We agree that this might seem counterintuitive, but it is a matter of cutoffs. for Fig2 we use a cutoff of $FDR \leq 0.05$, testing against the 0-hypothesis that there is a change smaller than abs(1.5)-fold. No genes pass this cutoff in the analysis for the *Jmjd1c* KO.

In Figure EV1D, in contrast, we plot all genes that are DEGs in *Jmjd1c* (=0 genes) OR *Tcf7l1* (many 100s of genes). Thus, the plotted genes will be almost exclusively dictated by the *Tcf7l1* KO, which can be seen by the separation on the y-axis. Nevertheless, we observe a global deregulation of virtually all genes that are deregulated in *Tcf7l1* KOs also in *Jmjd1c* KOs, suggesting a shared set of target genes. The exact mode of operation remains unclear and is beyond the scope of this study.

38. Fig 4a: what is the purpose of this figure? What do the colors mean?

R: We agree that the Figure panel carried little important information and have deleted it in the revised Figure 4.

39. What is Fraction of identity and why is it the right tool to compare RNA-seq with scSeq embryo data?

R: FOI is a tool to assign a degree of similarity between a sample and a known reference (Gong and Szustakowski, 2013). We have previously used the used FOI to compare human ESCs grown in different conditions to the embryo (Stirparo et al., 2018, Development). Results were consistent with the Pearson correlation analysis, but FOI analysis also allows highlighting differences. FOI analysis utilizes the entire transcriptome without any preselection of gene sets for comparison.

The results of FOI are consistent with measurements of the molecular phenotype, and with an independent PCA analysis performed in Fig 4F. We therefore conclude that the FOI analysis is the appropriate tool for the intended task.

40. Fig S4f-i: WT cells missing.

R: We apologize for not properly labelling the WT in this Figure. We highlighted the WT cells as white squares (since there is no marker change) for better visibility.

41. Fig S5a is not referenced anywhere.

R: This Figure is now changed and was replaced by EV3A to show the expression behavior of NAGs during WT differentiation.

42. Fig S6a,b,e: N24 instead of N2.

R: Thank you for pointing this out. We changed this to N24

43. Table S5: what is the NAGs column and the values in it?

R: This column indicates the NAGs in the cluster genes. In the revised version we changed this column to TRUE (for the gene being a NAG) and FALSE.

44. p211427: wrong fig reference.

R: Corrected.

45. Some apparently incorrect references, e.g. 51

46. p211429: why is that intriguing? Explain the relevance of these observations.

R: This reference was correct - Enrichr was used to check ChIP enrichments. However, we believe that the discussion about Zscan4 and Trim28 targets was a distraction. We have therefore decided to delete the section on Zscan4 deregulation to improve the clarity of the manuscript; hence, the citation is no longer relevant.

47. Fig 7B: many gene interactions indicated that are not supported by evidence, e.g. Fgf5-Dnmt3b. What is the basis for these diagrams? Authors should limit themselves to actual evidence.

R: We thank the reviewer for pointing out this oversight. We have removed the connections between formative factors, in absence of known interactions and acknowledging that many formative factors are not TFs.

48. p451950 xxul.

R: Corrected.

Dear Martin,

Thank you for submitting your revised manuscript (EMBOJ-2020-105776R) to The EMBO Journal. My apologies for getting back to you with delay due to protracted reviewer input. Your amended study was sent back to the referees for re-evaluation, and we have received comments from referees #1 and #3, which I enclose below. Please note that while referee #2 was at this time not able to reassess the work, we have editorially evaluated your response to his/her concerns and found them to be convincingly addressed. As you will see, the other referees stated that their issues have been comprehensively resolved and are now broadly in favour of publication, pending minor revision.

Thus, we are pleased to inform you that your manuscript has been accepted in principle for publication in The EMBO Journal.

Please consider the remaining minor issues stated by the referees carefully and address them by adding complementary experimentation or introducing caveats and discussion of the findings where appropriate.

In addition, we need you to take care of a number of points related to formatting and data representation as detailed below, which should be addressed at re-submission.

Further, I will share additional changes and comments from our production team during the next days to be considered.

Please contact me at any time if you have additional questions related to below points.

Thank you for giving us the chance to consider your manuscript for The EMBO Journal. I look forward to your final revision.

Again, please contact me at any time if you need any help or have further questions.

Kind regards,

Daniel

>> Add the title 'Conflict of Interest' to your competing interests statement.

>> Limit the number of keywords to maximally five.

- >> Release the privacy from your GEO data set.
- >> Please move the Material and Methods part up after the Discussion.
- >> Recheck if the two bioRxiv entries were published on the meantime and update the references in case.
- >> Provide a separate 'Statistical Analysis' section detailing the algorithms applied.
- >> Adjust the reference format to EMBO Journal style, limiting to 10 authors et al. .

Further information is available in our Guide For Authors:

The revision must be submitted online within 90 days; please click on the link below to submit the revision online before 18th Mar 2021.

Referee #1:

The manuscript should be accepted for publication.

The minor comments:

- 1) Appendix Figure S4 has its figure legends labeled as D-E-F instead of A-B-C.
- 2) Figure EV1 is missing the inset F that is supposed to demonstrate the western blot for the Csnk1a1 KO.

Referee #3:

My concerns have been almost completely addressed and I congratulate the authors with their informative and insightful manuscript. A few points remain but I believe it would require little work to address them.

(numbers refer to the point-by-point rebuttal of the authors)

10. Figure EV1C partially contains the data I asked for, and this shows that Rex1GFP is already downregulated in 2i Klf2 KD cells. It looks very similar as in Tcf7l1 KO N24 cells. This does not demonstrate that the differentiation delay of Tcf7l1 KO cells is partially dependent on Klf2, as stated in the manuscript, since the Rex1-GFP expression levels are very similar between Klf2 KD in 2i and Klf2 KD in Tcf7l1 KO at N24. In other words, the Tcf7l1 KO + Klf2KD maintained Rex1-GFP at N24 at the same level as in 2i + Klf2KD, consistent with a block in differentiation. To draw proper conclusions, the authors should also analyse Rex1-GFP in Tcf7l1KO + Klf2KD in 2i, as I requested. To me it seems that the finding here is that the reduction in Rex1-GFP seen in Klf2KD is dependent on Jarid2.

I must say that the wt N24 Rex1-GFP in EV1C looks quite different than in Fig 2B. Variations between experiments do occur of course, but that means that all controls should be present in the same experiment, i.e. fig 2B should include 2i samples.

12. The authors addressed my comment very well but chose not to include this data in the manuscript, why not? A sentence stating that the effects were not caused by effects on cell death (with data in supplement) would enhance the results.

In addition, the discussion draws several conclusions that are not supported by the data and should be modified or removed:

- Line 519 (already mentioned in the first review): "Our data support a cell fate transition model in which extinction of cell identity induced by the interplay of signalling and the chromatin modification machinery precedes the initiation of a novel cell fate." No criteria or assays to determine extinction of cell identity or initiation of a novel cell fate are presented so this conclusion is not supported.
- Line 531: "Our data suggests that the exit from naïve pluripotency requires coordinated activity changes of independent pathways under control of separable genetic networks. The observation that single gene depletion is not sufficient to prevent exit from the naïve state is consistent with this interpretation". The authors should consider here that they switched two signalling pathways to induce the naïve-formative transition, Wnt and Mek. If their statement, that coordinated changes of independent pathways are required for naïve exit, is true, than switching only one of these signalling pathways should not induce exit from the naïve state. This is not demonstrated and I don't think this is the case.

Some further suggestions regarding the discussion, up to the discretion of the authors.

-The discussion regarding reprogramming in the first paragraph is confusing since the manuscript concerns a differentiation rather than a reprogramming step.

-Line 596: " suggests that state-of-the-art self-renewal culture conditions may not yet capture the naïve state perfectly". The authors may have a look at Choi et al (2017 Nature 548: 219) and Yagi et al (2017 Nature 548, 224) to see what this state-of-the-art condition does to naïve pluripotency.

-I found these statements in contradiction: " Oct4 and Sox2, which are key TFs for both the naïve and formative states, provide cornerstones for both the naïve and formative regulatory networks to ensure lineage fidelity during differentiation" (line 517) versus "These programmed cell state transitions in development do not rely upon instructions delivered by one or two master transcription factors." (line 601).

The authors performed the requested editorial changes.

Point-to-point-response:

Referee #1:

The manuscript should be accepted for publication.

The minor comments:

1) Appendix Figure S4 has its figure legends labeled as D-E-F instead of A-B-C.

R: corrected.

2) Figure EV1 is missing the inset F that is supposed to demonstrate the western blot for the Csnk1a1 KO.

R: Loss of Csnk1a1 protein is shown in Appendix Fig S2A and referred to in line 77.

Referee #3:

My concerns have been almost completely addressed and I congratulate the authors with their informative and insightful manuscript. A few points remain but I believe it would require little work to address them.

(numbers refer to the point-by-point rebuttal of the authors)

10. Figure EV1C partially contains the data I asked for, and this shows that Rex1GFP is already downregulated in 2i Klf2 KD cells. It looks very similar as in Tcf7l1 KO N24 cells. This does not demonstrate that the differentiation delay of Tcf7l1 KO cells is partially dependent on Klf2, as stated in the manuscript, since the Rex1-GFP expression levels are very similar between Klf2 KD in 2i and Klf2 KD in Tcf7l1 KO at N24. In other words, the Tcf7l1 KO + Klf2KD maintained Rex1-GFP at N24 at the same level as in 2i + Klf2KD, consistent with a block in differentiation. To draw proper conclusions, the authors should also analyse Rex1-GFP in Tcf7l1KO + Klf2KD in 2i, as I requested.

To me it seems that the finding here is that the reduction in Rex1-GFP seen in Klf2KD is dependent on Jarid2.

R: The fact that *Klf2* is a key downstream target of *Tcf7l1* is in line with previous observations (Martello et al., 2012). In the context of our paper, the interaction between *Klf2* and *Tcf7l1* serves as a positive control. The key message of Figures 2b and EV1C is to show that the differentiation defect

of *Jarid2* KOs is independent of *Klf2*, in line with the fact that *Jarid2* KO cells at N24 do not show elevated *Klf2* levels (Fig 2A). Data shown in Fig 1B clearly illustrates this point.

I must say that the wt N24 Rex1-GFP in EV1C looks quite different than in Fig 2B. Variations between experiments do occur of course, but that means that all controls should be present in the same experiment, i.e. fig 2B should include 2i samples.

R: We agree with the reviewer that there is some variation between Rex1GFP based data obtained at different timepoints. In the case of Fig 2B and EV1C, Fig EV1C was obtained at the request of the reviewer during the revision of the manuscript using a different batch of media. We frequently observe N2B27 batch dependent differentiation kinetics. However, this has no impact on the interpretation of data. All experiments are internally controlled by Rex1GFP levels measured at N24, which allows us to determine the relative differentiation defect compared to WT cells. Showing Rex1GFP levels in 2i would provide no further information and unnecessarily complicate the Figures.

12. The authors addressed my comment very well but chose not to include this data in the manuscript, why not? A sentence stating that the effects were not caused by effects on cell death (with data in supplement) would enhance the results.

R: We agree with the reviewer and have included data on growth of WT, *Jmjd1c* and *Tcf7l1* KO cells in 2i, N24 and N30 in Appendix Fig S5H, referred to in the text in line 217.

In addition, the discussion draws several conclusions that are not supported by the data and should be modified or removed:

-Line 519 (already mentioned in the first review): "Our data support a cell fate transition model in which extinction of cell identity induced by the interplay of signalling and the chromatin modification machinery precedes the initiation of a novel cell fate." No criteria or assays to determine extinction of cell identity or initiation of a novel cell fate are presented so this conclusion is not supported.

R: We believe that our data are in line with this proposal made by Austin Smith (e.g. Austin Smith, Development, 2017). However, we agree that we do have no formal criteria to determine where one cell identity ends and the next one starts. In fact, such an exact pinpointing of specific cellular identities in a dynamic system is extremely difficult, if at all possible, based on expression profiles alone. We have therefore deleted this statement.

-Line 531: "Our data suggests that the exit from naïve pluripotency requires coordinated activity changes of independent pathways under control of separable genetic networks. The observation that single gene depletion is not sufficient to prevent exit from the naïve state is consistent with this interpretation". The authors should consider here that they switched two signalling pathways to induce the naïve-formative transition, Wnt and Mek. If their statement, that coordinated changes of independent pathways are required for naïve exit, is true, than switching only one of these signalling pathways should not induce exit from the naïve state. This is not demonstrated and I don't think this is the case.

R: The reviewer is correct in that presence of a single inhibitor (of the 2i components) is insufficient to sustain naïve pluripotency. However, we disagree with the reviewer's statement. The fact that in our model cell fate change requires coordinated activity changes of separable pathways does not imply that each individual pathway is essential for the cell fate transition. In our manuscript we identify these cooperating mechanisms that are involved in mediating proper exit from naïve pluripotency. Fgf/Erk and Wnt signalling are only two of the pathways that are known to regulate self-renewal and differentiation.

To clarify this point we have changed the text to:

Our data suggests that the exit from naïve pluripotency involves coordinated activity changes of independent but functionally partially redundant pathways under the control of separable genetic networks. (l. 519).

Some further suggestions regarding the discussion, up to the discretion of the authors.

-The discussion regarding reprogramming in the first paragraph is confusing since the manuscript concerns a differentiation rather than a reprogramming step.

R: We use the example of reprogramming to illustrate the point that transcription factors have been proposed as key drivers of cell identity decisions.

-Line 596: " suggests that state-of-the-art self-renewal culture conditions may not yet capture the naïve state perfectly". The authors may have a look at Choi et al (2017 Nature 548: 219) and Yagi et al (2017 Nature 548, 224) to see what this state-of-the-art condition does to naïve pluripotency.

R: The papers cited by the reviewer show some erosion of imprinting marks in 2i cultures. We think that this is not relevant in the context of our discussion, which doesn't address imprinting.

-I found these statements in contradiction: " Oct4 and Sox2, which are key TFs for both the naïve and formative states, provide cornerstones for both the naïve and formative regulatory networks to ensure lineage fidelity during differentiation" (line 517) versus "These programmed cell state transitions in development do not rely upon instructions delivered by one or two master transcription factors." (line 601).

R: We don't believe this is a contradiction. Oct4 and Sox2 are key transcription factors for naïve to primed pluripotent cells. We have added the sentence '*TF circuits are crucial to maintain cell identity and to ensure proper lineage identity, but our data suggests that lineage specific TFs do not stand at the top of cell fate decision hierarchies.*' to the discussion for clarification (l. 508).

Dear Martin,

Thank you for submitting the revised version of your manuscript. I have now evaluated your amended manuscript and concluded that the remaining minor concerns have been sufficiently addressed.

Thus, I am pleased to inform you that your manuscript has been accepted for publication in the EMBO Journal.

Please note that it is EMBO Journal policy for the transcript of the editorial process (containing referee reports and your response letter) to be published as an online supplement to each paper. I would accordingly like to ask for your consent to keeping the additional reviewer figures included in this file.

Also in case you might NOT want the transparent process file published at all, you will also need to inform us via email immediately. More information is available here:
http://emboj.embopress.org/about#Transparent_Process

Please note that in order to be able to start the production process, our publisher will need and contact you regarding the following forms:

- PAGE CHARGE AUTHORISATION (For Articles and Resources)

[http://onlinelibrary.wiley.com/journal/10.1002/\(ISSN\)1460-2075/homepage/tej_apc.pdf](http://onlinelibrary.wiley.com/journal/10.1002/(ISSN)1460-2075/homepage/tej_apc.pdf)

- LICENCE TO PUBLISH (for non-Open Access)

Your article cannot be published until the publisher has received the appropriate signed license agreement. Once your article has been received by Wiley for production you will receive an email from Wiley's Author Services system, which will ask you to log in and will present them with the appropriate license for completion.

- LICENCE TO PUBLISH for OPEN ACCESS papers

Authors of accepted peer-reviewed original research articles may choose to pay a fee in order for their published article to be made freely accessible to all online immediately upon publication. The EMBO Open fee is fixed at \$5,200 (+ VAT where applicable).

We offer two licenses for Open Access papers, CC-BY and CC-BY-NC-ND.

For more information on these licenses, please visit: <http://creativecommons.org/licenses/by/3.0/> and http://creativecommons.org/licenses/by-nc-nd/3.0/deed.en_US

- PAYMENT FOR OPEN ACCESS papers

You also need to complete our payment system for Open Access articles. Please follow this link and select EMBO Journal from the drop down list and then complete the payment process:

https://authorservices.wiley.com/bauthor/onlineopen_order.asp

Notably, please be reminded that under the DEAL agreement of European scientific institutions with our publisher Wiley, you could be eligible for free publication of your article in the open access format. Please contact either the administration at your institution or Wiley (embojournal@wiley.com) to clarify further questions.

Should you be planning a Press Release on your article, please get in contact with embojournal@wiley.com as early as possible, in order to coordinate publication and release dates.

On a different note, I would like to alert you that EMBO Press is currently developing a new format for a video-synopsis of work published with us, which essentially is a short, author-generated film explaining the core findings in hand drawings, and, as we believe, can be very useful to increase visibility of the work. This has proven to offer a nice opportunity for exposure i.p. for the first author of the study. Please see the following link for representative examples:
https://www.embopress.org/video_synopses

Finally, we have noted that the submitted version of your article is also posted on the preprint platform bioRxiv. We would thus appreciate if you could alert bioRxiv on the acceptance of this manuscript at The EMBO Journal in order to allow for an update of the entry status. Thank you in advance!

If you have any questions, please do not hesitate to call or email the Editorial Office.

Kind regards,

Daniel

Daniel Klimmeck, PhD
Senior Editor
The EMBO Journal
EMBO
Postfach 1022-40
Meyerhofstrasse 1
D-69117 Heidelberg

Corresponding Author Name: Martin Leeb

Journal Submitted to: EMBO J

Manuscript Number: EMBOJ-2020-105772